# LANGUAGE IN THE FLOW OF TIME: TIME-SERIES-PAIRED TEXTS WEAVED INTO A UNIFIED TEMPORAL NARRATIVE

**Zihao Li**[1], **Xiao Lin**[1], **Zhining Liu**[1], **Jiaru Zou**[1], **Ziwei Wu**[1], **Lecheng Zheng**[1], **Dongqi Fu**[2]
**Yada Zhu**[3], **Hendrik Hamann**[3], **Hanghang Tong**[1], **Jingrui He**[1]
[1]University of Illinois Urbana-Champaign, [2]Meta, [3]IBM Research

## ABSTRACT

While many advances in time series models focus exclusively on numerical data, research on multimodal time series, particularly those involving contextual textual information, remains in its infancy. With recent progress in large language models and time series learning, we revisit the integration of paired texts with time series through the *Platonic Representation Hypothesis* (Huh et al., 2024), which posits that representations of different modalities converge to shared spaces. In this context, we identify that time-series-paired texts may naturally exhibit periodic properties that closely mirror those of the original time series. Building on this insight, we propose a novel framework, Texts as Time Series (TaTS), which considers the time-series-paired texts to be auxiliary variables of the time series. TaTS can be plugged into *any* existing numerical-only time series models and effectively enable them to handle time series data with paired texts. Through extensive experiments on both multimodal time series forecasting and imputation tasks across benchmark datasets with various existing time series models, we demonstrate that TaTS can enhance multimodal predictive performance without modifying model architectures. Our Code is available at `https://github.com/iDEA-iSAIL-Lab-UIUC/TaTS`.

## 1 INTRODUCTION

Time series modeling plays an important role in a wide range of real-world applications, including finance (Sezer et al., 2020), healthcare (Zhang et al., 2024b), climate (Fu et al., 2024), and energy systems (Kotzur et al., 2017; Li et al., 2015). While extensive research has focused on approaches that rely solely on the numerical values of time series (Zhou et al., 2021; Liu et al., 2024c; Wu et al., 2023b; Wang et al., 2024c), real-world scenarios often involve additional modalities that co-occur with the time series and can provide valuable complementary information (De Baets & Harvey, 2023; Rai et al., 2023; Kyei & Antwi, 2017; Wang et al., 2024d).

In scenarios like pandemic policymaking, economic planning, or investment strategies, textual information can provide explanations, updates, or external factors that influence the underlying numerical patterns. However, research on effectively leveraging data from other modalities paired with time series remains in its early stages. In this work, we focus on **time-series paired with texts at each timestamp**, a common data format where textual descriptions are associated with time series at each timestamp in a parallel manner, as illustrated in Figure 3 (left). For instance, during a pandemic, infection rates are often accompanied by government announcements and news reports in real-time (Cinelli et al., 2020). On the one hand, numerical-only models overlook valuable contextual information that may influence or explain the patterns in the time series. On the other hand, the current state-of-the-art approach (Liu et al., 2024a) disregards the unique positional characteristics that time-series-paired texts may inherently possess. Such limitations raise a pivotal question:

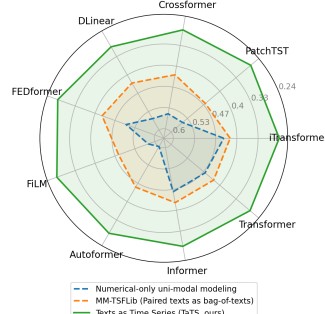

Figure 1: Mean Square Error of modeling frameworks of time series with paired texts. Full results in Appendix E.2.

*What unique attributes may characterize time-series-paired texts, and how can they be systematically integrated to improve time series modeling and predictions?*

In this paper, motivated by the *Platonic Representation Hypothesis (PRH)* (Huh et al., 2024), we pioneer the exploration of effectively leveraging paired texts to enrich time series analysis. We identify an intriguing phenomenon, which we term **"Chronological Textual Resonance"** (CTR): depending on data quality, time-series-paired texts may exhibit periodic patterns that closely reflect the temporal dynamics of their corresponding numerical time series. More specifically, despite variations in expressions, the hidden representations of two periodicity-lagged texts associated with time series may demonstrate high similarities, revealing a deeper alignment between textual and numerical modalities. We attribute this phenomenon to the fact that the paired texts inherently evolve in response to the dynamics of the time series itself. Further, we introduce **TT-Wasserstein**, a new metric designed to measure the level of CTR and quantify alignment quality.

Building on these insights, we propose **Texts as Time Series** (TaTS), a simple yet effective framework for integrating paired texts to enhance multimodal time series modeling. Previous studies have shown that different variables in one multivariate time series exhibit similar periodicity properties (Zhang & Yan, 2023; Wang et al., 2024d; Yi et al., 2024), and CTR suggests that time-series-paired texts follow a similar pattern. This observation implies that **paired texts can be considered as special auxiliary variables to augment the original time series**. Motivated by this, TaTS first transforms the paired textual information into a lower-dimensional representation, then combines the original time series with the textual representations as new variables to form an augmented time series. This augmented time series is subsequently fed into existing time series models, allowing them to capture both numerical and textual dynamics. TaTS offers two key benefits: (i) it effectively captures the evolving positional characteristics of texts paired with a time series; and (ii) it functions as a plug-in module, maintaining compatibility with existing time series models. Empirically, the proposed TaTS achieves state-of-the-art performance on both forecasting and imputation tasks. Notably, we observe that a higher CTR level (i.e., a lower TT-Wasserstein) correlates with greater improvements compared to numerical-only modeling. In summary, our contributions are:

- We revisit multimodal time series with the PRH and uncover a previously overlooked phenomenon, termed *Chronological Textual Resonance (CTR)*, that time-series-paired texts may exhibit periodic patterns closely aligned with their corresponding numerical time series. Further, we propose TT-Wasserstein to quantify the level of CTR and the alignment quality.

- Based on this phenomenon, we propose a plug-and-play multimodal time series forecasting framework, *Texts as Time Series (TaTS)*, which transforms text representations into auxiliary variables, seamlessly integrating them into existing time series models.

- Experiments on diverse benchmark datasets and multiple time series models demonstrate TaTS's superior performance without requiring modifications to model architectures.

## 2 PRELIMINARY

We use calligraphic letters (e.g., $\mathcal{A}$) for sets and bold capital letters for matrices (e.g., $\mathbf{A}$). For matrix indices, $\mathbf{A}[i, j]$ denotes the entry in the $i^{\text{th}}$ row and the $j^{\text{th}}$ column. For a vector $\mathbf{v}$, $v[i : j]$ represents the sub-vector sliced from the $i^{\text{th}}$ to the $j^{\text{th}}$ position, inclusively. $\mathbf{A}[i, :]$ returns the $i^{th}$ row in $\mathbf{A}$ and $\mathbf{A}[: i]$ returns the first $i$ rows of $\mathbf{A}$. In this paper, we focus on both forecasting and imputation.

**Time Series Forecasting.** A time series is denoted as $\mathbf{X} = \{\mathbf{x}_1, \mathbf{x}_2, \dots, \mathbf{x}_N\} \in \mathbb{R}^{T \times N}$, where $T$ represents the number of time steps and $N$ denotes the number of variables. $\mathbf{x}_i$ is the time series sequence of the $i^{\text{th}}$ variable. When $N > 1$, the time series is referred to as a multivariate time series. Let $\mathbf{X}_{a:b}$ represent the time slice of the series from timestamp $a$ to $b$, i.e., $\mathbf{X}_{a:b} = \{\mathbf{x}_1[a : b], \mathbf{x}_2[a : b], \dots, \mathbf{x}_N[a : b]\}$. The task of time series forecasting is to predict the future $H$ steps:

$$\widehat{\mathbf{X}}_{T+1:T+H} = \mathcal{F}(\mathbf{X}_{1:T}; \theta_{\text{forecast}}) \in \mathbb{R}^{H \times N}, \tag{1}$$

where $\mathcal{F}$ denotes the mapping function, and $\theta_{\text{forecast}}$ denotes the learnable parameters of $\mathcal{F}$.

**Time Series Imputation.** The goal of imputation is to estimate missing values in the time series $\mathbf{X}$, where the missing entries are denoted by a binary mask $\mathbf{M} \in \{0, 1\}^{T \times N}$. Specifically, $\mathbf{M}_{t,n} = 1$ indicates $\mathbf{X}_{t,n}$ being observed, $\mathbf{M}_{t,n} = 0$ indicates $\mathbf{X}_{t,n}$ being missing. Formally,

$$\widehat{\mathbf{X}}^{\text{Imputed}} = \mathcal{G}(\mathbf{X} \odot \mathbf{M}, \mathbf{M}; \theta_{\text{impute}}) \in \mathbb{R}^{T \times N}, \tag{2}$$

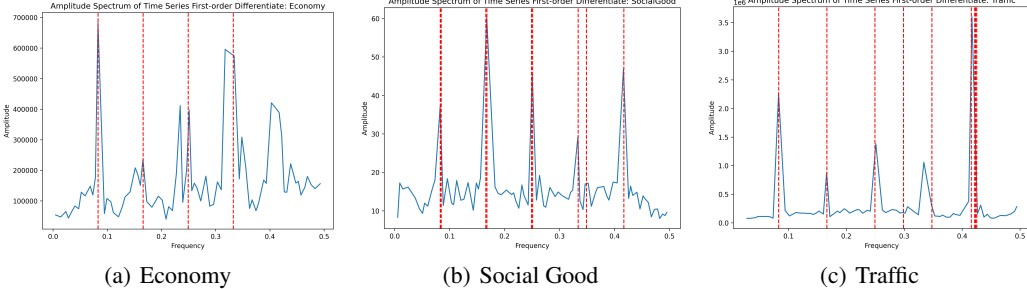

(a) Economy        (b) Social Good        (c) Traffic

Figure 2: By overlaying the top frequencies of paired texts (vertical dashed lines) onto the amplitude spectrum of the time series, it is observed that the time-series-paired texts exhibit similar periodic properties that closely mirror those of the original time series. We term this phenomenon *Chronological Textual Resonance*. More Details are provided in Appendix A.

where $\widehat{\mathbf{X}}^{\text{Imputed}}$ represents the imputed time series, $\mathcal{G}$ denotes the imputation function, $\theta_{\text{impute}}$ denotes its learnable parameters, and $\odot$ represents the element-wise multiplication. The imputation process aims to recover the missing entries such that $\widehat{\mathbf{X}}^{\text{Imputed}} \approx \mathbf{X}$ with respect to the actual $\mathbf{X}$ values.

**Extending Time Series with Paired Texts.** In addition to the numerical time series $\mathbf{X} \in \mathbb{R}^{T \times N}$, the dataset $\mathcal{D} = \{\mathbf{X}, \mathbf{S}\}$ includes textual information $\mathbf{S} = \{s_1, s_2, \dots, s_T\}$, where each $s_t$ represents the text at timestamp $t$. Each $s_t$ is a string that can be tokenized into a sequence of tokens, i.e., $\text{Tokenize}(s_t) = \{w_{t,1}, w_{t,2}, \dots, w_{t,L_t}\}$, where $L_t$ denotes the number of tokens in the text at time $t$. The textual data can be transformed into numerical representations using a textual encoder $\mathcal{H}_{\text{text}}$:

$$\mathbf{e}_t = \mathcal{H}_{\text{text}}(s_t; \theta_{\text{text}}) \in \mathbb{R}^{d_{\text{text}}}, \tag{3}$$

where $\mathbf{e}_t$ is the encoded text representation at time $t$, $d_{\text{text}}$ is text embedding dimension, and $\theta_{\text{text}}$ are encoder parameters. In this work, we leverage pre-trained large language models to encode the texts.

## 3 CHRONOLOGICAL TEXTUAL RESONANCE (CTR)

The association of time series at each timestamp may imbue time-series-paired texts with unique characteristics that can be effectively harnessed through an appropriate design.

**The Platonic Representation Hypothesis.** PRH (Huh et al., 2024) posits that different modalities describing the same object converge towards a shared, latent representation. Extending this hypothesis, if time series and paired text both describe the same changing event, their representations are dynamic projections from a common underlying source, and should exhibit similar periodic properties.

To illustrate this hypothesis for time series with paired texts, we analyze three real-world datasets (Liu et al., 2024a), including (i) Economy: The time series represents trade data of the U.S., while the texts describe the general economic conditions of the country. (ii) Social Good: The time series captures the unemployment rate in the U.S., and the texts include detailed unemployment reports. (iii) Traffic: The time series reflects monthly travel volume trends from the U.S. Department of Transportation, with corresponding texts derived from traffic volume reports issued by the same department.

For each dataset $\mathcal{D} = \{\mathbf{X}, \mathbf{S}\}$, we employ Fourier Transform (Nussbaumer, 1982; Sneddon, 1995) to analyze the frequency components of time series data and identify its dominant periodic components, illustrated by the blue curves in Figure 2. To examine the periodicity of texts, we embed each $s_t \in \mathbf{S}$ to obtain the text embedding $e_t$ at timestamp $t$. Then, we compute their lag-similarity, defined as $d_l = \sum_t \cos(e_t, e_{t+L})$ where $L$ is the lag and $\cos(\cdot, \cdot)$ represents the cosine similarity. If the text embeddings exhibit a significant periodic pattern, the lag-similarity $d_l$ will also fluctuate periodically as the lag $l$ increases (proof in Proposition A.1). Finally, we identify major frequencies (with the largest amplitudes) of the texts by applying FFT to text lag-similarity, and mark them with red dashed lines, as shown in Figure 2. Detailed process is provided in Appendix A. We find that the major frequencies of the paired texts closely match those of the time series. Specifically, the paired texts also show periodicity of 12 (frequency 0.083) for monthly sampled time series, indicating that the paired texts exhibit periodicity that is strongly aligned with the temporal dynamics of the time series.

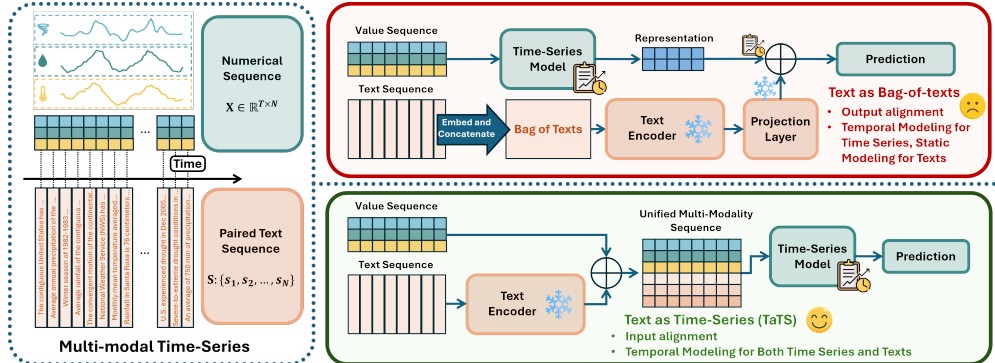

Figure 3: Texts as Time Series (TaTS) framework. As paired texts may exhibit behaviors similar to accompanying variables in a time series, TaTS transforms the paired texts into auxiliary variables. These variables augment the numerical sequence, forming a unified multimodal sequence that can be seamlessly integrated into any existing time series model.

**Why may CTR happen?** We present three key reasons for the observed alignment in periodicity between time series and their paired texts: (i) **Shared External Drivers**: Both the time series and their paired texts are often influenced by common external factors, such as seasonal changes, recurring events, or societal and economic cycles. These shared drivers naturally induce periodicity in both modalities. (ii) **Influence of Time Series on Texts**: Paired texts often serve as contextual reflections of the underlying time series, adapting and evolving in response to numerical trends. For instance, news articles or government reports accompanying economic indicators are frequently updated in response to the numerical trends. (iii) **Texts Contain Additional Variables with Aligned Periodicity**: Paired texts often include additional variables that are closely related to the time series. For example, if the time series represents economic GDP data, the accompanying texts may reference related variables such as stock market indices or inflation rates. These related variables often exhibit periodicity patterns aligned with the time series and affect the periodicity of the paired texts.

Table 1: TT-Wasserstein measure of Time-MMD (Liu et al., 2024a) datasets and their random shuffle.

| Dataset | | | Monthly Sampled | | | | Weekly Sampled | | Daily Sampled |
| --- | --- | --- | --- | --- | --- | --- | --- | --- | --- |
| | Agriculture | Climate | Economy | Security | Social Good | Traffic | Energy | Health | Environment |
| Original | 0.026 | 0.025 | 0.022 | 0.049 | 0.027 | 0.035 | 0.307 | 0.233 | 0.302 |
| TS shuffled | 0.088 | 0.032 | 0.098 | 0.054 | 0.069 | 0.102 | 0.320 | 0.268 | 0.358 |
| Text shuffled | 0.106 | 0.037 | 0.099 | 0.053 | 0.072 | 0.104 | 0.312 | 0.277 | 0.364 |

**Quantifying CTR level.** Of course we cannot expect every time series with paired texts at each timestamp to exhibit periodicity similarity. In some cases, texts aligned with timestamps may not provide meaningful or complementary information, for example, daily lottery winning numbers. To this end, we propose a new metric for CTR, TT-Wasserstein, defined as the Wasserstein distance between the normalized spectra of time series and texts. Formally, based on Wasserstein distance $W(P, Q)$ (Kantorovich, 1960), for time series with paired texts dataset $\mathcal{D} = \{\mathbf{X}, \mathbf{S}\}$, after computing normalized frequencies and amplitudes $\tilde{f}_{\text{texts}}, \tilde{f}_{\text{ts}}, \tilde{a}_{\text{texts}}, \tilde{a}_{\text{ts}}$,

$$\text{TT-Wasserstein}(\mathcal{D}) = W(P_{\text{texts}}, P_{\text{ts}}) = \inf_{\gamma \in \Pi(P_{\text{text}}, P_{\text{ts}})} \sum_{i=1}^{n} \sum_{j=1}^{m} \gamma_{ij} \cdot \left| \tilde{f}_{\text{texts}}^{(i)} - \tilde{f}_{\text{ts}}^{(j)} \right|$$

$$\text{Subject to} \quad \gamma_{ij} \geq 0, \quad \sum_{j=1}^{m} \gamma_{ij} = \tilde{a}_{\text{texts}}^{(i)}, \quad \sum_{i=1}^{n} \gamma_{ij} = \tilde{a}_{\text{ts}}^{(j)}$$

(4)

Intuitively, TT-Wasserstein($\mathcal{D}$) quantifies the discrepancy between the spectral distributions of paired texts and time series data. By design, a lower value of TT-Wasserstein($\mathcal{D}$) should indicate a higher alignment between the textual and numerical modalities. To validate this, we compute our TT-Wasserstein measure of Time-MMD (Liu et al., 2024a) datasets, whose web-retrieved texts are filtered, disentangled, and summarized for relevance and alignment. We also compute TT-Wasserstein

on time-series-shuffled or text-shuffled versions of the datasets to disrupt alignment. As shown in Table 1, the corrupted Time-MMD datasets yield much larger Wasserstein distances compared to the original datasets. This finding suggests that TT-Wasserstein can serve as an indicator for the alignment between modalities and a gauge of dataset quality. In Section 5, we will show that TT-Wasserstein can even predict the potential effectiveness of our proposed framework. Additionally, because TT-Wasserstein is an empirical statistical metric, we further analyze its sensitivity and estimation stability in Appendix A.1.

## 4 TEXTS AS TIME SERIES

An overview of the proposed U̲nderline{Texts} a̲s T̲ime S̲eries (TaTS) is illustrated in Figure 3.

**Concurrent Texts are Secretly Auxiliary Variables.** As we elucidated in Section 3, the properties of concurrent texts closely align with those of variables in a multivariate time series: similar to numerical variables, concurrent texts are influenced by shared external drivers and interact dynamically with the time series. Furthermore, mapping concurrent texts to structured variables enables capturing hidden variables embedded within the concurrent texts.

Given the dataset $\mathcal{D} = \{\mathbf{X} = \{\mathbf{x}_1, \mathbf{x}_2, \ldots, \mathbf{x}_N\}, \mathbf{S} = \{s_1, s_2, \ldots, s_T\}\}$, TaTS first embed the texts using a text encoder $\mathcal{H}_{\text{text}}$ to obtain text embeddings $\mathbf{E} = \{e_1, e_2, \ldots, e_T\} \in \mathbb{R}^{d_{\text{text}} \times T}$. Since the text embedding dimension $d_{\text{text}}$ is typically much larger than the number of variables in the time series, we reduce the dimensionality of the text embeddings by applying a Multi-Layer Perceptron (MLP), mapping them into a lower-dimensional space of reduced dimensionality $d_{\text{mapped}}$.

$$\mathbf{z}_t = \text{MLP}(\mathbf{e}_t; \theta_{\text{MLP}}) \in \mathbb{R}^{d_{\text{mapped}}} \tag{5}$$

**Unifying by Plugging-in a Time Series Model.** The resulting mapped embeddings $\mathbf{Z} = \{\mathbf{z}_1, \mathbf{z}_2, \ldots, \mathbf{z}_T\} \in \mathbb{R}^{d_{\text{mapped}} \times T}$ are then treated as auxiliary variables in the time series. Specifically, $\mathbf{Z}$ is concatenated with $\mathbf{X}$ to form a unified multimodal sequence:

$$\mathbf{U} = [\mathbf{X}; \mathbf{Z}^{\mathsf{T}}]_{\text{dim}=1} \in \mathbb{R}^{T \times (N + d_{\text{mapped}})} \tag{6}$$

The unified sequence $\mathbf{U}$ is then passed into an existing time series model for downstream tasks. Here, we formulate the example of forecasting the next $H$ steps of the time series

$$\widehat{\mathbf{X}}_{T+1:T+H} = \mathcal{F}(\mathbf{U}_{1:T}; \theta_{\text{forecast}})[:N] \in \mathbb{R}^{H \times N} \tag{7}$$

where $\mathcal{F}(\cdot; \theta_{\text{forecast}})$ denotes the time series forecasting model with parameters $\theta_{\text{forecast}}$, and $[:N]$ extracts the first $N$ variables corresponding to the original time series.

Finally, we joint train the time series model $\theta_{\text{forecast}}$ as well as the mapping MLP $\theta_{\text{MLP}}$ using the Mean Squared Error (MSE) loss.

$$\mathcal{L}_{\text{forecast}}(\mathbf{X}, \widehat{\mathbf{X}}) = \frac{1}{H \cdot N} \sum_{t=T+1}^{T+H} \sum_{i=1}^{N} \left(\mathbf{X}_{t,i} - \widehat{\mathbf{X}}_{t,i}\right)^2 \tag{8}$$

where $\mathbf{X}_{t,i}$ and $\widehat{\mathbf{X}}_{t,i}$ represent the ground truth and predicted values of the $i^{\text{th}}$ variable at time step $t$, respectively. TaTS for imputation follows a similar process and is omitted here due to space constraints. Detailed algorithms for both forecasting and imputation are provided in Appendix C.

To conclude this section, we note that while some existing literatures make similar but limited attempts to model covariates in multivariate time series or multiple time series. Our TaTS framework differentiates from those approaches by unifying multi-modality using deep learning architectures. In Section 5, we compare TaTS with these existing approaches.

## 5 EXPERIMENT

In this section, we empirically validate the effectiveness of the proposed TaTS framework. In particular, we use GPT2 (Radford et al., 2019) encoder to embed the paired texts. We also validate the performance of TaTS with other text encoders across different datasets in section 5.2 and Appendix E.7. Implementation details are provided in Appendix D.4.

Table 2: Time-series forecasting performance. Compared to numerical-only unimodal modeling and MM-TSFLib, TaTS consistently and significantly enhances existing time series models to effectively handle paired texts. The results are averaged across all prediction lengths. Full results reported in Appendix E.3. *Promotion* (positive or negative) denotes the percentage reduction (or increase) in MSE or MAE achieved by TaTS compared to the best-performing baseline.

| Models | | iTransformer (2024c) | | PatchTST (2023) | | Crossformer (2023) | | DLinear (2023) | | FEDformer (2022b) | | FiLM (2022a) | | Autoformer (2021) | | Informer (2021) | | Transformer (2017) | |
|---|---|---|---|---|---|---|---|---|---|---|---|---|---|---|---|---|---|---|---|
| Metric | | MSE | MAE | MSE | MAE | MSE | MAE | MSE | MAE | MSE | MAE | MSE | MAE | MSE | MAE | MSE | MAE | MSE | MAE |
| Agriculture | Uni-modal | 0.122 | 0.251 | 0.120 | 0.247 | 0.323 | 0.406 | 0.223 | 0.354 | 0.138 | 0.286 | 0.139 | 0.256 | 0.158 | 0.297 | 0.599 | 0.630 | 0.354 | 0.434 |
| | MM-TSFLib | 0.112 | 0.230 | 0.114 | 0.233 | 0.218 | 0.313 | 0.218 | 0.355 | 0.131 | 0.275 | 0.140 | 0.258 | 0.158 | 0.288 | 0.313 | 0.414 | 0.249 | 0.352 |
| | **TaTS (ours)** | 0.109 | 0.229 | 0.114 | 0.235 | 0.212 | 0.312 | 0.214 | 0.351 | 0.131 | 0.276 | 0.135 | 0.251 | 0.125 | 0.266 | 0.255 | 0.348 | 0.191 | 0.313 |
| | Promotion | 2.7% | 0.4% | 0.0% | -0.9% | 2.8% | 0.3% | 1.8% | 0.8% | 0.0% | -0.4% | 2.9% | 2.0% | 20.9% | 7.6% | 18.5% | 15.9% | 23.3% | 11.1% |
| Climate | Uni-modal | 1.183 | 0.871 | 1.220 | 0.895 | 1.124 | 0.837 | 1.190 | 0.872 | 1.192 | 0.893 | 1.270 | 0.911 | 1.131 | 0.865 | 1.110 | 0.841 | 1.092 | 0.839 |
| | MM-TSFLib | 1.044 | 0.810 | 1.030 | 0.806 | 1.002 | 0.772 | 1.104 | 0.837 | 1.011 | 0.797 | 1.179 | 0.871 | 1.053 | 0.827 | 1.001 | 0.792 | 0.998 | 0.783 |
| | **TaTS (ours)** | 1.028 | 0.804 | 1.004 | 0.798 | 0.938 | 0.755 | 0.931 | 0.759 | 0.926 | 0.760 | 0.945 | 0.772 | 0.980 | 0.789 | 0.930 | 0.756 | 0.920 | 0.753 |
| | Promotion | 1.5% | 0.7% | 2.5% | 1.0% | 6.4% | 2.2% | 15.7% | 9.3% | 8.4% | 4.6% | 19.8% | 11.4% | 6.9% | 4.6% | 7.1% | 4.5% | 7.8% | 3.8% |
| Economy | Uni-modal | 0.014 | 0.096 | 0.017 | 0.105 | 0.758 | 0.828 | 0.058 | 0.192 | 0.042 | 0.166 | 0.025 | 0.129 | 0.071 | 0.207 | 1.325 | 1.110 | 0.584 | 0.711 |
| | MM-TSFLib | 0.011 | 0.086 | 0.014 | 0.096 | 0.250 | 0.458 | 0.058 | 0.192 | 0.035 | 0.153 | 0.026 | 0.129 | 0.058 | 0.192 | 0.432 | 0.618 | 0.213 | 0.416 |
| | **TaTS (ours)** | 0.008 | 0.077 | 0.009 | 0.079 | 0.219 | 0.419 | 0.021 | 0.117 | 0.015 | 0.101 | 0.009 | 0.080 | 0.024 | 0.121 | 0.299 | 0.512 | 0.079 | 0.232 |
| | Promotion | 27.3% | 10.5% | 35.7% | 17.7% | 12.4% | 8.5% | 63.8% | 39.1% | 57.1% | 34.0% | 64.0% | 38.0% | 58.6% | 37.0% | 30.8% | 17.2% | 62.9% | 44.2% |
| Energy | Uni-modal | 0.269 | 0.375 | 0.269 | 0.376 | 0.293 | 0.406 | 0.291 | 0.396 | 0.240 | 0.351 | 0.278 | 0.385 | 0.319 | 0.428 | 0.309 | 0.425 | 0.297 | 0.405 |
| | MM-TSFLib | 0.267 | 0.378 | 0.272 | 0.379 | 0.291 | 0.407 | 0.289 | 0.395 | 0.238 | 0.354 | 0.279 | 0.385 | 0.320 | 0.428 | 0.301 | 0.413 | 0.293 | 0.405 |
| | **TaTS (ours)** | 0.265 | 0.376 | 0.258 | 0.371 | 0.279 | 0.394 | 0.283 | 0.388 | 0.237 | 0.355 | 0.271 | 0.379 | 0.314 | 0.430 | 0.284 | 0.396 | 0.279 | 0.395 |
| | Promotion | 0.7% | -0.3% | 4.1% | 1.3% | 4.1% | 3.0% | 2.1% | 1.8% | 0.4% | -1.1% | 2.5% | 1.6% | 1.6% | -0.5% | 5.6% | 4.1% | 4.8% | 2.5% |
| Environment | Uni-modal | 0.441 | 0.494 | 0.552 | 0.537 | 0.551 | 0.581 | 0.558 | 0.591 | 0.503 | 0.549 | 0.577 | 0.543 | 0.599 | 0.598 | 0.459 | 0.512 | 0.460 | 0.511 |
| | MM-TSFLib | 0.421 | 0.478 | 0.459 | 0.501 | 0.427 | 0.488 | 0.429 | 0.502 | 0.423 | 0.486 | 0.478 | 0.490 | 0.452 | 0.502 | 0.424 | 0.480 | 0.425 | 0.481 |
| | **TaTS (ours)** | 0.267 | 0.369 | 0.273 | 0.371 | 0.284 | 0.403 | 0.298 | 0.428 | 0.275 | 0.378 | 0.272 | 0.371 | 0.285 | 0.387 | 0.285 | 0.406 | 0.276 | 0.397 |
| | Promotion | 36.6% | 22.8% | 40.5% | 25.9% | 33.5% | 17.4% | 30.5% | 14.7% | 35.0% | 22.2% | 43.1% | 24.3% | 36.9% | 22.9% | 32.8% | 15.4% | 35.1% | 17.5% |
| Health | Uni-modal | 1.587 | 0.817 | 1.652 | 0.855 | 1.535 | 0.827 | 1.737 | 0.848 | 1.486 | 0.909 | 1.982 | 1.005 | 1.962 | 1.039 | 1.278 | 0.773 | 1.378 | 0.776 |
| | MM-TSFLib | 1.446 | 0.816 | 1.347 | 0.797 | 1.273 | 0.744 | 1.541 | 0.800 | 1.252 | 0.792 | 1.675 | 0.949 | 1.494 | 0.887 | 1.215 | 0.740 | 1.218 | 0.748 |
| | **TaTS (ours)** | 1.315 | 0.744 | 1.283 | 0.753 | 1.226 | 0.728 | 1.412 | 0.787 | 1.244 | 0.791 | 1.421 | 0.838 | 1.409 | 0.861 | 1.183 | 0.752 | 1.142 | 0.718 |
| | Promotion | 9.1% | 8.8% | 4.8% | 5.5% | 3.7% | 2.2% | 8.4% | 1.6% | 0.6% | 0.1% | 15.2% | 11.7% | 5.7% | 2.9% | 2.6% | -1.6% | 6.2% | 4.0% |
| Security | Uni-modal | 115.94 | 5.660 | 112.85 | 5.371 | 126.96 | 6.277 | 109.11 | 4.711 | 114.48 | 5.158 | 115.55 | 5.487 | 115.27 | 5.118 | 131.78 | 6.623 | 131.35 | 6.582 |
| | MM-TSFLib | 116.34 | 5.532 | 112.84 | 5.369 | 125.72 | 6.183 | 108.03 | 4.712 | 113.73 | 5.107 | 109.19 | 4.897 | 111.44 | 4.973 | 128.95 | 6.415 | 128.47 | 6.378 |
| | **TaTS (ours)** | 112.05 | 5.151 | 109.69 | 5.019 | 125.16 | 6.148 | 107.92 | 4.676 | 107.37 | 4.718 | 107.85 | 4.736 | 108.49 | 4.787 | 126.66 | 6.276 | 124.58 | 6.134 |
| | Promotion | 3.4% | 6.9% | 2.8% | 6.5% | 0.4% | 0.6% | 0.1% | 0.7% | 5.6% | 7.6% | 1.2% | 3.3% | 2.7% | 3.7% | 1.8% | 2.2% | 3.0% | 3.8% |
| Social Good | Uni-modal | 1.212 | 0.483 | 1.097 | 0.495 | 0.865 | 0.467 | 1.151 | 0.712 | 0.979 | 0.476 | 1.261 | 0.654 | 1.278 | 0.701 | 0.870 | 0.504 | 0.910 | 0.484 |
| | MM-TSFLib | 1.197 | 0.520 | 1.073 | 0.515 | 0.837 | 0.398 | 1.083 | 0.673 | 0.962 | 0.462 | 1.236 | 0.626 | 1.229 | 0.670 | 0.839 | 0.457 | 0.856 | 0.461 |
| | **TaTS (ours)** | 0.987 | 0.452 | 0.972 | 0.465 | 0.779 | 0.412 | 1.006 | 0.622 | 0.888 | 0.430 | 1.104 | 0.626 | 1.195 | 0.666 | 0.810 | 0.459 | 0.807 | 0.419 |
| | Promotion | 17.5% | 6.4% | 9.4% | 6.1% | 6.9% | -3.5% | 7.1% | 7.6% | 7.7% | 6.9% | 10.7% | 0.0% | 2.8% | 0.6% | 3.5% | -0.4% | 5.7% | 9.1% |
| Traffic | Uni-modal | 0.213 | 0.238 | 0.188 | 0.242 | 0.214 | 0.376 | 0.230 | 0.359 | 0.205 | 0.264 | 0.215 | 0.314 | 0.212 | 0.298 | 0.202 | 0.355 | 0.209 | 0.346 |
| | MM-TSFLib | 0.199 | 0.347 | 0.178 | 0.230 | 0.188 | 0.334 | 0.209 | 0.330 | 0.193 | 0.238 | 0.207 | 0.300 | 0.212 | 0.272 | 0.172 | 0.294 | 0.171 | 0.295 |
| | **TaTS (ours)** | 0.187 | 0.217 | 0.172 | 0.209 | 0.168 | 0.286 | 0.188 | 0.300 | 0.173 | 0.212 | 0.176 | 0.248 | 0.177 | 0.229 | 0.164 | 0.281 | 0.164 | 0.274 |
| | Promotion | 6.0% | 8.8% | 3.4% | 9.1% | 10.6% | 14.4% | 10.0% | 9.1% | 10.4% | 10.9% | 15.0% | 17.3% | 16.5% | 15.8% | 4.7% | 6.0% | 4.1% | 7.1% |

**Datasets.** We evaluate our framework on 18 real-world datasets from Time-MMD (Liu et al., 2024a), FNSPID Dong et al. (2024), and FNF Wang et al. (2024b). The datasets span diverse domains with sample frequencies ranging from daily to weekly and monthly. More details are in Appendix D.1.

**Time Series Models and Baselines.** To demonstrate the compatibility of TaTS with existing time series models, we integrate TaTS with 9 widely used models across different categories, including (i) Transformer-based models: iTransformer (Liu et al., 2024c), PatchTST (Nie et al., 2023), Crossformer (Zhang & Yan, 2023), Autoformer (Wu et al., 2021), Informer (Zhou et al., 2021) and Transformer (Vaswani et al., 2017). (ii) Linear models: DLinear (Zeng et al., 2023). (iii) Frequency-based models: FEDformer (Zhou et al., 2022b), FiLM (Zhou et al., 2022a).

We compare our TaTS framework with a wide array of baselines, including (a) numerical-only uni-modal modeling, which ignores the paired texts and utilizes only the numerical time series with the given time series model; (b) MM-TSFLib (Liu et al., 2024a), a recently library proposed together with Time-MMD datasets for multimodal time series forecasting; (c) covariate-based methods: N-BEATS (Oreshkin et al., 2020), N-HiTS (Challu et al., 2022); (d) convolution-based method TCN (Bai et al., 2018); (e) a recent multimodal time series foundation model ChatTime (Wang et al., 2025).

**Metrics.** We evaluate the performance of multimodal time series modeling using MSE, MAE, RMSE, MAPE, and MSPE. Due to space constraints, we report only the MSE and MAE results in the main paper, while the results for the other metrics are provided in Appendix E.

## 5.1 Main Results

**Improved Forecasting over Uni-modal Modeling and MM-TSFLib.** Table 2 presents the performance results for the time series forecasting task on Time-MMD datasets. For each dataset and time series model, we report the average performance across four different prediction lengths, and

Table 4: TaTS compared with several baselines on a variety of datasets. Best results are bolded and second-best results are underlined. Full results in Table 28.

| Methods | | TaTS (ours) + iTransformer | | TaTS (ours) + PatchTST | | TaTS (ours) + FiLM | | N-BEATS (2020) | | N-HiTS (2022) | | TCN (2018) | | ChatTime (2025) | | GPT4MTS (2024) | |
|---|---|---|---|---|---|---|---|---|---|---|---|---|---|---|---|---|---|
| Datasets | | MSE | MAE | MSE | MAE | MSE | MAE | MSE | MAE | MSE | MAE | MSE | MAE | MSE | MAE | MSE | MAE |
| Time-MMD: Multimodal Time Series (2024a) | Agriculture | **0.109** | **0.229** | 0.114 | 0.235 | 0.135 | 0.251 | 3.267 | 1.458 | 1.852 | 1.032 | 4.168 | 1.797 | 0.508 | 0.447 | 0.327 | 0.393 |
| | Climate | 1.028 | 0.804 | 1.004 | 0.798 | **0.945** | **0.772** | 1.093 | 0.861 | 1.103 | 0.858 | 1.098 | 0.866 | 1.568 | 1.019 | 1.127 | 0.873 |
| | Economy | **0.008** | **0.077** | 0.009 | 0.079 | 0.009 | 0.080 | 1.010 | 0.920 | 0.444 | 0.584 | 5.546 | 2.349 | 0.049 | 0.166 | 0.014 | 0.096 |
| | Energy | 0.265 | 0.376 | **0.258** | **0.371** | 0.271 | 0.379 | 0.329 | 0.424 | 0.372 | 0.463 | 0.430 | 0.512 | 0.305 | 0.417 | 0.269 | 0.378 |
| | Environment | **0.267** | **0.369** | 0.273 | 0.371 | 0.272 | 0.371 | 0.518 | 0.573 | 0.522 | 0.583 | 0.854 | 0.713 | 0.580 | 0.594 | 0.348 | 0.422 |
| | Health | 1.315 | **0.744** | **1.283** | 0.753 | 1.421 | 0.838 | 1.660 | 0.938 | 1.666 | 0.898 | 1.938 | 0.970 | 1.668 | 0.909 | 1.766 | 0.875 |
| | Security | 112.054 | 5.151 | 109.693 | 5.019 | **107.850** | **4.736** | 130.065 | 6.618 | 138.124 | 6.978 | 136.596 | 6.873 | 133.106 | 6.887 | 119.407 | 5.198 |
| | Social Good | 0.987 | **0.452** | 0.972 | 0.465 | 1.104 | 0.626 | 1.316 | 0.665 | 1.272 | 0.649 | 1.314 | 0.961 | 1.264 | 0.652 | 1.414 | 0.559 |
| | Traffic | 0.187 | 0.217 | **0.172** | **0.209** | 0.176 | 0.248 | 0.347 | 0.464 | 0.268 | 0.382 | 0.708 | 0.744 | 0.363 | 0.429 | 0.195 | 0.246 |
| FNSPID: Company Stock Price (2024) | Delta Airlines (DAL) | 0.087 | 0.197 | **0.086** | **0.192** | 0.095 | 0.201 | 0.286 | 0.444 | 0.226 | 0.379 | 0.294 | 0.466 | 0.098 | 0.202 | 0.093 | 0.201 |
| | IBM (IBM) | 0.564 | 0.501 | **0.550** | **0.490** | 0.891 | 0.695 | 1.105 | 0.777 | 1.215 | 0.806 | 1.936 | 1.123 | 0.602 | 0.536 | 0.639 | 0.525 |
| | JPMorgan (JPM) | **1.693** | **0.970** | 1.872 | 0.990 | 2.513 | 1.096 | 2.419 | 1.175 | 3.426 | 1.232 | 3.764 | 1.711 | 2.037 | 1.043 | 2.133 | 1.122 |
| | NVIDIA (NVDA) | **0.043** | **0.141** | 0.048 | 0.156 | 0.050 | 0.174 | 0.272 | 0.434 | 0.122 | 0.262 | 0.457 | 0.574 | 0.053 | 0.161 | 0.054 | 0.159 |
| | Pfizer (PFE) | **0.326** | **0.416** | 0.347 | 0.422 | 0.448 | 0.477 | 0.676 | 0.572 | 0.824 | 0.663 | 0.628 | 0.559 | 0.408 | 0.477 | 0.369 | 0.439 |
| | Tesla (TSLA) | 0.142 | 0.281 | 0.158 | 0.297 | **0.110** | **0.244** | 0.188 | 0.332 | 0.254 | 0.416 | 3.476 | 1.851 | 0.158 | 0.300 | 0.181 | 0.303 |
| FNF: Forecast with News (2024b) | Bitcoin Price | 2.609 | 1.112 | **2.339** | **1.045** | 2.775 | 1.142 | 141.738 | 10.355 | 159.019 | 10.535 | 57.826 | 6.744 | 3.590 | 1.344 | 2.721 | 1.152 |
| | Web Traffic | 17.744 | 2.712 | 17.964 | **2.653** | 18.110 | 2.772 | 22.233 | 3.017 | 23.263 | 3.011 | 21.626 | 2.842 | 20.748 | 2.909 | 19.260 | 2.827 |
| | Electricity Demand | 0.280 | 0.396 | **0.254** | **0.358** | 0.266 | 0.377 | 0.416 | 0.496 | 0.364 | 0.468 | 0.516 | 0.564 | 0.532 | 0.566 | 0.296 | 0.395 |

the full results for each prediction length are provided in Appendix E. For datasets with relatively few samples, we perform short-term forecasting with prediction lengths of $\{6, 8, 10, 12\}$. In contrast, for datasets with a larger number of samples, we perform long-term forecasting with prediction lengths of $\{48, 96, 192, 336\}$. From the results, compared to uni-modal modeling or MM-TSFLib, our TaTS consistently achieves the best performance across all datasets. Notably, by plugging in TaTS to various existing time series models, it achieves an average performance improvement of over $5\%$ on 6 out of 9 datasets and delivers a remarkable performance boost of over $30\%$ on the largest dataset, Environment. The results also demonstrate that TaTS is highly compatible with a wide range of existing time series forecasting models, consistently delivering performance improvements across all of them in both long-term forecasting and short-term forecasting tasks. We provide a visualization of the performance boost in Appendix E.2, showcasing the improvements achieved by different time series models on each dataset.

**Improved Time Series Imputation.** We evaluate the performance of our TaTS framework on the imputation task using the Climate, Economy, and Traffic datasets, each with an imputation length of 24. Though MM-TSFLib only supports forecasting tasks, we extend it to serve as an imputation baseline by applying a similar linear interpolation. We select one representative time series model from each category and present the results in Table 3. The results demonstrate that TaTS consistently enhances the imputation capabilities of existing time series models, achieving improvements of up to $30\%$ compared to baseline methods.

From the above results, effectively leveraging the text modality provides significant benefits when paired texts are available, and our TaTS framework achieves notable improvements over baseline methods that disregard positional information in time-series-paired texts.

Table 3: Imputation task performance. Full results in Appendix E.4. *Promotion*: percentage reduction by TaTS over the best-performing baseline.

| Models | | PatchTST (2023) | | DLinear (2023) | | FiLM (2022a) | |
|---|---|---|---|---|---|---|---|
| Metric | | MSE | MAE | MSE | MAE | MSE | MAE |
| Climate | Uni-modal | 1.111 | 0.846 | 0.969 | 0.801 | 1.123 | 0.829 |
| | MM-TSFLib | 1.010 | 0.821 | 0.963 | 0.802 | 1.130 | 0.833 |
| | **TaTS (ours)** | 0.878 | 0.720 | 0.912 | 0.757 | 0.820 | 0.718 |
| | Promotion | 13.1% | 12.3% | 5.3% | 5.5% | 27.0% | 13.4% |
| Economy | Uni-modal | 0.029 | 0.138 | 0.057 | 0.190 | 0.077 | 0.209 |
| | MM-TSFLib | 0.026 | 0.137 | 0.061 | 0.196 | 0.075 | 0.203 |
| | **TaTS (ours)** | 0.017 | 0.045 | 0.045 | 0.171 | 0.054 | 0.168 |
| | Promotion | 34.6% | 67.2% | 21.0% | 11.2% | 28.0% | 17.2% |
| Traffic | Uni-modal | 0.210 | 0.339 | 0.245 | 0.417 | 0.175 | 0.311 |
| | MM-TSFLib | 0.189 | 0.341 | 0.179 | 0.335 | 0.169 | 0.288 |
| | **TaTS (ours)** | 0.131 | 0.248 | 0.134 | 0.297 | 0.137 | 0.242 |
| | Promotion | 30.7% | 26.8% | 25.1% | 11.3% | 18.9% | 16.0% |

**TaTS Benefits from Better Alignment (i.e., lower TT-Wasserstein).** We compute the average forecasting improvement of TaTS compared to numerical-only modeling, as well as the ratio of TT-Wasserstein between original and shuffled Time-MMD datasets. The results, shown in Table 5, reveal that, within the same sampling frequency, lower TT-Wasserstein original-shuffled ratios tend to correlate with larger performance gains from TaTS (except for Climate). In other words, for paired time series and texts that have stronger CTR, i.e., more relevant, our TaTS could improve performance more. Thus, TT-Wasserstein can indicate the usefulness of paired texts and the potential effectiveness of TaTS when text quality and cross-modal alignment are uncertain.

**Comparison beyond Uni-modal Modeling and MM-TSFLib.** Besides numerical-only time series models and the MM-TSFLib framework, we compare our TaTS with several other baselines on three dataset sources, including Time-MMD (Liu et al., 2024a), FNSPID (Dong et al., 2024), and

Table 5: TaTS Improvements positively correlated with TT-Wasserstein measure of Time-MMD (Liu et al., 2024a) datasets. "Original-Shuffled TT Ratio" means the ratio of TT-Wasserstein on the original dataset to TT-Wasserstein on the shuffled dataset. "TaTS Improvement" means the average improvement of TaTS over uni-modal time series modeling.

| Dataset | | Monthly Sampled | | | | Weekly Sampled | | Daily Sampled |
|---|---|---|---|---|---|---|---|---|
| | Agriculture | Climate | Economy | Security | Social Good | Traffic | Energy | Health | Environment |
| Original-Shuffled TT Ratio (%) | 26.8 | 72.4 | 22.3 | 91.5 | 38.2 | 34.6 | 97.1 | 85.6 | 83.6 |
| TaTS Improvement (%) | 20.70 | 18.03 | 64.80 | 4.05 | 10.99 | 16.78 | 3.62 | 19.51 | 36.00 |

FNF (Wang et al., 2024b). Although N-BEATS, N-HiTS and TCN are not specifically designed for time series with paired texts, we adapt them by feeding both time series and text embeddings to them. We also include several baselines that explicitly incorporate multimodal information.. For ChatTime, we concatenate all paired texts into a single long text and perform zero-shot inference. For GPT4MTS, we convert the datasets we used into their prescribed input format and directly apply their pipeline. The results averaged from multiple prediction lengths are shown in Table 4, with full results in Table 28. The results show that covariate-based and convolution-based models perform comparably or worse than iTransformer, and consistently underperform compared to our TaTS. Although ChatTime achieves competitive results even in a zero-shot setting, highlighting the value of integrating texts when high-quality texts are available, our TaTS outperforms it through supervised learning. We further compare TaTS with several multimodal methods from concurrent preprints that also evaluate on Time-MMD. Using their reported results (Table 12), we find that TaTS elevates standard time-series backbones into highly competitive models, rivaling recent approaches with substantially more complex designs.

## 5.2 FURTHER ANALYSIS

**Hyperparameter Sensitivity.** We perform hyperparameter studies to evaluate the impact of (i) the learning rate and (ii) $d_{\text{mapped}}$, the dimension to which high-dimensional text embeddings are projected by the MLP, as defined in Equation 5. The results are presented in Figure 4, subfigures (a) and (b), with full results available in Appendix E.5 and Appendix E.6. The findings indicate that TaTS maintains robust performance across different choices of the learning rate and the text projection dimension $d_{\text{mapped}}$.

**Ablation with Different Text Encoders in TaTS.** While GPT-2 was used as the primary text encoder in our main experiments to demonstrate the effectiveness of TaTS, we further evaluate the performance of TaTS with different text encoders, including BERT (Devlin et al., 2019), GPT-2 (Radford et al., 2019), and LLaMA2 (Touvron et al., 2023). We utilize the official implementations available on Hugging Face and present the results in Figure 4, with full results provided in Appendix E.7. The results show that TaTS remains robust across different text encoders and consistently outperforms both the uni-modal and MM-TSFLib baselines. Notably, as the size of the language models used in TaTS increases from 110M (BERT) to 1.5B (GPT-2) and further to 7B (LLaMA2), we observe a slight improvement in performance. Investigating the relationship between the text encoder size and TaTS's effectiveness remains an open direction for future research.

**Ablation with Other Design Choices to Combine Modalities.** In our main experiments, TaTS transforms the paired texts into auxiliary variables through embedding, projection and concatenation. Our TaTS framework is readily extensible to other fusion design choices. We evaluate two alternative architectures that replace the MLP projection with:

Table 6: TaTS Ablation on Alternative Modality Fusion Architectures.

| Dataset | Climate | | Security | | Traffic | | # Parameters | |
|---|---|---|---|---|---|---|---|---|
| TaTS Settings | MSE | MAE | MSE | MAE | MSE | MAE | Fuser | Total |
| MLP projection (original) | 0.992 | 0.791 | 109.865 | 4.968 | 0.178 | 0.224 | 74988 | 6396658 |
| gated residual | 0.998 | 0.794 | 110.254 | 4.965 | 0.177 | 0.227 | 9396 | 6331066 |
| cross-attention | 0.989 | 0.792 | 109.729 | 4.976 | 0.178 | 0.221 | 99340 | 6421010 |

(a) a gated residual and (b) a cross-attention module between the time series and text embeddings. We report the average MSE and MAE across TaTS with iTransformer, PatchTST, and FiLM in Table 6. From the results, these alternative mechanisms achieve similar performance to the MLP-based design. One possible explanation is that linear projections are already highly competitive for time-series representation learning, especially when paired with strong backbone forecasters that account for the majority of the model parameters. We leave the exploration of more fine-grained multimodal fusion designs as an interesting direction for future work.

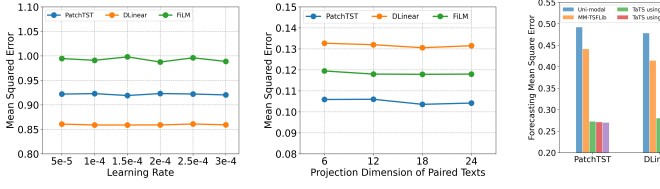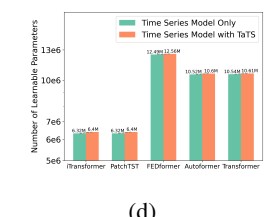

| (a) | (b) | (c) | (d) |

Figure 4: Further analysis of our TaTS framework. (a) Learning rate sensitivity: TaTS maintains stable performance across different learning rates (full results in Appendix E.5). (b) Text Projection Dimension sensitivity: TaTS remains robust across varying $d_{\mathrm{mapped}}$ (full results in Appendix E.6). (c) Varying text encoder: TaTS consistently outperforms baselines across different text encoders (full results in Appendix E.7). (d) Efficiency: TaTS introduces only a minor parameter increase ($\sim 1\%$) but significantly improves the performance according to Table 2.

**Ablation on Corrupted Datasets.** To validate that the performance gains stem from useful textual cues, we conduct corruption experiments where the textual information was randomly shuffled across timestamps. All other settings are unchanged, and the results are in Table 7. When textual alignment is destroyed, the performance drops significantly to matching or even being worse than the uni-modal baseline. This is expected, as randomly shuffled text acts as noise rather than a meaningful signal. Yet our TaTS stays relatively robust because it can learn to assign a small weight to textual information during optimization as a mitigation to noisy texts.

Table 7: TaTS Ablation on text-shuffled datasets.

| Dataset | Climate | | Security | | Traffic | |
|---|---|---|---|---|---|---|
| Settings | MSE | MAE | MSE | MAE | MSE | MAE |
| iTransformer + TaTS + original data | 1.028 | 0.804 | 112.05 | 5.151 | 0.187 | 0.217 |
| iTransformer + TaTS + corrupted data | 1.242 | 0.895 | 117.82 | 5.767 | 0.223 | 0.265 |
| iTransformer + uni-modal + original data | 1.183 | 0.871 | 115.94 | 5.660 | 0.213 | 0.238 |

**Ablation on Text-Randomly-Dropped Datasets.** Similar to corruption ablations, we conduct experiments with randomly dropped texts. Dropped texts are filled with "no information available." The results are in Table 8. While performance degrades slightly as more text is randomly dropped, TaTS remains effective, performing comparably to MM-TSF Lib even with 25% of the text missing. This experiment shows that whenever textual information is available, our TaTS could benefit from it to improve time series modeling.

Table 8: TaTS Ablation on text-dropped datasets.

| Dataset | Climate | | Security | | Traffic | |
|---|---|---|---|---|---|---|
| Settings | MSE | MAE | MSE | MAE | MSE | MAE |
| TaTS + original data | 1.028 | 0.804 | 112.05 | 5.151 | 0.187 | 0.217 |
| TaTS + 10% text randomly dropped | 1.032 | 0.809 | 112.94 | 5.372 | 0.194 | 0.239 |
| TaTS + 25% text randomly dropped | 1.056 | 0.818 | 114.59 | 5.231 | 0.203 | 0.254 |

**Mitigating Negative Effects of Extremely Noisy Texts.** Our ablation on corrupted datasets shows that highly noisy text can cause TaTS to even slightly underperform numerical-only modeling. To address this, we evaluate a simple mitigation strategy: randomly dropping a portion of the noisy texts and replacing them with "no information available." We test drop rates of 40% and 80%. As shown in Table 9, dropping noisy texts allows TaTS to recover performance close to the unimodal baseline, demonstrating the effectiveness of this strategy in handling unreliable text. Practitioners can refer to Appendix B.2 for how TaTS can adapt to scenarios with low-quality texts.

Table 9: Dropping extremely noisy texts mitigates negative effects from them.

| Dataset | Climate | | Security | | Traffic | |
|---|---|---|---|---|---|---|
| Settings | MSE | MAE | MSE | MAE | MSE | MAE |
| TaTS + corrupted data | 1.242 | 0.895 | 117.82 | 5.767 | 0.223 | 0.265 |
| TaTS + 60% corrupted data | 1.218 | 0.879 | 117.43 | 5.730 | 0.215 | 0.260 |
| TaTS + 20% corrupted data | 1.201 | 0.878 | 116.68 | 5.713 | 0.216 | 0.255 |
| uni-modal + original data | 1.183 | 0.871 | 115.94 | 5.660 | 0.213 | 0.238 |

**Computational Overhead vs. Performance Gain.** We also evaluate the efficiency of our proposed TaTS by measuring the training time per epoch and the total number of model parameters. Figure 4 (d) presents the total number of parameters for the best-performing models in our forecasting experiments. TaTS introduces only a lightweight three-layer MLP to project high-dimensional text embeddings into a lower-dimensional space, adding a minimal number of parameters compared to the original time series models. As a result, the overall parameter count increases by only about 1%.

Due to the inclusion of augmented time series with auxiliary variables from paired texts, the training time per epoch increases slightly, as shown in Figure 5, with average performance of each framework indicated by cross markers. Full results for all datasets are provided in Appendix E.8. Notably, this marginal efficiency trade-off ($\sim 1\%$ in terms of number of learnable parameters and $\sim 8\%$ in terms of training time) leads to significant improvements ($\sim 14\%$) in forecasting performance.

## 6 RELATED WORK

**Numerical-only Time Series Modeling.** Recently, various deep learning models have been developed for time series analysis, which can be broadly categorized into three categories. (1) **Patch-based models.** PatchTST (Nie et al., 2023) segments time series into subseries-level patches to capture dependencies, while Crossformer (Zhang & Yan, 2023) employs a two-stage attention mechanism to model both cross-time and cross-variable dependencies efficiently. Autoformer (Wu et al., 2021) introduces decomposition blocks to separate seasonal and trend-cyclical components. (2) **Global representation models.** iTransformer (Liu et al., 2024c) utilizes attention over global series representations to capture multivariate correlations. Informer (Zhou et al., 2021) reduces self-attention complexity using ProbSparse self-attention for improved efficiency. Dlinear (Zeng et al., 2023) demonstrates that simple linear regression in the raw space can perform competitively on MTS tasks. (3) **Frequency-aware models.** FEDformer (Zhou et al., 2022b)

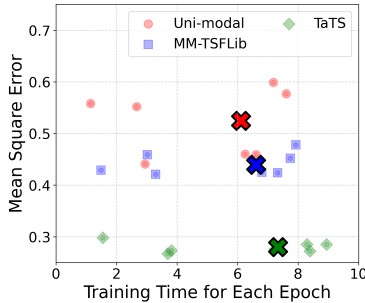

Figure 5: While TaTS incurs a slight increase in training time due to augmented auxiliary variables, it significantly improves forecasting. Full results in Appendix E.8.

represents series through randomly selected Fourier components, while FiLM (Zhou et al., 2022a) enhances representations with frequency-based layers to reduce noise and accelerate training. In this work, our proposed TaTS is compatible with all of the models listed above.

**Time Series with other data sources.** In the financial domain, several early works have explored integrating time series with textual data, albeit not in a timestamp-aligned manner, or often leveraging general machine learning models rather than time series-specific architectures. For example, StockNet (Xu & Cohen, 2018) uses a VAE-like model for chaotic stock-text data, while (Rodrigues et al., 2019) fuses time series with a single event-related document. BoEC (Farimani et al., 2021) applies a bag-of-economic-concepts approach, and Dandelion (Zhou et al., 2020) leverages multimodal attention for feature aggregation from multiple text sources. Some works also explored time series with vision information (Liu & Cai, 2012; Gerard et al., 2023; Lütjens et al., 2024) for spatio-temporal analysis. Recently, Time-MMD (Liu et al., 2024a) constructs a dataset of time series paired with parallel text, covering multiple domains, which we used in our main experiments.

**Large Language Models on Time Series.** The rapid advancement of Large Language Models (LLMs) has inspired a new line of research that transforms time series into natural language representations, enabling LLMs to perform downstream tasks (Zhang et al., 2024c). While these approaches demonstrate strong generalization capabilities due to the power of LLMs (Gruver et al., 2023; Cao et al., 2024; Jin et al., 2024; Xue & Salim, 2024), they also inherit limitations such as hallucination (Huang et al., 2023) and context length constraints (Wang et al., 2024a; Liu et al., 2024b). Notably, a recent work (Tan et al., 2024) suggests that replacing complex LLM architectures with basic attention layers does not necessarily degrade the performance. Another emerging line of work focuses on generating texts from time series, with applications in temporal reasoning and time-series question answering (Chang et al., 2025). MTBench (Chen et al., 2025) provides a comprehensive evaluation framework for assessing whether LLMs can jointly reason over structured numerical trends and unstructured textual narratives. TSAIA (Ye et al., 2025) benchmarks LLMs' multi-step temporal reasoning capabilities. TimeXL (Jiang et al., 2025) further introduces collaborating LLM agents to generate interpretable natural-language explanations alongside time-series forecasts. One notable future direction is to enable latent communications (Zou et al., 2025a;b) between the time series model and language model to build general time series intelligence.

## 7 CONCLUSION

Real-world time series data often comes with textual descriptions, yet prior studies have largely overlooked this modality. We identify *Chronological Textual Resonance*, where text embeddings exhibit periodic patterns similar to their paired time series. To leverage this insight, we propose a plug-and-play framework that transforms text representations into auxiliary variables, seamlessly integrating them into existing time series models. Extensive experiments across various forecasting models and real-world datasets demonstrate the state-of-the-art performance of our approach.

## ACKNOWLEDGEMENT

This work is supported by NSF 2433308, IBM-Illinois Discovery Accelerator Institute, and MIT-IBM Watson AI Lab. The content of the information in this document does not necessarily reflect the position or the policy of the Government, and no official endorsement should be inferred. The U.S. Government is authorized to reproduce and distribute reprints for Government purposes notwithstanding any copyright notation here on.

## STATEMENTS

### ETHICS STATEMENT

Our work is solely focused on the technical challenge of multimodal time series and does not involve any elements that could pose ethical risks.

### REPRODUCIBILITY STATEMENT

We provide our full experiment code in the supplementary materials, accompanied by a detailed README to facilitate reproduction. The experimental environment is described in Appendix D, including hardware and software configurations. All benchmark datasets used in this work are publicly available. Comprehensive experiment details, including dataset specifications, hyperparameters, and optimizer settings, are also documented in Appendix D. Reported results are averaged across multiple prediction lengths. Furthermore, Section 5 presents hyperparameter studies, demonstrating the robustness of our proposed TaTS framework.

### LARGE LANGUAGE MODEL USAGE STATEMENT

Large Language Models did not play a significant role in research ideation and/or writing to the extent that they could be regarded as a contributor. They were used only for minor refinement of writing and as components within our proposed TaTS framework for handling the text modality.

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

# Appendix

**Roadmap.** In this appendix, we provide a detailed overview of our methodology and experimental setup. Appendix A outlines the complete process of frequency analysis for both time series and paired texts. Appendix B discusses of preprints that are related to this work. Appendix C illustrates the TaTS algorithms for time series forecasting and imputation. Appendix D includes details on datasets, hyperparameters, evaluation metrics, and additional implementation specifics. Due to space constraints in the main text, Appendix E presents the full experimental results, including comprehensive forecasting and imputation outcomes, hyperparameter and ablation studies, efficiency evaluations, and visualizations. The table of contents is provided below for reference.

## TABLE OF CONTENTS

# A    DETAILED FREQUENCY ANALYSIS PROCESS OF TIME SERIES WITH PAIRED TEXTS

Here, we provide a detailed explanation of the frequency analysis process for both the time series and their paired texts. In Proposition A.1 and A.2, we simplify the proof by analyzing each periodic component individually as a single cosine. This is without loss of generality, as the operations involved are linear. Therefore, the conclusions naturally extend to signals composed of multiple periodic components through linear superposition.

**Proposition A.1.** *The computation of lag similarity preserves the original periodicities of the data.*

*Proof.* Let $\mathbf{S} = \{s_t\}_{t=1}^{T}$ represent the paired texts or data sequence, where each $s_t$ corresponds to a time step $t$. Define the lag similarity at lag $k$ as:

$$\mathrm{LagSim}(k) = \frac{1}{T-k} \sum_{t=1}^{T-k} \mathrm{sim}(s_t, s_{t+k}), \tag{9}$$

where $\mathrm{sim}(\cdot, \cdot)$ is a similarity measure (e.g., cosine similarity).

Now, consider the periodic component of the data sequence $\mathbf{S}$, which can be represented as:

$$s_t = A \cos\left(\frac{2\pi t}{P} + \phi\right), \tag{10}$$

where $A$ is the amplitude, $P$ is the period, and $\phi$ is the phase.

For two points separated by lag $k$, the similarity $\mathrm{sim}(s_t, s_{t+k})$ depends on the relative difference between their phases:

$$s_{t+k} = A \cos\left(\frac{2\pi(t+k)}{P} + \phi\right) = A \cos\left(\frac{2\pi t}{P} + \frac{2\pi k}{P} + \phi\right). \tag{11}$$

The lag similarity is then computed as:

$$\mathrm{LagSim}(k) = \frac{1}{T-k} \sum_{t=1}^{T-k} \mathrm{sim}\left(A\cos\left(\frac{2\pi t}{P} + \phi\right), A\cos\left(\frac{2\pi t}{P} + \frac{2\pi k}{P} + \phi\right)\right). \tag{12}$$

Since the cosine function is periodic with period $P$, the similarity $\mathrm{sim}(s_t, s_{t+k})$ also inherits this periodicity. Therefore, the overall lag similarity $\mathrm{LagSim}(k)$ retains the periodicities of the original sequence $\mathbf{S}$.

Thus, the computation of lag similarity preserves the original periodicities of the data. $\square$

**Proposition A.2.** *The stabilization of a data sequence using first-order differentiation preserves its original periodicities.*

*Proof.* Let $\mathbf{S} = \{s_t\}_{t=1}^{T}$ represent a data sequence, where $s_t$ is the value at time step $t$. The first-order differentiation of the sequence is defined as:

$$\Delta s_t = s_{t+1} - s_t, \quad t = 1, 2, \ldots, T-1. \tag{13}$$

Suppose the sequence $\mathbf{S}$ exhibits periodic behavior with period $P$ and can be represented as:

$$s_t = A \cos\left(\frac{2\pi t}{P} + \phi\right), \tag{14}$$

where $A$ is the amplitude, $P$ is the period, and $\phi$ is the phase.

The first-order difference of $s_t$ is:

$$\Delta s_t = s_{t+1} - s_t = A \cos\left(\frac{2\pi(t+1)}{P} + \phi\right) - A \cos\left(\frac{2\pi t}{P} + \phi\right). \tag{15}$$

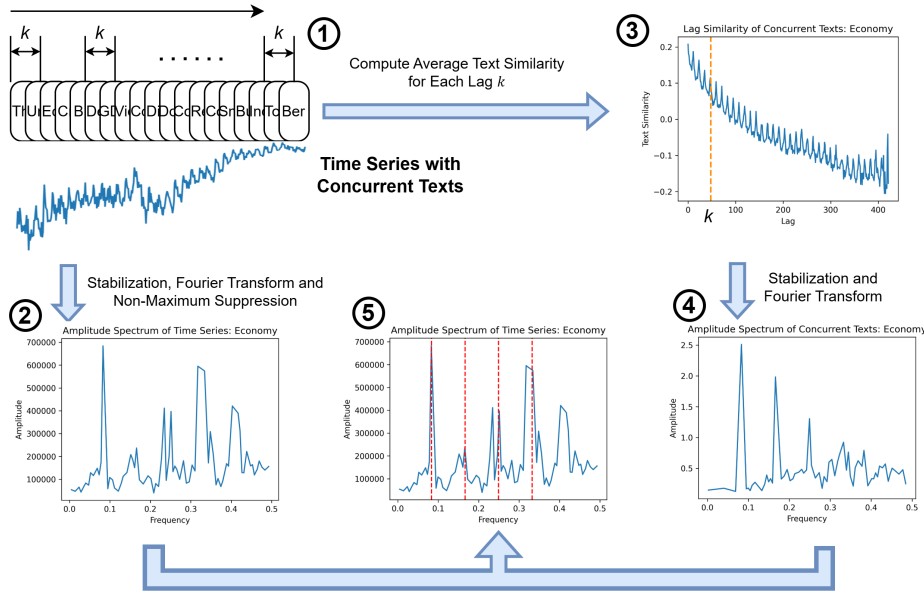

Figure 6: Illustration of the frequency analysis process for time series with paired texts in the Economy dataset. Step ①: Compute the average text similarity for each lag $k$. Step ②: Stabilize the time series using first-order differentiation, apply Fourier Transform, and perform Non-Maximum Suppression (NMS) to obtain the amplitude spectrum. Step ③: Visualize the lag similarity of paired texts. Step ④: Stabilize the paired texts, compute the Fourier Transform, and visualize the amplitude spectrum. Step ⑤: Overlay the top-$l$ (here $l$=4) frequencies of paired texts onto the time series amplitude spectrum to highlight shared periodic patterns. All the data transformation operations in this process are periodicity-preserving according to Proposition A.1 and Proposition A.2.

Using the trigonometric identity for the difference of cosines:

$$\cos(x+y) - \cos(x) = -2\sin\left(\frac{y}{2}\right)\sin\left(x + \frac{y}{2}\right), \tag{16}$$

we set $x = \frac{2\pi t}{P} + \phi$ and $y = \frac{2\pi}{P}$, giving:

$$\Delta s_t = -2A\sin\left(\frac{\pi}{P}\right)\sin\left(\frac{2\pi t}{P} + \phi + \frac{\pi}{P}\right). \tag{17}$$

The first term, $\sin\left(\frac{\pi}{P}\right)$, is a constant dependent on the period $P$. The second term, $\sin\left(\frac{2\pi t}{P} + \phi + \frac{\pi}{P}\right)$, retains the periodicity of $P$, as it is a sinusoidal function with the same frequency as the original sequence.

Thus, the first-order difference $\Delta s_t$ preserves the periodicity $P$ of the original data sequence. $\square$

The overall process of frequency analysis for the time series $\mathbf{X} = \{\mathbf{x}_1, \mathbf{x}_2, \ldots, \mathbf{x}_N\} \in \mathbb{R}^{T \times N}$ and paired texts $\mathbf{S} = \{s_1, s_2, \ldots, s_T\}$ in the Economy dataset $\mathcal{D}_{\text{Economy}} = \{\mathbf{X}, \mathbf{S}\}$ is illustrated in Figure 6. In this process, starting from the original dataset (subfigure ①), the time series and paired texts are analyzed independently. **For the time series $\mathbf{X} = \{\mathbf{x}_1, \mathbf{x}_2, \ldots, \mathbf{x}_N\}$, we perform standard frequency analysis**, stabilizing the data through first-order differentiation.

$$\Delta\mathbf{X}_t = \mathbf{X}_{t+1} - \mathbf{X}_t, \quad \text{for } t = 1, 2, \ldots, T-1, \tag{18}$$

where $\Delta\mathbf{X}_t$ represents the first-order difference of the time series. This step removes long-term trends and ensures that the data is stationary, allowing for a more accurate analysis of its frequency components.

Then, we compute the Fourier Transform (Nussbaumer, 1982; Sneddon, 1995) of $\Delta \mathbf{X}$ to analyze its frequency components. The Fourier Transform of $\Delta \mathbf{X}$ is defined as:

$$\mathcal{F}_{\Delta \mathbf{X}}(f) = \sum_{t=1}^{T-1} \Delta \mathbf{X}_t e^{-i 2\pi f t}, \tag{19}$$

where $f$ represents the frequency, $\Delta \mathbf{X}_t$ is the first-order difference of the time series at time $t$, and $i$ is the imaginary unit. The resulting $\mathcal{F}_{\Delta \mathbf{X}}(f)$ provides the amplitude and phase information of each frequency component present in the time series.

The magnitude spectrum, which represents the amplitude of each frequency component, is computed as:

$$|\mathcal{F}_{\Delta \mathbf{X}}(f)| = \sqrt{\operatorname{Re}(\mathcal{F}_{\Delta \mathbf{X}}(f))^2 + \operatorname{Im}(\mathcal{F}_{\Delta \mathbf{X}}(f))^2}, \tag{20}$$

where $\operatorname{Re}(\mathcal{F}_{\Delta \mathbf{X}}(f))$ and $\operatorname{Im}(\mathcal{F}_{\Delta \mathbf{X}}(f))$ are the real and imaginary parts of $\mathcal{F}_{\Delta \mathbf{X}}(f)$, respectively.

By analyzing $|\mathcal{F}_{\Delta \mathbf{X}}(f)|$, we identify the dominant frequencies in the time series, which reveal its periodic patterns. To further highlight the dominant frequencies, we apply Non-Maximum Suppression (NMS) to the magnitude spectrum $|\mathcal{F}_{\Delta \mathbf{X}}(f)|$. NMS ensures that only the most prominent frequencies are retained while suppressing nearby less significant frequencies. The NMS operation is defined as follows:

$$\mathcal{N}(f) = \begin{cases} |\mathcal{F}_{\Delta \mathbf{X}}(f)|, & \text{if } |\mathcal{F}_{\Delta \mathbf{X}}(f)| > |\mathcal{F}_{\Delta \mathbf{X}}(f')| \, \forall f' \in \mathcal{N}(f), \\ 0, & \text{otherwise}, \end{cases} \tag{21}$$

where $\mathcal{N}(f)$ represents a local neighborhood around the frequency $f$. The operation compares the magnitude of $|\mathcal{F}_{\Delta \mathbf{X}}(f)|$ with those of neighboring frequencies and retains only the largest value within the neighborhood. Frequencies that do not satisfy the condition are set to zero.

After applying NMS, the remaining frequencies represent the dominant periodic components of the time series, making it easier to identify significant periodic patterns. This process eliminates noise and reduces the influence of minor frequency components, enhancing the interpretability of the spectrum. The final visualization of the amplitude spectrum of the time series is shown in Figure 6, subfigure ②.

For the paired texts $\mathbf{S} = \{s_1, s_2, \ldots, s_T\}$, we first embed each $s_t$ using the text encoder $\mathcal{H}_{\text{text}}$:

$$\mathbf{e}_t = \mathcal{H}_{\text{text}}(s_t; \theta_{\text{text}}) \in \mathbb{R}^{d_{\text{text}}}, \tag{22}$$

where $\theta_{\text{text}}$ represents the parameters of the text encoder, and $\mathbf{e}_t$ is the resulting text embedding at timestamp $t$.

Since the text embeddings are typically close in the embedding space, leading to similar cosine similarity values, we normalize the embeddings by centering them around their mean to improve numerical stability and enhance sensitivity to differences. Specifically, we compute the mean embedding:

$$\mathbf{e}_{\text{mean}} = \frac{1}{T} \sum_{t=1}^{T} \mathbf{e}_t, \tag{23}$$

and shift all embeddings by subtracting the mean:

$$\mathbf{e}'_t = \mathbf{e}_t - \mathbf{e}_{\text{mean}}, \tag{24}$$

where $\mathbf{e}'_t$ represents the centered (shifted) embeddings.

Next, we compute the average text similarity for each lag $k \in \{1, T-1\}$ as:

$$\operatorname{Sim}(k) = \frac{1}{T-k} \sum_{t=1}^{T-k} \operatorname{sim}(\mathbf{e}'_t, \mathbf{e}'_{t+k}), \tag{25}$$

where $\operatorname{sim}(\mathbf{e}'_t, \mathbf{e}'_{t+k})$ denotes the similarity measure (e.g., cosine similarity) between the centered embeddings at time $t$ and $t+k$, defined as:

$$\operatorname{sim}(\mathbf{e}'_t, \mathbf{e}'_{t+k}) = \frac{\mathbf{e}'_t \cdot \mathbf{e}'_{t+k}}{\|\mathbf{e}'_t\| \|\mathbf{e}'_{t+k}\|}. \tag{26}$$

We visualize the lag similarity of paired texts, $\mathrm{Sim}(k)$, in Figure 6, subfigure ③. Subsequently, we stabilize the data by applying first-order differentiation and perform a Fourier Transform, following a similar process as previously described for the time series frequency analysis. The final visualization of the amplitude spectrum of the paired texts is presented in Figure 6, subfigure ④.

Then, we compute the frequencies with the top-$l$ amplitudes from the lag similarity $\mathrm{Sim}(k)$. We apply the Fourier Transform to $\mathrm{Sim}(k)$:

$$\mathcal{F}_{\text{text}}(f) = \sum_{t=1}^{T-1} \mathrm{Sim}(k)\, e^{-i2\pi f t}, \tag{27}$$

where $f$ is the frequency, and $\mathcal{F}_{\text{text}}(f)$ represents the complex Fourier coefficients corresponding to each frequency $f$.

Next, we compute the amplitude spectrum as:

$$|\mathcal{F}_{\text{text}}(f)| = \sqrt{\mathrm{Re}(\mathcal{F}_{\text{text}}(f))^2 + \mathrm{Im}(\mathcal{F}_{\text{text}}(f))^2}, \tag{28}$$

where $\mathrm{Re}(\mathcal{F}_{\text{text}}(f))$ and $\mathrm{Im}(\mathcal{F}_{\text{text}}(f))$ are the real and imaginary parts of $\mathcal{F}_{\text{text}}(f)$, respectively.

We then identify the top-$l$ dominant frequencies by selecting the $l$ frequencies corresponding to the largest amplitudes:

$$\mathcal{F}_{\text{top}} = \{ f_i \mid |\mathcal{F}_{\text{text}}(f_i)| \text{ is among the top-}l \text{ largest amplitudes} \}. \tag{29}$$

These top-$l$ frequencies represent the most significant periodic components of the paired texts, revealing their dominant temporal patterns. we overlay the top-$l$ (in the Economy dataset $l$=4) frequencies of the paired texts onto the amplitude spectrum of the time series, as illustrated in Figure 6, subfigure ⑤.

We also visualize the frequency analysis process in the Social Good dataset and Traffic dataset respectively in Figure 7 and Figure 8.

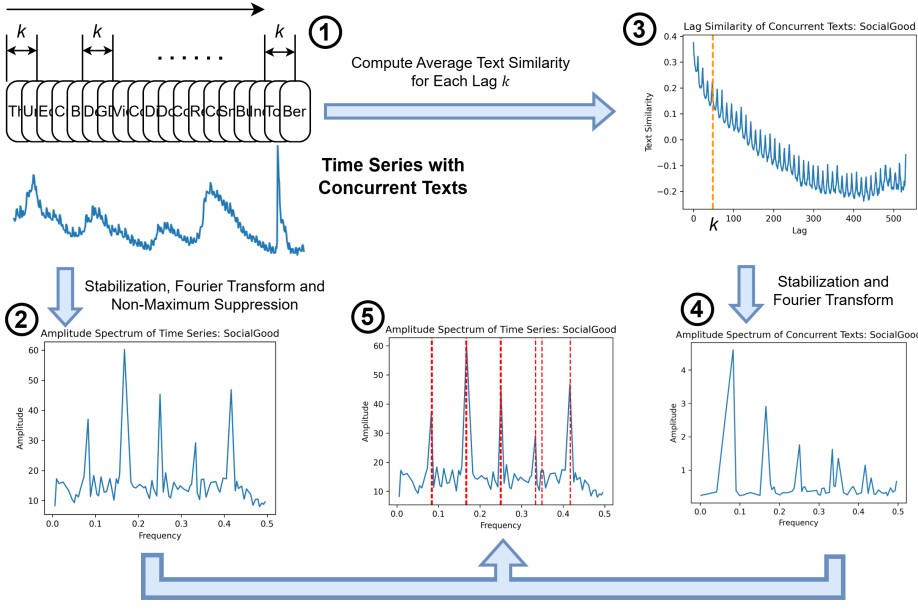

Figure 7: Illustration of the frequency analysis process for time series with paired texts in the Social Good dataset. In Step ⑤, we overlay the top-9 frequencies of paired texts onto the time series amplitude spectrum.

## A.1    ESTIMATION STABILITY OF TT-WASSERSTEIN

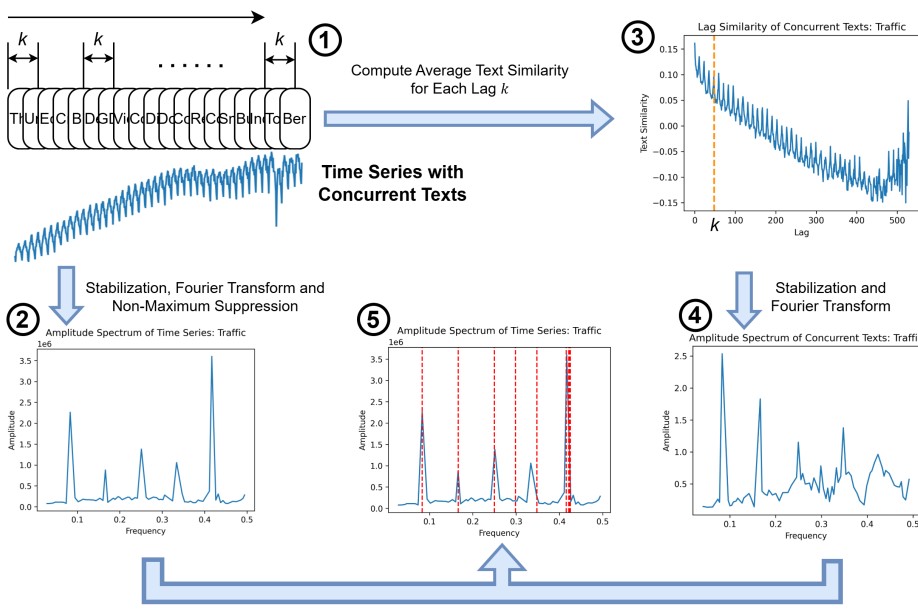

Figure 8: Illustration of the frequency analysis process for time series with paired texts in the Traffic dataset. In Step ⑤, we overlay the top-7 frequencies of paired texts onto the time series amplitude spectrum.

As TT-Wasserstein is an empirically estimated statistical metric, we provide a detailed sensitivity and robustness analysis in this appendix. Our default configuration uses a rectangular window, $\ell_1$-normalization of spectral amplitudes, and a frequency grid determined by the full length of the time series (native FFT resolution). To assess the stability of TT-Wasserstein, we perform ablations that vary each of these components: windowing schemes, normalization strategies, and frequency resolutions, while keeping all other settings fixed. The results are summarized in Table 10, where each row reflects the effect of a single deviation from the default configuration. From the results, our TT-Wasserstein is stable across various configuration choices. We also implemented block bootstrap with a bootstrap sample size of 10 for time series and texts to compute the confidence intervals for the default settings in Table 11.

Table 10: TT-Wasserstein measure of Time-MMD (Liu et al., 2024a) datasets. Our TT-Wasserstein is generally stable on different frequency computation configurations.

| Dataset | Monthly Sampled | | | | | | Weekly Sampled | | Daily Sampled |
| --- | --- | --- | --- | --- | --- | --- | --- | --- | --- |
| | Agriculture | Climate | Economy | Security | Social Good | Traffic | Energy | Health | Environment |
| Default | 0.026 | 0.025 | 0.022 | 0.049 | 0.027 | 0.035 | 0.307 | 0.233 | 0.302 |
| Hann Window | 0.031 | 0.029 | 0.016 | 0.041 | 0.040 | 0.057 | 0.304 | 0.243 | 0.306 |
| Hamming Window | 0.054 | 0.025 | 0.016 | 0.041 | 0.037 | 0.046 | 0.305 | 0.242 | 0.305 |
| Blackman Window | 0.061 | 0.029 | 0.020 | 0.036 | 0.043 | 0.055 | 0.297 | 0.245 | 0.298 |
| Minmax Normalization | 0.032 | 0.035 | 0.025 | 0.043 | 0.031 | 0.049 | 0.345 | 0.291 | 0.332 |
| Log Normalization | 0.016 | 0.060 | 0.046 | 0.049 | 0.050 | 0.048 | 0.267 | 0.170 | 0.286 |
| Zero-padded frequency resolution ($2\times$) | 0.046 | 0.058 | 0.052 | 0.051 | 0.028 | 0.050 | 0.280 | 0.187 | 0.292 |
| Down-sampled frequency resolution ($\frac{1}{2}\times$) | 0.011 | 0.026 | 0.039 | 0.050 | 0.051 | 0.025 | 0.259 | 0.197 | 0.298 |

Table 11: Standard deviation of default TT-Wasserstein measure of Time-MMD (Liu et al., 2024a) datasets by block bootstrap.

| Dataset | Monthly Sampled | | | | | | Weekly Sampled | | Daily Sampled |
| --- | --- | --- | --- | --- | --- | --- | --- | --- | --- |
| | Agriculture | Climate | Economy | Security | Social Good | Traffic | Energy | Health | Environment |
| Default | ±0.06 | ±0.04 | ±0.03 | ±0.03 | ±0.04 | ±0.06 | ±0.21 | ±0.16 | ±0.12 |

## B FURTHER DISCUSSION

### B.1 CONCURRENT WORKS

We discuss concurrent preprints available online that relate to this research, providing a more comprehensive understanding of existing works and ongoing efforts to integrate texts into time series modeling.

(Williams et al., 2024) introduces a benchmark (CiK) for time-series forecasting that integrates both numerical data and textual context. While both (Williams et al., 2024) and our work emphasize the integration of textual context into time series forecasting, the textual contexts in (Williams et al., 2024) are descriptions regarding the time series itself, but in our work, we do not have such a constraint.

(Kim et al., 2024) develops a hybrid forecaster that jointly predicts both time series and textual data by projecting time series to the language space and fine-tuning the pre-trained LLM. Their assumption is that though LLMs are originally for language but not for time series, they can be fine-tuned to adapt to time series.

(Zhang et al., 2024a) introduces a dual-adapter model for time series and textual data, but their data format is similar to (Wang et al., 2025), where the whole time series is paired with one text.

(Xie et al., 2024) proposes a novel multimodal LLM framework (TS-MLLM) that treats time series as a modality akin to images. But they are using synthetic generation and focus on NLP tasks rather than time series forecasting.

**Comparison with Concurrent Multimodal Time Series Forecasting Methods.** Several concurrent works also explore incorporating multimodal information to enhance time series forecasting and report results on the Time-MMD benchmark (Liu et al., 2024a). TFHTS (Zhou et al., 2025) adopts a dual-tower architecture to fuse time series and textual signals, followed by large language models for forecasting. MCD-TSF (Su et al., 2025a) proposes a multimodal-conditioned diffusion model that adaptively aligns textual context with temporal dynamics. TeR-TSF (Su et al., 2025b) introduces text reinforcement, generating augmented textual descriptions to improve downstream prediction.

For a fair comparison, we follow their evaluation protocol and exclude datasets that exhibit substantial performance gaps, suggesting that their preprocessing pipelines may differ from ours. we report averaged performance over prediction lengths 6, 12, and 18 for monthly-frequency datasets (Agriculture, Climate, Social Good); 12, 24, and 36 for weekly-frequency datasets (Energy, Health); and 48, 96, and 192 for the daily-frequency Environment dataset. Note that, unlike our main experiments, we use a historical window of 36 steps for monthly, 96 for weekly, and 192 for daily datasets. The results are summarized in Table 12. From the results, TaTS achieves the best or second-best performance on 9 out of the 12 metrics. To provide an overall comparison, we compute an average ranking as follows: for each dataset, we assign each method a rank from 1 to 4 for both MSE and MAE (with lower values indicating better performance), average the two ranks to obtain a per-dataset score, and then average these scores across all datasets. Under this ranking scheme, TaTS obtains the smallest average ranking, indicating the best overall performance among concurrent multimodal time series forecasting methods.

### B.2 LIMITATIONS

This work focuses on revealing and quantifying Chronological Textual Resonance (CTR) and designing the TaTS framework to leverage it. However, several limitations remain. First, we do not thoroughly investigate the data construction processes that may induce CTR, such as biases introduced during text collection or the choice of contextual information. Understanding how these factors affect CTR would provide deeper insights into the robustness and generalizability of our approach. Second, we do not analyze how text embeddings are utilized at the neural level within the TaTS framework. A more detailed study on how the model learns temporal patterns from paired texts could inform improvements in architecture design. Third, the effectiveness of TaTS is influenced by the quality and relevance of paired texts. While we study cases where texts are randomly shuffled, real-world texts could have different patterns, and the model's performance may degrade. Future work could explore methods to assess and enhance text quality or develop more robust models to handle noisy input. Lastly, while TaTS demonstrates strong empirical performance across benchmark datasets,

Table 12: TaTS compared with several concurrent works that have been evaluated on Time-MMD. We use the reported performance in the concurrent works for comparison. Best results are bolded and second-best results are underlined. Full results in Table 28.

| Methods | | TaTS (Ours) + iTransformer | | TFHTS (2025) | | MCD-TSF (2025a) | | TeR-TSF (2025b) | |
|---|---|---|---|---|---|---|---|---|---|
| Datasets | | MSE | MAE | MSE | MAE | MSE | MAE | MSE | MAE |
| Time-MMD: Multimodal Time Series (2024a) | Agriculture | **0.201** | **0.321** | 0.571 | 0.564 | 0.222 | 0.322 | 0.338 | 0.402 |
| | Climate | **1.227** | 0.890 | 1.782 | 0.948 | 1.583 | 0.971 | 1.348 | **0.874** |
| | Energy | 0.214 | 0.321 | 0.290 | 0.403 | **0.153** | **0.293** | 0.202 | 0.324 |
| | Environment | 0.267 | 0.374 | 0.262 | 0.368 | 0.275 | 0.379 | **0.251** | **0.346** |
| | Health | **1.372** | 0.763 | 1.514 | 0.809 | 1.496 | 0.811 | 1.384 | **0.756** |
| | Social Good | 1.146 | **0.558** | 1.352 | 0.677 | **1.035** | 0.569 | 1.199 | 0.587 |
| Average Ranking | | 1.83 | | 3.5 | | 2.58 | | 2.08 | |

its generalization to other types of multimodal time series or domains with fundamentally different patterns remains untested, for example, time series with paired images or audio. We leave these investigations and improvements for future work.

Regarding our third limitation on text quality, we briefly clarify how TaTS may adapt under imperfect textual inputs. When texts are missing at some timestamps, TaTS can simply use a placeholder token (e.g., "no information available"), as validated in our text-random-drop ablations. When temporal alignment is uncertain, small lead–lag shifts mainly affect phase rather than frequency, so TaTS can still leverage the preserved periodic structure. For larger misalignments, standard preprocessing alignment methods may be applied, with TT-Wasserstein serving as a diagnostic to guide how strongly to rely on the text modality.

### B.3 ETHICAL IMPACT STATEMENT

Our work is solely focused on the technical challenge of multimodal time series and does not involve any elements that could pose ethical risks.

# C  ALGORITHMS

## C.1  PSEUDO CODE

---

**Algorithm 1:** Texts as Time Series for Forecasting Task

---

**Input:** Time series with concurrent texts embeddings
$\mathcal{D} = \{\mathbf{X} = \{\mathbf{x}_1, \mathbf{x}_2, \ldots, \mathbf{x}_N\}, \mathbf{E} = \{e_1, e_2, \ldots, e_T\}\}$ in the input training dataset;
prediction length $H$.

**Output:** TaTS model parameters $\Theta = \{\theta_{\text{forecast}}, \theta_{\text{MLP}}\}$.

1 Prepare training samples of sequence length $L$ and prediction length $H$ as:

$$\{\mathbf{X}^{(i)} = \mathbf{X}_{l_i+1:l_i+L}, \mathbf{E}^{(i)} = \mathbf{X}_{l_i+1:l_i+L}, \mathbf{Y}^{(i)} = \mathbf{X}_{l_i+L+1:l_i+L+H}\}_{i=1}^{n}$$

Initialize time series model $\mathcal{F}(\cdot; \theta_{\text{forecast}})$; projector $\text{MLP}(\cdot; \theta_{\text{MLP}})$ for dimensionality reduction.

2 **while** *not converged* **do**

3     **for** *each training sample* $\{\mathbf{X}^{(i)}, \mathbf{E}^{(i)}, \mathbf{Y}^{(i)}\}$ **do**

4        Map $\mathbf{E}^{(i)}$ to $\mathbf{Z}^{(i)}$: $\mathbf{Z}^{(i)}[j] = \text{MLP}(\mathbf{E}^{(i)}[j]; \theta_{\text{MLP}})$

5        Compute $\mathbf{U} = [\mathbf{X}^{(i)}; (\mathbf{Z}^{(i)})^{\mathsf{T}}]_{\text{dim}=1}$ as shown in Eq. (6).

6        Forecast: $\widehat{\mathbf{X}}^{(i)} = \mathcal{F}(\mathbf{U}; \theta_{\text{forecast}})[: N]$

7        Optimize: $\arg\min_{\Theta=\{\theta_{\text{forecast}}, \theta_{\text{MLP}}\}} \mathcal{L}_{\text{forecast}}(\mathbf{X}^{(i)}, \widehat{\mathbf{X}}^{(i)})$ as shown in Eq. (8)

8 **return** TaTS model parameters $\Theta = \{\theta_{\text{forecast}}, \theta_{\text{MLP}}\}$

---

---

**Algorithm 2:** Texts as Time Series for Imputation Task

---

**Input:** Time series with concurrent text embeddings
$\mathcal{D} = \{\mathbf{X} = \{\mathbf{x}_1, \mathbf{x}_2, \ldots, \mathbf{x}_N\}, \mathbf{E} = \{e_1, e_2, \ldots, e_T\}\}$ in the input training dataset;
binary mask $\mathbf{M} \in \{0,1\}^{T \times N}$.

**Output:** TaTS model parameters $\Theta = \{\theta_{\text{impute}}, \theta_{\text{MLP}}\}$.

1 Prepare training samples with observed entries:

$$\{\mathbf{X}^{(i)} = \mathbf{X}_{l_i+1:l_i+L}, \mathbf{E}^{(i)} = \mathbf{E}_{l_i+1:l_i+L}, \mathbf{M}^{(i)} = \mathbf{M}_{l_i+1:l_i+L}\}_{i=1}^{n}$$

Initialize time series imputation model $\mathcal{G}(\cdot; \theta_{\text{impute}})$; projector $\text{MLP}(\cdot; \theta_{\text{MLP}})$ for dimensionality reduction.

2 **while** *not converged* **do**

3     **for** *each training sample* $\{\mathbf{X}^{(i)}, \mathbf{E}^{(i)}, \mathbf{M}^{(i)}\}$ **do**

4        Map $\mathbf{E}^{(i)}$ to $\mathbf{Z}^{(i)}$: $\mathbf{Z}^{(i)}[j] = \text{MLP}(\mathbf{E}^{(i)}[j]; \theta_{\text{MLP}})$

5        Construct the augmented input: $\mathbf{U} = [(\mathbf{X}^{(i)} \odot \mathbf{M}^{(i)}); (\mathbf{Z}^{(i)})^{\mathsf{T}}]_{\text{dim}=1}$

6        Impute missing values: $\widehat{\mathbf{X}}^{\text{Imputed}(i)} = \mathcal{G}(\mathbf{U}; \theta_{\text{impute}})$

7        Optimize: $\arg\min_{\Theta=\{\theta_{\text{impute}}, \theta_{\text{MLP}}\}} \mathcal{L}_{\text{impute}}(\mathbf{X}^{(i)}, \widehat{\mathbf{X}}^{\text{Imputed}(i)})$

8 **return** TaTS model parameters $\Theta = \{\theta_{\text{impute}}, \theta_{\text{MLP}}\}$

---

## C.2  INCREMENTAL COMPLEXITY INTRODUCED BY THE TEXT MODALITY

As analyzed in Section 5.2, incorporating textual information introduces some additional computational overhead. This overhead arises from two components: (1) *the cost of encoding the text tokens*, and (2) *the cost of training the forecasting model on the augmented input*. We have empirically evaluated the latter in Section 5.2; here, we provide a theoretical analysis of the former.

Let the language encoder process each token in time $O(t_{\text{token}})$, and let each timestamp have an associated text description with an average length of $l$ tokens. If the time series contains $T$ timestamps, then the total time complexity for embedding all paired texts is:

$$O(t_{\text{token}} \cdot l \cdot T).$$

This follows because each of the $T$ timestamps requires encoding, on average, $l$ tokens, and each token requires $O(t_{\text{token}})$ time to process. In practice, from our experiments reported in Table 13, this cost is modest for lightweight text encoders (e.g., small Transformers or MLP-based tokenizers), and the embedding step is performed only once per dataset rather than per training iteration. As a result, the added modality introduces only a manageable overhead relative to the overall cost of training modern time-series forecasting models.

Table 13: Wall-clock costs (seconds) to embed texts for Time-MMD (Liu et al., 2024a) datasets.

| | | | Monthly Sampled | | | | Weekly Sampled | | Daily Sampled |
| | Agriculture | Climate | Economy | Security | Social Good | Traffic | Energy | Health | Environment |
|---|---|---|---|---|---|---|---|---|---|
| BERT | 0.12 | 0.21 | 0.23 | 0.23 | 1.7 | 0.21 | 0.26 | 0.46 | 0.29 |
| GPT-2 | 0.18 | 0.25 | 0.31 | 0.28 | 2.5 | 0.30 | 0.28 | 0.55 | 0.35 |
| LLaMA-2-7B | 0.28 | 0.29 | 0.37 | 0.35 | 10.5 | 0.40 | 0.32 | 0.62 | 0.42 |

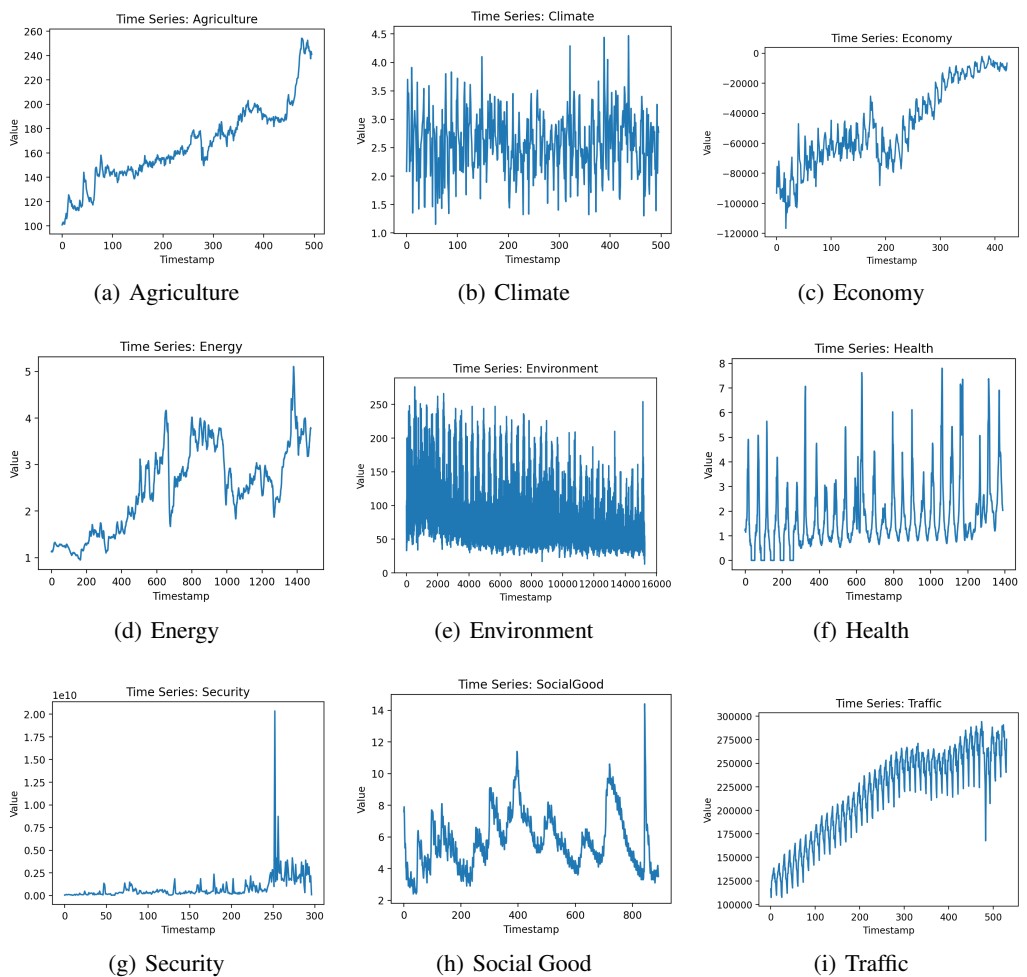

Figure 9: Visualization of the numerical data in each multimodal time series dataset.

# D    EXPERIMENT DETAILS

## D.1    DATASET STATISTICS AND DETAILS

Table 14, Table 15 and Table 16 provide summaries of the statistics for the publicly available real-world datasets. Additionally, we visualize the numerical data for each Time-MMD multimodal time series dataset in Figure 9. For further details, please refer to the original work that introduced these datasets and benchmarks (Liu et al., 2024a).

Table 14: Overview of the numerical data in the Time-MMD datasets (Liu et al., 2024a). "Prediction Length" refers to the number of future time points to be forecasted, with each dataset including four distinct prediction settings. refers to the number of variables (or variates) in each dataset.

| Dataset Name/Domain | Prediction Length | Dimension | Frequency | Number of Samples | Timespan |
|---|---|---|---|---|---|
| Agriculture | {6, 8, 10, 12} | 1 | Monthly | 496 | 1983 - Present |
| Climate | {6, 8, 10, 12} | 5 | Monthly | 496 | 1983 - Present |
| Economy | {6, 8, 10, 12} | 3 | Monthly | 423 | 1989 - Present |
| Energy | {12, 24, 36, 48} | 9 | Weekly | 1479 | 1996 - Present |
| Environment | {48, 96, 192, 336} | 4 | Daily | 11102 | 1982 - 2023 |
| Health | {12, 24, 36, 48} | 11 | Weekly | 1389 | 1997 - Present |
| Security | {6, 8, 10, 12} | 1 | Monthly | 297 | 1999 - Present |
| Social Good | {6, 8, 10, 12} | 1 | Monthly | 900 | 1950 - Present |
| Traffic | {6, 8, 10, 12} | 1 | Monthly | 531 | 1980 - Present |

Table 15: Overview of the numerical data in the FNSPID (Financial News and Stock Price Integration Dataset) datasets (Dong et al., 2024). "Prediction Length" refers to the number of future time points to be forecasted, with each dataset including four distinct prediction settings. refers to the number of variables (or variates) in each dataset.

| Company/Stock Name | Prediction Length | Dimension | Frequency | Number of Samples | Timespan |
|---|---|---|---|---|---|
| Delta Airlines (DAL) | {6, 8, 10, 12} | 1 | Bi-daily | 1358 | 2009 - 2020 |
| IBM (IBM) | {6, 8, 10, 12} | 1 | Bi-daily | 493 | 2016 - 2020 |
| JPMorgan Chase (JPM) | {6, 8, 10, 12} | 1 | Bi-daily | 565 | 2018 - 2020 |
| NVIDIA (NVDA) | {6, 8, 10, 12} | 1 | Bi-daily | 1203 | 2011 - 2020 |
| Pfizer (PFE) | {6, 8, 10, 12} | 1 | Bi-daily | 812 | 2016 - 2020 |
| Tesla (TSLA) | {6, 8, 10, 12} | 1 | Bi-daily | 294 | 2019 - 2020 |

Table 16: Overview of the numerical data in the FNF (From News to Forecast) datasets (Wang et al., 2024b). "Prediction Length" refers to the number of future time points to be forecasted, with each dataset including four distinct prediction settings. refers to the number of variables (or variates) in each dataset.

| Dataset Name/Domain | Prediction Length | Dimension | Frequency | Number of Samples | Timespan |
|---|---|---|---|---|---|
| Bitcoin Price | {6, 8, 10, 12} | 1 | Daily | 1237 | 2018 - 2021 |
| Web Traffic | {6, 8, 10, 12} | 1 | Daily | 728 | 2015 - 2016 |
| Electricity Demand | {6, 8, 10, 12} | 1 | Daily | 1097 | 2019 - 2021 |

## D.2 HYPERPARAMETERS

We use Adam optimizer (Kingma & Ba, 2015) when training the neural networks. The default choices of hyperparameters in our code are provided in Table 17. For LLM-based text encoders, we initialize them using the default configurations provided by Hugging Face[1]. Consistent with existing works (Wu et al., 2023a; Liu et al., 2024c), we apply instance normalization to standardize the time series data within each dataset.

## D.3 METRICS

Throughout this paper, we use the following metrics to evaluate performance:

MSE (Mean Squared Error): Measures the average squared difference between the predicted and actual values. It penalizes larger errors more heavily, making it sensitive to outliers.

$$\text{MSE} = \frac{1}{n} \sum_{i=1}^{n} (y_i - \hat{y}_i)^2, \tag{30}$$

where $y_i$ and $\hat{y}_i$ denote the ground truth and predicted values, respectively, and $n$ is the number of data points.

MAE (Mean Absolute Error): Represents the average absolute difference between the predicted and actual values, providing a more interpretable measure of average error magnitude.

$$\text{MAE} = \frac{1}{n} \sum_{i=1}^{n} |y_i - \hat{y}_i| . \tag{31}$$

RMSE (Root Mean Squared Error): The square root of MSE, which provides an error measure in the same units as the original data. It is more sensitive to large deviations than MAE.

$$\text{RMSE} = \sqrt{\frac{1}{n} \sum_{i=1}^{n} (y_i - \hat{y}_i)^2}. \tag{32}$$

---

[1] https://huggingface.co/

Table 17: Default hyperparameters for the TaTS framework

| Hyperparameter | Description | Value or Choices |
|---|---|---|
| batch_size | The batch size for training | 32 |
| criterion | The criterion for calculating loss | Mean Square Error (MSE) |
| learning_rate | The learning rate for the optimizer | {0.0001, 0.00005, 0.00001} |
| seq_len | Input sequence length | 24 |
| label_len | Start token length for prediction | 12 |
| prior_weight | Weight for prior combination | {0, 0.1, 0.2, 0.3, 0.5} |
| train_epochs | Number of training epochs | 50 |
| patience | Early stopping patience | 20 |
| text_emb | Dimension of text embeddings | {6, 12, 24} |
| learning_rate2 | Learning rate for MLP layers | {0.005, 0.01, 0.02, 0.05} |
| pool_type | Pooling type for embeddings | "avg" |
| init_method | Initialization method for combined weights | "normal" |
| dropout | dropout | 0.1 |
| use_norm | whether to use normalize | True |

MAPE (Mean Absolute Percentage Error): Expresses errors as a percentage of the actual values, offering a scale-independent metric that facilitates comparisons across datasets.

$$\text{MAPE} = \frac{1}{n} \sum_{i=1}^{n} \left| \frac{y_i - \hat{y}_i}{y_i} \right| \times 100. \tag{33}$$

MSPE (Mean Squared Percentage Error): Similar to MAPE but squares the percentage error, penalizing larger percentage deviations more heavily.

$$\text{MSPE} = \frac{1}{n} \sum_{i=1}^{n} \left( \frac{y_i - \hat{y}_i}{y_i} \right)^2. \tag{34}$$

These metrics collectively provide a comprehensive evaluation of model performance, capturing both absolute and relative errors as well as their sensitivity to outliers. For all metrics, **lower values indicate better performance**.

### D.4 IMPLEMENTATION DETAILS

#### D.4.1 CODE AND REPRODUCIBILITY

The code for the experiments is included in the supplementary material, accompanied by a comprehensive README file. We provide detailed commands, scripts, and instructions to facilitate running the code. Additionally, the datasets used in the experiments are provided in the supplementary material as CSV files.

#### D.4.2 HARDWARE AND ENVIRONMENT

We conducted all experiments on an Ubuntu 22.04 machine equipped with an Intel(R) Xeon(R) Gold 6240R CPU @ 2.40GHz, 1.5TB of RAM, and a 32GB NVIDIA V100 GPU. The CUDA version used was 12.4. All algorithms were implemented in Python (version 3.11.11). To run our code, users must install several commonly used libraries, including pandas, scikit-learn, patool, tqdm, sktime, matplotlib, transformers, and others. Detailed installation instructions can be found in the README file within the code directory. We have optimized our code to ensure efficiency. Our tests confirmed that the CPU memory usage remains below 16 GB, while the GPU memory usage is under 20 GB. Additionally, the execution time for a single experiment is less than 10 minutes on our machine.

### D.4.3 DATA SPLITTING AND LEAKAGE PREVENTION

To ensure forecasting without access to contemporaneous or future information, we adopt a chronological data-splitting protocol for all datasets. Timestamps are divided into $80\%$ training, $10\%$ validation, and $10\%$ test windows without shuffling. For a forecast at timestamp $T$, the model receives only time-series values and textual inputs with timestamps $\leq T - 1$, while all inputs with timestamps $\geq T$ are masked during prediction.

**Timestamp Alignment of Texts.** All datasets provide pre-aligned text–time-series pairs by associating each text sample with the timestamp at which it originally became publicly available (e.g., publication or release time). We directly adopt these timestamps, ensuring that the textual inputs available to the model at time $T - 1$ correctly reflect real-time accessibility.

**Auditing for Retrospective Leakage.** Although we do not apply automated filtering procedures, we conduct human audits on random subsets of samples from each dataset to detect retrospective or outcome-summarizing leakage (e.g., texts describing events that occur after the associated timestamp). Across all datasets, we did not observe such leakage. Combined with causal masking, this protocol prevents the models from accessing contemporaneous or future information from either modality.

# E    FULL EXPERIMENT RESULTS

## E.1    WHY SHOULD WE LEVERAGE CTR?

Table 18: Performance on time series forecasting task. Leveraging periodicity with a single-dimension feature can significantly reduce the prediction error.

| Method | Economy | | Social Good | | Traffic | |
|---|---|---|---|---|---|---|
| | **MSE($\downarrow$)** | **MAE($\downarrow$)** | **MSE($\downarrow$)** | **MAE($\downarrow$)** | **MSE($\downarrow$)** | **MAE($\downarrow$)** |
| Uniformly Random ($+$) | 5.673 | 2.356 | 2.059 | 1.230 | 1.207 | 0.995 |
| Uniformly Random ($\pm$) | 11.535 | 2.879 | 8.860 | 2.404 | 3.794 | 1.618 |
| Normally Random | 9.284 | 2.878 | 2.926 | 1.374 | 3.163 | 1.511 |
| Exponentially Random | 4.521 | 1.960 | 3.724 | 1.564 | 1.141 | 0.911 |
| **Using 1D Text only** | **1.995** | **1.404** | **1.315** | **0.853** | **0.714** | **0.797** |

Parallel text provides complementary information and expert knowledge that can significantly enhance the understanding of time series data. To demonstrate the benefits of utilizing periodicity in the text modality, we present an illustrative example in the univariate forecasting task, where the goal is to use $\mathbf{X}_{1:T} = \{\mathbf{x}_1\} \in \mathbb{R}^{T \times 1}$ to predict the next $H$ values, $\widehat{\mathbf{X}}_{T+1:T+H}$. We first concatenate the text embeddings to form $\mathbf{E} = [e_1; e_2; \ldots; e_T]_{\text{dim}=1} \in \mathbb{R}^{d_{\text{text}} \times T}$, then replace $\mathbf{x}_1$ with only the first dimension of the text embeddings to leverage very partial text periodicity, $\mathbf{x}_1' = (\mathbf{E}[1,:])^\intercal \in \mathbb{R}^{T \times 1}$. In other words, we rely solely on a single evolving dimension of the paired text features to forecast future time series values.

As shown in Table 18, leveraging just the periodicity of a single text feature significantly outperforms random time series forecasting. The random baselines mean that the forecasts are purely random according to several different distributions. These random baselines use the training data as well, for example, normally random computes the mean and standard deviation of a normal distribution to forecast the time series. These results highlight that even partial periodicity from the text modality contributes valuable insights for improving forecasting accuracy.

## E.2    FULL FORECASTING PERFORMANCE COMPARISON VISUALIZATION

To provide a comprehensive comparison of different frameworks for modeling time series with paired texts, we visualize the forecasting performance using radar plots in Figure E.2. Each subfigure corresponds to a dataset, with each axis representing a different time series model. The axes are inverted, where values closer to the center indicate worse performance, and larger areas signify better results. The results demonstrate that TaTS consistently outperforms both baselines across all datasets while maintaining compatibility with various time series models.

## E.3    FULL FORECASTING RESULTS

Due to space limitations, we provide the full results of the time series forecasting task on paired time series and text in the appendix. We conduct extensive experiments across 9 datasets using 9 existing time series models, evaluating various prediction lengths as detailed in Table 14. The complete results are presented from Table 19 to Table 27. Overall, TaTS consistently achieves the best performance across all datasets, time series models, and prediction lengths. The averaged results across all prediction lengths are summarized in the main text (Table 2). For better readability, we also visualize the performance of different frameworks using radar plots, as detailed in Appendix E.2 and Figure 10.

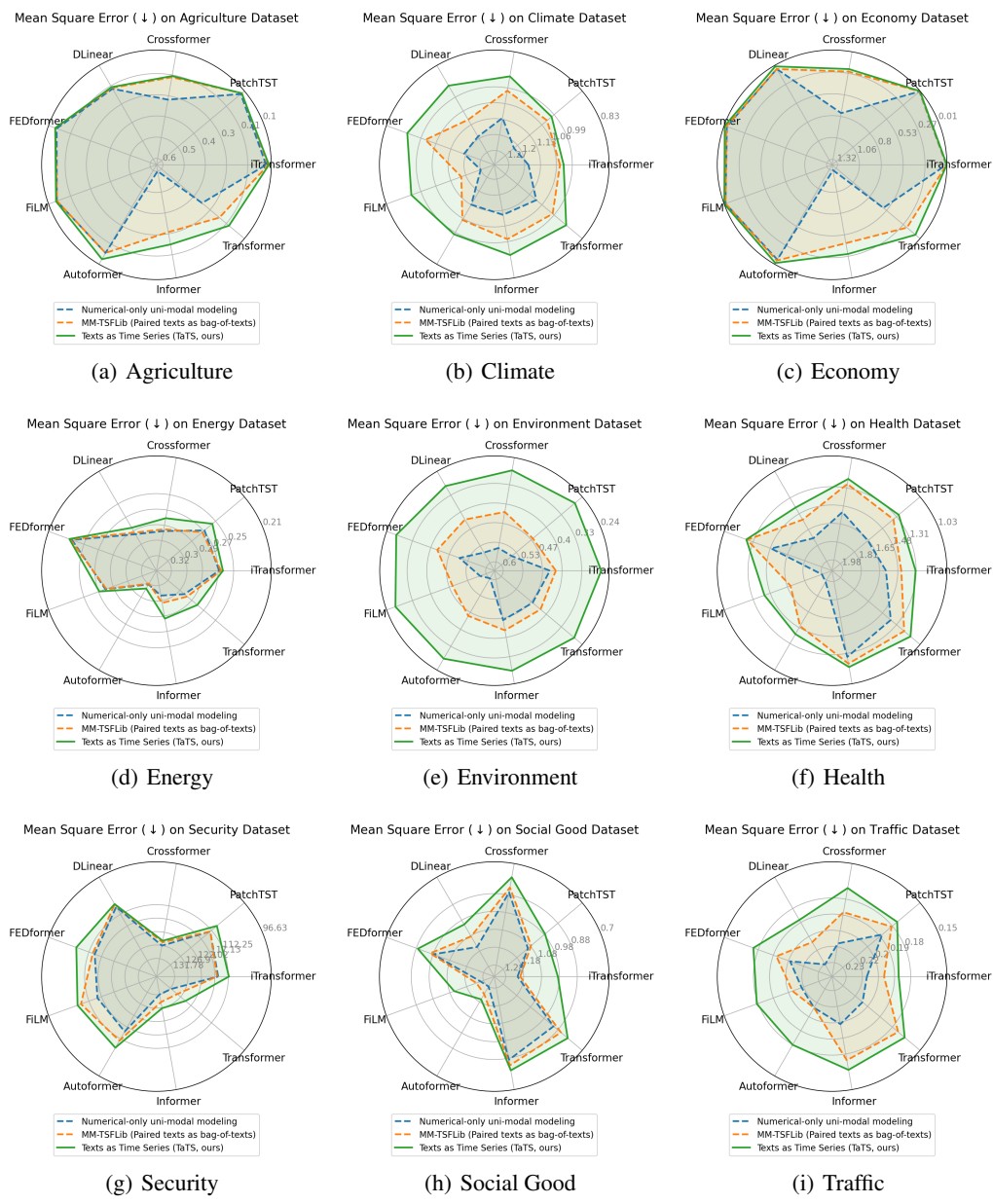

Figure 10: Comparison of different frameworks for modeling time series with paired texts. Our TaTS achieves the best performance across all datasets and is compatible with various existing time series models.

Table 19: Full forecasting results for the Agriculture, Climate, and Economy datasets using iTransformer, PatchTST, and Crossformer as time series models. Compared to numerical-only unimodal modeling and MM-TSFLib, our TaTS framework seamlessly enhances existing time series models to effectively handle time series with concurrent texts. Avg: the average results across all prediction lengths.

| | Models | | iTransformer (2024c) | | | | | PatchTST (2023) | | | | | Crossformer (2023) | | | | |
|---|---|---|---|---|---|---|---|---|---|---|---|---|---|---|---|---|---|
| | Method | | MSE | MAE | RMSE | MAPE | MSPE | MSE | MAE | RMSE | MAPE | MSPE | MSE | MAE | RMSE | MAPE | MSPE |
| Agriculture | Uni-modal | 6 | 0.077 | 0.200 | 0.274 | 0.090 | 0.014 | 0.074 | 0.197 | 0.269 | 0.090 | 0.014 | 0.222 | 0.331 | 0.412 | 0.136 | 0.031 |
| | | 8 | 0.104 | 0.228 | 0.315 | 0.100 | 0.017 | 0.104 | 0.234 | 0.317 | 0.104 | 0.018 | 0.304 | 0.406 | 0.496 | 0.168 | 0.044 |
| | | 10 | 0.142 | 0.273 | 0.372 | 0.119 | 0.024 | 0.136 | 0.262 | 0.357 | 0.112 | 0.021 | 0.357 | 0.435 | 0.530 | 0.176 | 0.049 |
| | | 12 | 0.167 | 0.301 | 0.400 | 0.128 | 0.026 | 0.168 | 0.294 | 0.396 | 0.124 | 0.025 | 0.409 | 0.451 | 0.582 | 0.181 | 0.054 |
| | | Avg | 0.122 | 0.251 | 0.340 | 0.109 | 0.020 | 0.120 | 0.247 | 0.335 | 0.107 | 0.020 | 0.323 | 0.406 | 0.505 | 0.165 | 0.044 |
| | MM-TSFLib | 6 | 0.070 | 0.189 | 0.261 | 0.085 | 0.013 | 0.071 | 0.183 | 0.261 | 0.082 | 0.012 | 0.146 | 0.259 | 0.331 | 0.106 | 0.020 |
| | | 8 | 0.091 | 0.212 | 0.296 | 0.093 | 0.016 | 0.093 | 0.212 | 0.296 | 0.093 | 0.015 | 0.171 | 0.278 | 0.354 | 0.112 | 0.022 |
| | | 10 | 0.130 | 0.248 | 0.346 | 0.105 | 0.019 | 0.126 | 0.251 | 0.344 | 0.108 | 0.020 | 0.252 | 0.344 | 0.428 | 0.136 | 0.032 |
| | | 12 | 0.158 | 0.272 | 0.377 | 0.112 | 0.022 | 0.168 | 0.288 | 0.391 | 0.119 | 0.024 | 0.304 | 0.372 | 0.471 | 0.144 | 0.036 |
| | | Avg | 0.112 | 0.230 | 0.320 | 0.099 | 0.018 | 0.114 | 0.233 | 0.323 | 0.100 | 0.018 | 0.218 | 0.313 | 0.396 | 0.124 | 0.027 |
| | TaTS (ours) | 6 | 0.067 | 0.184 | 0.256 | 0.083 | 0.012 | 0.066 | 0.171 | 0.247 | 0.076 | 0.011 | 0.148 | 0.264 | 0.332 | 0.108 | 0.020 |
| | | 8 | 0.094 | 0.210 | 0.297 | 0.091 | 0.015 | 0.096 | 0.217 | 0.300 | 0.094 | 0.016 | 0.197 | 0.298 | 0.374 | 0.119 | 0.026 |
| | | 10 | 0.122 | 0.251 | 0.341 | 0.109 | 0.020 | 0.126 | 0.260 | 0.349 | 0.113 | 0.021 | 0.216 | 0.315 | 0.397 | 0.124 | 0.028 |
| | | 12 | 0.153 | 0.271 | 0.374 | 0.112 | 0.022 | 0.166 | 0.292 | 0.394 | 0.123 | 0.025 | 0.289 | 0.371 | 0.466 | 0.145 | 0.036 |
| | | Avg | 0.109 | 0.229 | 0.317 | 0.099 | 0.017 | 0.114 | 0.235 | 0.392 | 0.101 | 0.018 | 0.212 | 0.312 | 0.392 | 0.124 | 0.027 |
| Climate | Uni-modal | 6 | 1.127 | 0.843 | 1.052 | 2.549 | 50.662 | 1.259 | 0.915 | 1.122 | 3.168 | 105.890 | 1.159 | 0.852 | 1.076 | 3.036 | 148.840 |
| | | 8 | 1.191 | 0.876 | 1.088 | 3.208 | 134.908 | 1.208 | 0.878 | 1.099 | 2.778 | 64.950 | 1.104 | 0.829 | 1.051 | 2.600 | 123.988 |
| | | 10 | 1.215 | 0.885 | 1.100 | 3.169 | 123.266 | 1.218 | 0.894 | 1.103 | 2.746 | 55.831 | 1.105 | 0.829 | 1.057 | 3.246 | 193.108 |
| | | 12 | 1.199 | 0.879 | 1.091 | 2.752 | 65.916 | 1.197 | 0.891 | 1.092 | 3.177 | 115.617 | 1.105 | 0.837 | 1.047 | 3.322 | 194.844 |
| | | Avg | 1.183 | 0.871 | 1.083 | 2.920 | 93.688 | 1.220 | 0.895 | 1.104 | 2.967 | 85.572 | 1.124 | 0.837 | 1.058 | 3.051 | 165.195 |
| | MM-TSFLib | 6 | 1.031 | 0.787 | 1.013 | 2.298 | 38.389 | 1.003 | 0.800 | 1.001 | 2.652 | 57.709 | 0.995 | 0.758 | 0.996 | 2.393 | 51.345 |
| | | 8 | 1.039 | 0.809 | 1.017 | 2.598 | 49.903 | 1.012 | 0.790 | 1.005 | 2.387 | 42.258 | 1.016 | 0.773 | 1.007 | 2.824 | 107.865 |
| | | 10 | 1.049 | 0.817 | 1.020 | 2.547 | 55.014 | 1.036 | 0.813 | 1.016 | 2.293 | 28.894 | 0.999 | 0.779 | 0.997 | 2.740 | 95.790 |
| | | 12 | 1.057 | 0.828 | 1.025 | 2.894 | 87.976 | 1.071 | 0.822 | 1.032 | 2.965 | 95.248 | 0.997 | 0.777 | 0.994 | 2.240 | 43.968 |
| | | Avg | 1.044 | 0.810 | 1.019 | 2.584 | 57.821 | 1.030 | 0.806 | 1.014 | 2.574 | 56.027 | 1.002 | 0.772 | 0.998 | 2.549 | 74.742 |
| | TaTS (ours) | 6 | 1.020 | 0.797 | 1.007 | 2.563 | 56.209 | 0.976 | 0.782 | 0.987 | 2.318 | 33.843 | 0.924 | 0.747 | 0.961 | 1.895 | 20.405 |
| | | 8 | 1.025 | 0.797 | 1.011 | 2.391 | 38.756 | 0.995 | 0.803 | 0.997 | 2.574 | 57.639 | 0.923 | 0.757 | 0.961 | 2.318 | 51.592 |
| | | 10 | 1.033 | 0.808 | 1.014 | 2.647 | 77.697 | 1.022 | 0.796 | 1.007 | 2.657 | 76.287 | 0.963 | 0.764 | 0.979 | 2.439 | 59.881 |
| | | 12 | 1.033 | 0.812 | 1.013 | 2.607 | 58.107 | 1.022 | 0.810 | 1.009 | 2.436 | 42.471 | 0.943 | 0.754 | 0.967 | 2.220 | 40.186 |
| | | Avg | 1.028 | 0.804 | 1.011 | 2.552 | 57.692 | 1.004 | 0.798 | 1.000 | 2.496 | 52.560 | 0.938 | 0.755 | 0.967 | 2.218 | 43.016 |
| Economy | Uni-modal | 6 | 0.015 | 0.099 | 0.124 | 0.035 | 0.002 | 0.017 | 0.104 | 0.129 | 0.036 | 0.002 | 0.659 | 0.749 | 0.806 | 0.257 | 0.076 |
| | | 8 | 0.014 | 0.098 | 0.120 | 0.034 | 0.002 | 0.016 | 0.104 | 0.128 | 0.036 | 0.002 | 0.661 | 0.767 | 0.808 | 0.263 | 0.076 |
| | | 10 | 0.014 | 0.094 | 0.119 | 0.033 | 0.002 | 0.017 | 0.104 | 0.130 | 0.036 | 0.002 | 0.836 | 0.884 | 0.911 | 0.303 | 0.097 |
| | | 12 | 0.013 | 0.091 | 0.112 | 0.032 | 0.002 | 0.018 | 0.109 | 0.135 | 0.037 | 0.002 | 0.875 | 0.912 | 0.932 | 0.313 | 0.101 |
| | | Avg | 0.014 | 0.096 | 0.119 | 0.034 | 0.002 | 0.017 | 0.105 | 0.131 | 0.036 | 0.002 | 0.758 | 0.828 | 0.864 | 0.284 | 0.087 |
| | MM-TSFLib | 6 | 0.011 | 0.081 | 0.103 | 0.028 | 0.001 | 0.014 | 0.094 | 0.118 | 0.033 | 0.002 | 0.209 | 0.420 | 0.450 | 0.143 | 0.024 |
| | | 8 | 0.011 | 0.085 | 0.106 | 0.029 | 0.001 | 0.015 | 0.099 | 0.123 | 0.034 | 0.002 | 0.214 | 0.424 | 0.456 | 0.145 | 0.024 |
| | | 10 | 0.012 | 0.090 | 0.110 | 0.031 | 0.001 | 0.013 | 0.091 | 0.115 | 0.031 | 0.002 | 0.277 | 0.463 | 0.522 | 0.158 | 0.032 |
| | | 12 | 0.012 | 0.090 | 0.110 | 0.031 | 0.001 | 0.016 | 0.099 | 0.124 | 0.034 | 0.002 | 0.299 | 0.526 | 0.540 | 0.180 | 0.034 |
| | | Avg | 0.011 | 0.086 | 0.107 | 0.030 | 0.001 | 0.014 | 0.096 | 0.120 | 0.033 | 0.002 | 0.250 | 0.458 | 0.492 | 0.156 | 0.029 |
| | TaTS (ours) | 6 | 0.008 | 0.077 | 0.090 | 0.027 | 0.001 | 0.009 | 0.080 | 0.097 | 0.028 | 0.001 | 0.140 | 0.312 | 0.367 | 0.106 | 0.016 |
| | | 8 | 0.008 | 0.077 | 0.090 | 0.027 | 0.001 | 0.008 | 0.078 | 0.091 | 0.027 | 0.001 | 0.212 | 0.426 | 0.454 | 0.145 | 0.024 |
| | | 10 | 0.009 | 0.079 | 0.093 | 0.027 | 0.001 | 0.009 | 0.079 | 0.092 | 0.027 | 0.001 | 0.302 | 0.510 | 0.544 | 0.174 | 0.034 |
| | | 12 | 0.008 | 0.076 | 0.091 | 0.026 | 0.001 | 0.009 | 0.080 | 0.096 | 0.028 | 0.001 | 0.222 | 0.428 | 0.466 | 0.145 | 0.025 |
| | | Avg | 0.008 | 0.077 | 0.091 | 0.027 | 0.001 | 0.009 | 0.079 | 0.094 | 0.028 | 0.001 | 0.219 | 0.419 | 0.458 | 0.142 | 0.025 |

Table 20: Full forecasting results for the Energy, Environment, and Health datasets using iTransformer, PatchTST, and Crossformer as time series models. Compared to numerical-only unimodal modeling and MM-TSFLib, our TaTS framework seamlessly enhances existing time series models to effectively handle time series with concurrent texts. Avg: the average results across all prediction lengths.

| | Models | | iTransformer (2024c) | | | | | PatchTST (2023) | | | | | Crossformer (2023) | | | | |
|---|---|---|---|---|---|---|---|---|---|---|---|---|---|---|---|---|---|
| | Method | | MSE | MAE | RMSE | MAPE | MSPE | MSE | MAE | RMSE | MAPE | MSPE | MSE | MAE | RMSE | MAPE | MSPE |
| Energy | Uni-modal | 12 | 0.112 | 0.231 | 0.305 | 0.940 | 10.877 | 0.105 | 0.229 | 0.301 | 1.120 | 40.016 | 0.138 | 0.262 | 0.338 | 1.121 | 31.808 |
| | | 24 | 0.222 | 0.352 | 0.438 | 1.626 | 35.711 | 0.241 | 0.363 | 0.458 | 1.507 | 41.679 | 0.281 | 0.399 | 0.482 | 1.619 | 46.008 |
| | | 36 | 0.306 | 0.409 | 0.511 | 1.767 | 58.611 | 0.304 | 0.408 | 0.512 | 1.782 | 60.280 | 0.331 | 0.443 | 0.541 | 3.296 | 229.673 |
| | | 48 | 0.435 | 0.509 | 0.617 | 2.454 | 95.250 | 0.427 | 0.502 | 0.610 | 2.405 | 91.255 | 0.422 | 0.521 | 0.626 | 4.574 | 347.533 |
| | | Avg | 0.269 | 0.375 | 0.468 | 1.697 | 50.112 | 0.269 | 0.376 | 0.470 | 1.704 | 58.307 | 0.293 | 0.406 | 0.497 | 2.652 | 163.756 |
| | MM-TSFLib | 12 | 0.107 | 0.228 | 0.300 | 0.999 | 17.267 | 0.115 | 0.242 | 0.311 | 1.357 | 30.739 | 0.126 | 0.251 | 0.326 | 1.141 | 30.653 |
| | | 24 | 0.223 | 0.354 | 0.443 | 1.692 | 45.580 | 0.236 | 0.361 | 0.453 | 1.506 | 35.353 | 0.268 | 0.389 | 0.477 | 1.875 | 69.289 |
| | | 36 | 0.313 | 0.424 | 0.520 | 1.916 | 63.383 | 0.311 | 0.413 | 0.518 | 1.885 | 64.280 | 0.338 | 0.454 | 0.553 | 2.956 | 202.647 |
| | | 48 | 0.423 | 0.507 | 0.615 | 2.494 | 101.822 | 0.426 | 0.501 | 0.610 | 2.404 | 95.489 | 0.433 | 0.534 | 0.637 | 4.716 | 415.767 |
| | | Avg | 0.267 | 0.378 | 0.469 | 1.775 | 57.013 | 0.272 | 0.379 | 0.473 | 1.788 | 56.465 | 0.291 | 0.407 | 0.498 | 2.672 | 179.589 |
| | TaTS (ours) | 12 | 0.106 | 0.234 | 0.302 | 1.116 | 30.716 | 0.106 | 0.234 | 0.300 | 1.024 | 17.615 | 0.127 | 0.254 | 0.326 | 1.053 | 18.277 |
| | | 24 | 0.226 | 0.355 | 0.439 | 1.555 | 34.060 | 0.206 | 0.336 | 0.417 | 1.491 | 35.509 | 0.253 | 0.376 | 0.462 | 1.823 | 62.566 |
| | | 36 | 0.306 | 0.411 | 0.512 | 1.790 | 57.729 | 0.305 | 0.412 | 0.506 | 1.781 | 57.481 | 0.312 | 0.431 | 0.524 | 2.602 | 144.944 |
| | | 48 | 0.421 | 0.502 | 0.612 | 2.370 | 91.346 | 0.416 | 0.501 | 0.607 | 2.427 | 93.658 | 0.425 | 0.516 | 0.619 | 3.314 | 183.016 |
| | | Avg | 0.265 | 0.376 | 0.466 | 1.708 | 53.463 | 0.258 | 0.371 | 0.457 | 1.681 | 51.066 | 0.279 | 0.394 | 0.483 | 2.198 | 102.201 |
| Environment | Uni-modal | 48 | 0.415 | 0.473 | 0.604 | 2.218 | 214.087 | 0.492 | 0.500 | 0.644 | 2.315 | 238.477 | 0.495 | 0.509 | 0.651 | 2.275 | 235.830 |
| | | 96 | 0.439 | 0.493 | 0.630 | 2.338 | 250.510 | 0.541 | 0.538 | 0.693 | 2.390 | 256.244 | 0.562 | 0.585 | 0.712 | 1.880 | 130.832 |
| | | 192 | 0.454 | 0.506 | 0.661 | 2.462 | 259.311 | 0.581 | 0.556 | 0.741 | 2.727 | 356.251 | 0.567 | 0.607 | 0.739 | 1.659 | 90.234 |
| | | 336 | 0.455 | 0.505 | 0.671 | 2.446 | 276.348 | 0.593 | 0.553 | 0.763 | 2.898 | 424.764 | 0.582 | 0.621 | 0.761 | 1.656 | 80.409 |
| | | Avg | 0.441 | 0.494 | 0.641 | 2.366 | 250.064 | 0.552 | 0.537 | 0.710 | 2.583 | 318.934 | 0.551 | 0.581 | 0.716 | 1.867 | 134.326 |
| | MM-TSFLib | 48 | 0.413 | 0.470 | 0.600 | 2.187 | 211.597 | 0.441 | 0.489 | 0.621 | 1.954 | 159.971 | 0.421 | 0.472 | 0.604 | 1.982 | 161.068 |
| | | 96 | 0.420 | 0.478 | 0.616 | 2.233 | 213.865 | 0.461 | 0.495 | 0.643 | 2.063 | 177.553 | 0.425 | 0.485 | 0.617 | 1.839 | 128.131 |
| | | 192 | 0.423 | 0.482 | 0.636 | 2.320 | 225.635 | 0.462 | 0.508 | 0.664 | 2.203 | 214.665 | 0.427 | 0.495 | 0.638 | 1.830 | 125.150 |
| | | 336 | 0.429 | 0.483 | 0.651 | 2.308 | 236.965 | 0.472 | 0.513 | 0.683 | 2.210 | 220.009 | 0.434 | 0.500 | 0.656 | 1.867 | 128.536 |
| | | Avg | 0.421 | 0.478 | 0.626 | 2.262 | 222.016 | 0.459 | 0.501 | 0.653 | 2.107 | 193.049 | 0.427 | 0.488 | 0.629 | 1.879 | 135.721 |
| | TaTS (ours) | 48 | 0.268 | 0.370 | 0.480 | 1.140 | 21.157 | 0.271 | 0.377 | 0.483 | 1.115 | 19.355 | 0.274 | 0.373 | 0.485 | 1.152 | 20.952 |
| | | 96 | 0.267 | 0.370 | 0.488 | 1.123 | 20.955 | 0.279 | 0.376 | 0.491 | 1.176 | 21.868 | 0.284 | 0.391 | 0.505 | 1.127 | 18.484 |
| | | 192 | 0.272 | 0.366 | 0.508 | 1.215 | 23.115 | 0.272 | 0.366 | 0.508 | 1.215 | 23.342 | 0.283 | 0.415 | 0.523 | 1.039 | 14.150 |
| | | 336 | 0.261 | 0.369 | 0.508 | 1.174 | 22.437 | 0.269 | 0.366 | 0.516 | 1.222 | 23.081 | 0.294 | 0.431 | 0.541 | 1.031 | 12.734 |
| | | Avg | 0.267 | 0.369 | 0.496 | 1.163 | 21.916 | 0.273 | 0.371 | 0.501 | 1.182 | 21.912 | 0.284 | 0.403 | 0.513 | 1.087 | 16.580 |
| Health | Uni-modal | 12 | 1.171 | 0.674 | 0.967 | 2.596 | 149.272 | 1.267 | 0.735 | 0.999 | 3.651 | 262.953 | 1.453 | 0.809 | 1.086 | 2.699 | 145.979 |
| | | 24 | 1.594 | 0.807 | 1.169 | 2.832 | 136.133 | 1.681 | 0.846 | 1.167 | 3.371 | 160.263 | 1.537 | 0.825 | 1.153 | 2.744 | 100.232 |
| | | 36 | 1.742 | 0.862 | 1.253 | 2.893 | 118.496 | 1.819 | 0.918 | 1.281 | 3.751 | 243.238 | 1.565 | 0.831 | 1.173 | 2.687 | 108.731 |
| | | 48 | 1.840 | 0.923 | 1.313 | 3.263 | 153.559 | 1.842 | 0.921 | 1.310 | 3.192 | 152.876 | 1.586 | 0.841 | 1.208 | 2.730 | 165.914 |
| | | Avg | 1.587 | 0.817 | 1.175 | 2.896 | 139.365 | 1.652 | 0.855 | 1.189 | 3.491 | 204.832 | 1.535 | 0.827 | 1.155 | 2.715 | 130.214 |
| | MM-TSFLib | 12 | 0.987 | 0.696 | 0.928 | 3.334 | 190.818 | 1.029 | 0.710 | 0.945 | 3.400 | 182.961 | 1.017 | 0.664 | 0.933 | 2.796 | 171.177 |
| | | 24 | 1.388 | 0.787 | 1.113 | 3.457 | 169.217 | 1.288 | 0.769 | 1.053 | 3.207 | 129.039 | 1.318 | 0.747 | 1.072 | 2.917 | 161.566 |
| | | 36 | 1.672 | 0.877 | 1.216 | 3.674 | 176.904 | 1.460 | 0.831 | 1.152 | 3.492 | 168.807 | 1.357 | 0.775 | 1.113 | 3.210 | 192.422 |
| | | 48 | 1.737 | 0.902 | 1.270 | 3.531 | 155.800 | 1.612 | 0.878 | 1.231 | 3.317 | 127.827 | 1.399 | 0.788 | 1.146 | 2.821 | 149.248 |
| | | Avg | 1.446 | 0.816 | 1.132 | 3.499 | 173.185 | 1.347 | 0.797 | 1.095 | 3.354 | 152.159 | 1.273 | 0.744 | 1.066 | 2.936 | 168.603 |
| | TaTS (ours) | 12 | 0.939 | 0.649 | 0.892 | 2.764 | 138.387 | 0.990 | 0.659 | 0.912 | 2.721 | 151.988 | 0.974 | 0.661 | 0.911 | 2.838 | 163.149 |
| | | 24 | 1.251 | 0.712 | 1.032 | 2.744 | 112.080 | 1.288 | 0.764 | 1.050 | 3.364 | 155.937 | 1.254 | 0.741 | 1.062 | 3.104 | 170.871 |
| | | 36 | 1.489 | 0.781 | 1.147 | 2.885 | 112.555 | 1.397 | 0.780 | 1.122 | 2.638 | 93.114 | 1.306 | 0.741 | 1.083 | 2.805 | 152.713 |
| | | 48 | 1.581 | 0.834 | 1.215 | 2.909 | 107.995 | 1.456 | 0.808 | 1.165 | 2.623 | 88.216 | 1.369 | 0.770 | 1.136 | 2.770 | 144.765 |
| | | Avg | 1.315 | 0.744 | 1.071 | 2.825 | 117.754 | 1.283 | 0.753 | 1.062 | 2.837 | 122.314 | 1.226 | 0.728 | 1.048 | 2.879 | 157.874 |

Table 21: Full forecasting results for the Security, Social Good, and Traffic datasets using iTransformer, PatchTST, and Crossformer as time series models. Compared to numerical-only unimodal modeling and MM-TSFLib, our TaTS framework seamlessly enhances existing time series models to effectively handle time series with concurrent texts. Avg: the average results across all prediction lengths.

| Models | | | iTransformer (2024c) | | | | | PatchTST (2023) | | | | | Crossformer (2023) | | | | |
|---|---|---|---|---|---|---|---|---|---|---|---|---|---|---|---|---|---|
| | Method | | MSE | MAE | RMSE | MAPE | MSPE | MSE | MAE | RMSE | MAPE | MSPE | MSE | MAE | RMSE | MAPE | MSPE |
| Security | Uni-modal | 6 | 113.573 | 5.698 | 10.657 | 6.008 | 1020.167 | 108.795 | 5.119 | 10.430 | 4.255 | 549.239 | 124.972 | 6.077 | 11.179 | 2.435 | 113.813 |
| | | 8 | 115.878 | 5.743 | 10.765 | 5.977 | 1035.504 | 113.534 | 5.524 | 10.655 | 3.681 | 424.901 | 126.513 | 6.237 | 11.248 | 1.909 | 67.155 |
| | | 10 | 116.950 | 5.615 | 10.814 | 3.109 | 332.477 | 114.820 | 5.484 | 10.715 | 2.523 | 185.108 | 127.701 | 6.346 | 11.300 | 1.678 | 70.642 |
| | | 12 | 117.396 | 5.585 | 10.835 | 2.645 | 262.692 | 114.253 | 5.357 | 10.689 | 1.951 | 117.032 | 128.686 | 6.449 | 11.344 | 1.484 | 35.512 |
| | | Avg | 115.949 | 5.660 | 10.768 | 4.435 | 662.710 | 112.850 | 5.371 | 10.622 | 3.103 | 319.070 | 126.968 | 6.277 | 11.268 | 1.877 | 71.781 |
| | MM-TSFLib | 6 | 115.170 | 5.403 | 10.732 | 4.284 | 482.301 | 109.240 | 5.189 | 10.452 | 4.386 | 560.668 | 123.515 | 5.958 | 11.114 | 2.771 | 175.140 |
| | | 8 | 116.158 | 5.544 | 10.778 | 5.742 | 985.203 | 113.248 | 5.445 | 10.642 | 3.237 | 322.665 | 124.714 | 6.100 | 11.168 | 2.169 | 126.854 |
| | | 10 | 117.340 | 5.633 | 10.832 | 3.090 | 320.102 | 114.109 | 5.442 | 10.682 | 2.500 | 196.618 | 127.701 | 6.346 | 11.300 | 1.678 | 70.642 |
| | | 12 | 116.694 | 5.548 | 10.803 | 2.560 | 243.369 | 114.764 | 5.398 | 10.713 | 2.136 | 165.771 | 126.985 | 6.327 | 11.269 | 1.510 | 42.745 |
| | | Avg | 116.341 | 5.532 | 10.786 | 3.919 | 507.744 | 112.840 | 5.369 | 10.622 | 3.065 | 311.430 | 125.729 | 6.183 | 11.213 | 2.032 | 103.845 |
| | TaTS (ours) | 6 | 107.113 | 4.856 | 10.350 | 3.600 | 320.907 | 106.160 | 4.696 | 10.303 | 3.316 | 282.430 | 122.887 | 5.915 | 11.085 | 2.831 | 191.546 |
| | | 8 | 112.560 | 5.204 | 10.609 | 3.258 | 345.121 | 108.803 | 5.052 | 10.431 | 3.082 | 312.449 | 124.302 | 6.067 | 11.149 | 2.656 | 178.725 |
| | | 10 | 113.789 | 5.227 | 10.667 | 2.488 | 204.306 | 111.110 | 5.090 | 10.541 | 2.423 | 205.196 | 126.203 | 6.258 | 11.234 | 1.720 | 60.667 |
| | | 12 | 114.754 | 5.318 | 10.712 | 2.057 | 140.148 | 112.699 | 5.237 | 10.616 | 2.185 | 171.929 | 127.263 | 6.352 | 11.281 | 1.433 | 36.934 |
| | | Avg | 112.054 | 5.151 | 10.584 | 2.851 | 252.621 | 109.693 | 5.019 | 10.473 | 2.752 | 243.001 | 125.164 | 6.148 | 11.187 | 2.160 | 116.968 |
| Social Good | Uni-modal | 6 | 1.129 | 0.438 | 0.741 | 1.395 | 72.570 | 1.047 | 0.443 | 0.749 | 1.420 | 70.855 | 0.791 | 0.431 | 0.681 | 1.320 | 39.179 |
| | | 8 | 1.133 | 0.466 | 0.770 | 1.448 | 67.692 | 1.102 | 0.458 | 0.741 | 1.502 | 68.611 | 0.832 | 0.414 | 0.668 | 0.917 | 13.263 |
| | | 10 | 1.275 | 0.496 | 0.808 | 1.718 | 112.603 | 1.089 | 0.469 | 0.749 | 1.568 | 75.225 | 0.916 | 0.463 | 0.728 | 0.709 | 6.748 |
| | | 12 | 1.313 | 0.534 | 0.841 | 2.052 | 159.814 | 1.148 | 0.610 | 0.866 | 2.185 | 94.833 | 0.921 | 0.559 | 0.766 | 1.682 | 54.438 |
| | | Avg | 1.212 | 0.483 | 0.790 | 1.653 | 103.170 | 1.097 | 0.495 | 0.776 | 1.669 | 77.381 | 0.865 | 0.467 | 0.711 | 1.157 | 28.407 |
| | MM-TSFLib | 6 | 1.061 | 0.451 | 0.752 | 1.399 | 60.240 | 1.023 | 0.442 | 0.741 | 1.333 | 62.404 | 0.753 | 0.350 | 0.604 | 0.753 | 8.426 |
| | | 8 | 1.175 | 0.498 | 0.789 | 1.390 | 59.283 | 1.111 | 0.533 | 0.807 | 1.295 | 34.722 | 0.814 | 0.357 | 0.618 | 0.576 | 5.614 |
| | | 10 | 1.253 | 0.561 | 0.863 | 1.627 | 66.938 | 1.049 | 0.520 | 0.799 | 1.504 | 54.694 | 0.866 | 0.492 | 0.726 | 1.127 | 10.642 |
| | | 12 | 1.298 | 0.570 | 0.872 | 1.710 | 72.310 | 1.110 | 0.566 | 0.840 | 1.632 | 50.501 | 0.917 | 0.395 | 0.650 | 0.693 | 3.899 |
| | | Avg | 1.197 | 0.520 | 0.819 | 1.531 | 64.693 | 1.073 | 0.515 | 0.797 | 1.441 | 50.580 | 0.837 | 0.398 | 0.649 | 0.787 | 7.145 |
| | TaTS (ours) | 6 | 0.942 | 0.398 | 0.677 | 1.221 | 49.723 | 0.923 | 0.436 | 0.722 | 1.153 | 18.695 | 0.711 | 0.407 | 0.631 | 0.749 | 6.772 |
| | | 8 | 0.967 | 0.433 | 0.713 | 1.430 | 50.480 | 0.900 | 0.461 | 0.713 | 1.279 | 22.919 | 0.748 | 0.453 | 0.646 | 0.963 | 7.987 |
| | | 10 | 0.994 | 0.463 | 0.740 | 1.538 | 54.466 | 0.996 | 0.461 | 0.728 | 1.348 | 42.272 | 0.800 | 0.373 | 0.615 | 0.724 | 5.977 |
| | | 12 | 1.045 | 0.514 | 0.780 | 1.753 | 61.884 | 1.069 | 0.501 | 0.770 | 1.549 | 60.142 | 0.857 | 0.415 | 0.663 | 0.827 | 7.195 |
| | | Avg | 0.987 | 0.452 | 0.728 | 1.486 | 54.138 | 0.972 | 0.465 | 0.733 | 1.332 | 36.007 | 0.779 | 0.412 | 0.639 | 0.816 | 6.983 |
| Traffic | Uni-modal | 6 | 0.203 | 0.228 | 0.393 | 0.216 | 0.307 | 0.182 | 0.252 | 0.377 | 0.285 | 0.513 | 0.227 | 0.394 | 0.472 | 0.340 | 0.388 |
| | | 8 | 0.209 | 0.236 | 0.399 | 0.224 | 0.321 | 0.167 | 0.226 | 0.348 | 0.255 | 0.450 | 0.216 | 0.382 | 0.458 | 0.324 | 0.333 |
| | | 10 | 0.211 | 0.243 | 0.398 | 0.229 | 0.321 | 0.178 | 0.242 | 0.362 | 0.270 | 0.464 | 0.202 | 0.363 | 0.441 | 0.313 | 0.327 |
| | | 12 | 0.231 | 0.246 | 0.392 | 0.281 | 0.532 | 0.226 | 0.250 | 0.393 | 0.300 | 0.580 | 0.212 | 0.365 | 0.450 | 0.337 | 0.399 |
| | | Avg | 0.213 | 0.238 | 0.395 | 0.238 | 0.370 | 0.188 | 0.242 | 0.370 | 0.278 | 0.502 | 0.214 | 0.376 | 0.455 | 0.329 | 0.362 |
| | MM-TSFLib | 6 | 0.187 | 0.338 | 0.422 | 0.292 | 0.324 | 0.165 | 0.229 | 0.353 | 0.248 | 0.391 | 0.184 | 0.335 | 0.418 | 0.307 | 0.371 |
| | | 8 | 0.197 | 0.355 | 0.433 | 0.297 | 0.303 | 0.163 | 0.218 | 0.346 | 0.237 | 0.373 | 0.183 | 0.331 | 0.416 | 0.305 | 0.367 |
| | | 10 | 0.190 | 0.338 | 0.422 | 0.285 | 0.283 | 0.174 | 0.235 | 0.357 | 0.252 | 0.402 | 0.184 | 0.331 | 0.416 | 0.303 | 0.363 |
| | | 12 | 0.222 | 0.358 | 0.459 | 0.333 | 0.410 | 0.211 | 0.237 | 0.378 | 0.281 | 0.524 | 0.200 | 0.340 | 0.431 | 0.329 | 0.427 |
| | | Avg | 0.199 | 0.347 | 0.434 | 0.302 | 0.330 | 0.178 | 0.230 | 0.359 | 0.255 | 0.422 | 0.188 | 0.334 | 0.420 | 0.311 | 0.382 |
| | TaTS (ours) | 6 | 0.174 | 0.218 | 0.351 | 0.227 | 0.348 | 0.155 | 0.204 | 0.334 | 0.225 | 0.359 | 0.159 | 0.275 | 0.372 | 0.273 | 0.392 |
| | | 8 | 0.177 | 0.213 | 0.357 | 0.215 | 0.319 | 0.162 | 0.210 | 0.334 | 0.235 | 0.389 | 0.166 | 0.294 | 0.385 | 0.279 | 0.369 |
| | | 10 | 0.186 | 0.225 | 0.366 | 0.223 | 0.324 | 0.167 | 0.214 | 0.340 | 0.233 | 0.368 | 0.163 | 0.285 | 0.378 | 0.277 | 0.386 |
| | | 12 | 0.213 | 0.212 | 0.361 | 0.241 | 0.451 | 0.204 | 0.209 | 0.356 | 0.249 | 0.468 | 0.184 | 0.291 | 0.395 | 0.296 | 0.428 |
| | | Avg | 0.187 | 0.217 | 0.359 | 0.227 | 0.361 | 0.172 | 0.209 | 0.341 | 0.235 | 0.396 | 0.168 | 0.286 | 0.383 | 0.281 | 0.394 |

A conference paper at ICLR 2026

Table 22: Full forecasting results for the Agriculture, Climate, and Economy datasets using DLinear, FEDformer, and FiLM as time series models. Compared to numerical-only unimodal modeling and MM-TSFLib, our TaTS framework seamlessly enhances existing time series models to effectively handle time series with concurrent texts. Avg: the average results across all prediction lengths.

| Models | | | DLinear (2023) | | | | | FEDformer (2022b) | | | | | FiLM (2022a) | | | | |
|---|---|---|---|---|---|---|---|---|---|---|---|---|---|---|---|---|---|
| | Method | | MSE | MAE | RMSE | MAPE | MSPE | MSE | MAE | RMSE | MAPE | MSPE | MSE | MAE | RMSE | MAPE | MSPE |
| Agriculture | Uni-modal | 6 | 0.170 | 0.312 | 0.395 | 0.135 | 0.028 | 0.091 | 0.239 | 0.301 | 0.110 | 0.019 | 0.088 | 0.207 | 0.290 | 0.092 | 0.015 |
| | | 8 | 0.195 | 0.340 | 0.423 | 0.145 | 0.031 | 0.127 | 0.281 | 0.355 | 0.127 | 0.024 | 0.115 | 0.230 | 0.318 | 0.097 | 0.017 |
| | | 10 | 0.218 | 0.355 | 0.448 | 0.150 | 0.034 | 0.154 | 0.313 | 0.392 | 0.142 | 0.030 | 0.151 | 0.255 | 0.351 | 0.103 | 0.020 |
| | | 12 | 0.308 | 0.407 | 0.514 | 0.165 | 0.041 | 0.180 | 0.313 | 0.415 | 0.134 | 0.028 | 0.200 | 0.333 | 0.433 | 0.140 | 0.030 |
| | | Avg | 0.223 | 0.354 | 0.445 | 0.149 | 0.034 | 0.138 | 0.286 | 0.366 | 0.128 | 0.025 | 0.139 | 0.256 | 0.348 | 0.108 | 0.021 |
| | MM-TSFLib | 6 | 0.166 | 0.314 | 0.396 | 0.137 | 0.028 | 0.079 | 0.219 | 0.280 | 0.102 | 0.016 | 0.089 | 0.208 | 0.291 | 0.092 | 0.015 |
| | | 8 | 0.193 | 0.342 | 0.425 | 0.148 | 0.032 | 0.112 | 0.269 | 0.335 | 0.125 | 0.024 | 0.112 | 0.225 | 0.314 | 0.095 | 0.016 |
| | | 10 | 0.214 | 0.358 | 0.450 | 0.153 | 0.035 | 0.153 | 0.292 | 0.387 | 0.128 | 0.026 | 0.156 | 0.262 | 0.357 | 0.106 | 0.020 |
| | | 12 | 0.300 | 0.408 | 0.514 | 0.167 | 0.042 | 0.180 | 0.319 | 0.416 | 0.137 | 0.028 | 0.204 | 0.338 | 0.437 | 0.142 | 0.031 |
| | | Avg | 0.218 | 0.355 | 0.446 | 0.151 | 0.034 | 0.131 | 0.275 | 0.366 | 0.128 | 0.024 | 0.140 | 0.258 | 0.350 | 0.109 | 0.021 |
| | TaTS (ours) | 6 | 0.164 | 0.311 | 0.392 | 0.136 | 0.028 | 0.082 | 0.222 | 0.286 | 0.102 | 0.016 | 0.087 | 0.205 | 0.289 | 0.091 | 0.015 |
| | | 8 | 0.192 | 0.343 | 0.425 | 0.148 | 0.032 | 0.111 | 0.256 | 0.330 | 0.115 | 0.020 | 0.110 | 0.223 | 0.312 | 0.094 | 0.016 |
| | | 10 | 0.215 | 0.358 | 0.451 | 0.152 | 0.035 | 0.147 | 0.288 | 0.377 | 0.126 | 0.025 | 0.146 | 0.249 | 0.345 | 0.100 | 0.019 |
| | | 12 | 0.287 | 0.392 | 0.496 | 0.159 | 0.038 | 0.183 | 0.339 | 0.418 | 0.145 | 0.029 | 0.196 | 0.328 | 0.429 | 0.138 | 0.029 |
| | | Avg | 0.214 | 0.351 | 0.441 | 0.149 | 0.033 | 0.131 | 0.276 | 0.353 | 0.122 | 0.023 | 0.135 | 0.251 | 0.344 | 0.106 | 0.020 |
| Climate | Uni-modal | 6 | 1.158 | 0.866 | 1.076 | 3.462 | 204.312 | 1.206 | 0.909 | 1.098 | 4.551 | 394.050 | 1.277 | 0.912 | 1.129 | 4.953 | 410.091 |
| | | 8 | 1.191 | 0.874 | 1.091 | 3.400 | 190.434 | 1.175 | 0.897 | 1.084 | 3.680 | 176.984 | 1.158 | 0.862 | 1.076 | 3.282 | 162.026 |
| | | 10 | 1.225 | 0.882 | 1.107 | 3.392 | 185.912 | 1.199 | 0.893 | 1.095 | 3.215 | 134.937 | 1.160 | 0.857 | 1.077 | 2.152 | 48.217 |
| | | 12 | 1.185 | 0.868 | 1.089 | 2.856 | 126.200 | 1.190 | 0.875 | 1.091 | 2.883 | 99.696 | 1.487 | 1.012 | 1.219 | 3.993 | 164.875 |
| | | Avg | 1.190 | 0.872 | 1.091 | 3.277 | 176.715 | 1.192 | 0.893 | 1.092 | 3.582 | 201.417 | 1.270 | 0.911 | 1.125 | 3.595 | 196.302 |
| | MM-TSFLib | 6 | 1.074 | 0.822 | 1.036 | 2.997 | 154.862 | 1.012 | 0.795 | 1.006 | 2.492 | 59.880 | 1.165 | 0.869 | 1.079 | 3.798 | 270.733 |
| | | 8 | 1.108 | 0.844 | 1.053 | 3.117 | 160.555 | 1.003 | 0.787 | 1.001 | 2.265 | 34.593 | 1.173 | 0.867 | 1.083 | 3.599 | 191.009 |
| | | 10 | 1.123 | 0.844 | 1.059 | 3.233 | 167.366 | 1.011 | 0.810 | 1.004 | 2.133 | 24.604 | 1.245 | 0.843 | 1.065 | 1.946 | 31.959 |
| | | 12 | 1.112 | 0.838 | 1.054 | 2.815 | 116.283 | 1.019 | 0.795 | 1.007 | 2.321 | 40.325 | 1.245 | 0.904 | 1.116 | 3.641 | 223.644 |
| | | Avg | 1.104 | 0.837 | 1.050 | 3.040 | 149.767 | 1.011 | 0.797 | 1.004 | 2.303 | 39.851 | 1.179 | 0.871 | 1.086 | 3.246 | 179.336 |
| | TaTS (ours) | 6 | 0.905 | 0.749 | 0.951 | 2.025 | 36.745 | 0.893 | 0.748 | 0.945 | 1.806 | 13.948 | 0.912 | 0.758 | 0.955 | 2.598 | 82.597 |
| | | 8 | 0.926 | 0.756 | 0.962 | 2.018 | 37.680 | 0.937 | 0.771 | 0.968 | 1.893 | 16.622 | 0.917 | 0.751 | 0.957 | 1.939 | 26.990 |
| | | 10 | 0.943 | 0.764 | 0.971 | 1.989 | 34.385 | 0.924 | 0.751 | 0.960 | 1.756 | 14.796 | 0.947 | 0.759 | 0.972 | 1.597 | 10.958 |
| | | 12 | 0.950 | 0.766 | 0.973 | 1.938 | 30.964 | 0.952 | 0.770 | 0.974 | 1.921 | 21.544 | 1.005 | 0.821 | 1.003 | 2.686 | 68.310 |
| | | Avg | 0.931 | 0.759 | 0.964 | 1.992 | 34.944 | 0.926 | 0.760 | 0.962 | 1.844 | 16.727 | 0.945 | 0.772 | 0.972 | 2.205 | 47.214 |
| Economy | Uni-modal | 6 | 0.056 | 0.189 | 0.236 | 0.065 | 0.006 | 0.042 | 0.168 | 0.203 | 0.058 | 0.005 | 0.021 | 0.115 | 0.146 | 0.039 | 0.002 |
| | | 8 | 0.056 | 0.191 | 0.237 | 0.066 | 0.006 | 0.039 | 0.162 | 0.197 | 0.056 | 0.005 | 0.028 | 0.132 | 0.168 | 0.045 | 0.003 |
| | | 10 | 0.047 | 0.173 | 0.216 | 0.060 | 0.006 | 0.036 | 0.153 | 0.188 | 0.053 | 0.004 | 0.026 | 0.134 | 0.162 | 0.045 | 0.003 |
| | | 12 | 0.075 | 0.216 | 0.272 | 0.074 | 0.009 | 0.053 | 0.183 | 0.225 | 0.063 | 0.006 | 0.027 | 0.133 | 0.163 | 0.046 | 0.003 |
| | | Avg | 0.058 | 0.192 | 0.240 | 0.066 | 0.007 | 0.042 | 0.166 | 0.203 | 0.058 | 0.005 | 0.025 | 0.129 | 0.160 | 0.044 | 0.003 |
| | MM-TSFLib | 6 | 0.059 | 0.200 | 0.243 | 0.070 | 0.007 | 0.035 | 0.153 | 0.188 | 0.053 | 0.004 | 0.018 | 0.104 | 0.133 | 0.036 | 0.002 |
| | | 8 | 0.062 | 0.194 | 0.248 | 0.068 | 0.007 | 0.043 | 0.170 | 0.206 | 0.059 | 0.005 | 0.031 | 0.138 | 0.177 | 0.047 | 0.003 |
| | | 10 | 0.064 | 0.195 | 0.253 | 0.068 | 0.008 | 0.040 | 0.160 | 0.200 | 0.056 | 0.005 | 0.026 | 0.133 | 0.160 | 0.045 | 0.003 |
| | | 12 | 0.049 | 0.180 | 0.221 | 0.062 | 0.006 | 0.024 | 0.129 | 0.157 | 0.045 | 0.003 | 0.029 | 0.140 | 0.170 | 0.048 | 0.003 |
| | | Avg | 0.058 | 0.192 | 0.241 | 0.067 | 0.007 | 0.035 | 0.153 | 0.188 | 0.053 | 0.004 | 0.026 | 0.129 | 0.160 | 0.044 | 0.003 |
| | TaTS (ours) | 6 | 0.020 | 0.115 | 0.141 | 0.040 | 0.002 | 0.012 | 0.093 | 0.111 | 0.033 | 0.002 | 0.009 | 0.080 | 0.096 | 0.028 | 0.001 |
| | | 8 | 0.020 | 0.115 | 0.142 | 0.040 | 0.002 | 0.014 | 0.099 | 0.120 | 0.034 | 0.002 | 0.009 | 0.079 | 0.096 | 0.028 | 0.001 |
| | | 10 | 0.019 | 0.112 | 0.137 | 0.039 | 0.002 | 0.016 | 0.106 | 0.126 | 0.037 | 0.002 | 0.009 | 0.079 | 0.096 | 0.027 | 0.001 |
| | | 12 | 0.025 | 0.127 | 0.157 | 0.044 | 0.003 | 0.017 | 0.107 | 0.129 | 0.037 | 0.002 | 0.009 | 0.081 | 0.096 | 0.028 | 0.001 |
| | | Avg | 0.021 | 0.117 | 0.144 | 0.041 | 0.002 | 0.015 | 0.101 | 0.121 | 0.035 | 0.002 | 0.009 | 0.080 | 0.096 | 0.028 | 0.001 |

Table 23: Full forecasting results for the Energy, Environment, and Health datasets using DLinear, FEDformer, and FiLM as time series models. Compared to numerical-only unimodal modeling and MM-TSFLib, our TaTS framework seamlessly enhances existing time series models to effectively handle time series with concurrent texts. Avg: the average results across all prediction lengths.

| Models | | | DLinear (2023) | | | | | FEDformer (2022b) | | | | | FiLM (2022a) | | | | |
|---|---|---|---|---|---|---|---|---|---|---|---|---|---|---|---|---|---|---|
| | Method | | MSE | MAE | RMSE | MAPE | MSPE | MSE | MAE | RMSE | MAPE | MSPE | MSE | MAE | RMSE | MAPE | MSPE |
| Energy | Uni-modal | 12 | 0.136 | 0.264 | 0.335 | 0.947 | 10.373 | 0.095 | 0.212 | 0.281 | 0.895 | 9.380 | 0.118 | 0.245 | 0.314 | 1.208 | 25.502 |
| | | 24 | 0.261 | 0.385 | 0.467 | 1.453 | 33.690 | 0.170 | 0.303 | 0.385 | 1.573 | 47.676 | 0.221 | 0.347 | 0.433 | 1.414 | 29.039 |
| | | 36 | 0.335 | 0.437 | 0.528 | 1.780 | 56.178 | 0.249 | 0.373 | 0.475 | 2.165 | 133.305 | 0.335 | 0.434 | 0.533 | 1.852 | 60.033 |
| | | 48 | 0.431 | 0.498 | 0.603 | 2.294 | 83.886 | 0.445 | 0.514 | 0.643 | 4.141 | 340.071 | 0.437 | 0.512 | 0.620 | 2.414 | 89.205 |
| | | Avg | 0.291 | 0.396 | 0.483 | 1.619 | 46.032 | 0.240 | 0.351 | 0.446 | 2.194 | 132.608 | 0.278 | 0.385 | 0.475 | 1.722 | 50.945 |
| | MM-TSFLib | 12 | 0.133 | 0.262 | 0.332 | 0.950 | 10.725 | 0.098 | 0.227 | 0.289 | 1.083 | 15.260 | 0.119 | 0.247 | 0.315 | 1.202 | 24.419 |
| | | 24 | 0.256 | 0.380 | 0.461 | 1.428 | 32.596 | 0.172 | 0.300 | 0.436 | 1.425 | 47.727 | 0.224 | 0.349 | 0.436 | 1.415 | 28.417 |
| | | 36 | 0.340 | 0.442 | 0.533 | 1.803 | 56.982 | 0.252 | 0.376 | 0.477 | 2.120 | 124.465 | 0.332 | 0.431 | 0.531 | 1.839 | 59.454 |
| | | 48 | 0.428 | 0.496 | 0.601 | 2.287 | 83.443 | 0.430 | 0.511 | 0.611 | 2.775 | 146.406 | 0.440 | 0.514 | 0.623 | 2.416 | 89.684 |
| | | Avg | 0.289 | 0.395 | 0.482 | 1.617 | 45.936 | 0.238 | 0.354 | 0.441 | 1.851 | 83.465 | 0.279 | 0.385 | 0.476 | 1.718 | 50.493 |
| | TaTS (ours) | 12 | 0.132 | 0.260 | 0.330 | 0.954 | 11.225 | 0.090 | 0.210 | 0.275 | 0.880 | 12.877 | 0.118 | 0.245 | 0.314 | 1.202 | 25.174 |
| | | 24 | 0.236 | 0.359 | 0.441 | 1.359 | 29.647 | 0.172 | 0.305 | 0.387 | 1.375 | 44.042 | 0.222 | 0.347 | 0.434 | 1.410 | 28.640 |
| | | 36 | 0.340 | 0.442 | 0.533 | 1.805 | 57.123 | 0.250 | 0.372 | 0.472 | 2.412 | 152.144 | 0.311 | 0.414 | 0.514 | 1.773 | 55.535 |
| | | 48 | 0.425 | 0.493 | 0.597 | 2.285 | 84.710 | 0.435 | 0.531 | 0.631 | 3.501 | 222.058 | 0.434 | 0.510 | 0.618 | 2.399 | 88.602 |
| | | Avg | 0.283 | 0.388 | 0.475 | 1.601 | 45.676 | 0.237 | 0.355 | 0.441 | 2.042 | 107.780 | 0.271 | 0.379 | 0.470 | 1.696 | 49.488 |
| Environment | Uni-modal | 48 | 0.478 | 0.531 | 0.646 | 1.791 | 119.315 | 0.505 | 0.543 | 0.671 | 1.932 | 128.444 | 0.494 | 0.502 | 0.644 | 2.323 | 241.494 |
| | | 96 | 0.562 | 0.608 | 0.724 | 1.539 | 68.485 | 0.465 | 0.524 | 0.657 | 2.440 | 237.225 | 0.581 | 0.546 | 0.707 | 2.639 | 314.062 |
| | | 192 | 0.592 | 0.608 | 0.748 | 1.938 | 132.178 | 0.510 | 0.556 | 0.702 | 2.656 | 297.619 | 0.612 | 0.559 | 0.759 | 3.081 | 461.654 |
| | | 336 | 0.600 | 0.618 | 0.768 | 1.938 | 134.420 | 0.531 | 0.573 | 0.725 | 2.528 | 300.421 | 0.621 | 0.566 | 0.779 | 2.989 | 440.819 |
| | | Avg | 0.558 | 0.591 | 0.722 | 1.801 | 113.600 | 0.503 | 0.549 | 0.689 | 2.389 | 240.927 | 0.577 | 0.543 | 0.722 | 2.758 | 364.507 |
| | MM-TSFLib | 48 | 0.414 | 0.474 | 0.598 | 1.888 | 142.797 | 0.413 | 0.468 | 0.599 | 1.867 | 128.047 | 0.469 | 0.484 | 0.652 | 2.046 | 177.399 |
| | | 96 | 0.420 | 0.489 | 0.615 | 1.743 | 112.990 | 0.413 | 0.479 | 0.613 | 1.998 | 138.883 | 0.476 | 0.493 | 0.662 | 1.987 | 173.914 |
| | | 192 | 0.439 | 0.521 | 0.650 | 1.659 | 92.080 | 0.423 | 0.489 | 0.635 | 2.205 | 197.315 | 0.477 | 0.486 | 0.674 | 2.135 | 218.811 |
| | | 336 | 0.444 | 0.526 | 0.664 | 1.680 | 95.029 | 0.445 | 0.507 | 0.665 | 1.850 | 125.600 | 0.490 | 0.497 | 0.696 | 2.233 | 243.544 |
| | | Avg | 0.429 | 0.502 | 0.632 | 1.742 | 110.724 | 0.423 | 0.486 | 0.628 | 1.980 | 147.461 | 0.478 | 0.490 | 0.671 | 2.100 | 203.417 |
| | TaTS (ours) | 48 | 0.272 | 0.385 | 0.486 | 1.083 | 17.610 | 0.272 | 0.369 | 0.484 | 1.172 | 21.826 | 0.269 | 0.373 | 0.481 | 1.124 | 19.765 |
| | | 96 | 0.301 | 0.427 | 0.528 | 1.034 | 13.731 | 0.271 | 0.371 | 0.495 | 1.143 | 22.305 | 0.271 | 0.377 | 0.499 | 1.180 | 21.986 |
| | | 192 | 0.306 | 0.443 | 0.544 | 1.007 | 11.494 | 0.277 | 0.379 | 0.501 | 1.145 | 22.438 | 0.271 | 0.367 | 0.507 | 1.202 | 22.318 |
| | | 336 | 0.313 | 0.456 | 0.557 | 0.996 | 10.408 | 0.278 | 0.393 | 0.524 | 1.137 | 18.379 | 0.267 | 0.366 | 0.514 | 1.210 | 22.394 |
| | | Avg | 0.298 | 0.428 | 0.529 | 1.030 | 13.311 | 0.275 | 0.378 | 0.501 | 1.149 | 21.237 | 0.272 | 0.371 | 0.500 | 1.179 | 21.616 |
| Health | Uni-modal | 12 | 1.595 | 0.808 | 1.113 | 2.599 | 252.012 | 1.051 | 0.756 | 0.975 | 4.069 | 371.454 | 1.900 | 1.008 | 1.278 | 6.208 | 1054.818 |
| | | 24 | 1.778 | 0.835 | 1.170 | 2.742 | 312.129 | 1.493 | 0.933 | 1.176 | 4.818 | 434.785 | 1.946 | 0.973 | 1.287 | 4.194 | 322.873 |
| | | 36 | 1.759 | 0.850 | 1.241 | 2.738 | 263.854 | 1.661 | 0.980 | 1.252 | 5.182 | 426.708 | 2.029 | 1.013 | 1.352 | 4.495 | 346.381 |
| | | 48 | 1.818 | 0.899 | 1.299 | 2.964 | 278.936 | 1.737 | 0.969 | 1.285 | 4.775 | 392.270 | 2.054 | 1.026 | 1.379 | 4.356 | 305.131 |
| | | Avg | 1.737 | 0.848 | 1.206 | 2.761 | 276.733 | 1.486 | 0.909 | 1.172 | 4.711 | 406.304 | 1.982 | 1.005 | 1.324 | 4.813 | 507.301 |
| | MM-TSFLib | 12 | 1.403 | 0.763 | 1.061 | 2.065 | 64.534 | 0.978 | 0.685 | 0.930 | 3.371 | 273.283 | 1.406 | 0.915 | 1.141 | 4.787 | 421.324 |
| | | 24 | 1.544 | 0.782 | 1.114 | 2.167 | 68.258 | 1.272 | 0.804 | 1.073 | 3.538 | 212.521 | 1.730 | 0.975 | 1.257 | 4.258 | 251.992 |
| | | 36 | 1.600 | 0.824 | 1.185 | 2.264 | 56.729 | 1.344 | 0.828 | 1.121 | 3.768 | 222.549 | 1.769 | 0.944 | 1.267 | 3.785 | 205.906 |
| | | 48 | 1.617 | 0.830 | 1.222 | 2.368 | 87.198 | 1.416 | 0.852 | 1.163 | 3.913 | 243.392 | 1.793 | 0.960 | 1.286 | 3.626 | 178.720 |
| | | Avg | 1.541 | 0.800 | 1.145 | 2.216 | 69.180 | 1.252 | 0.792 | 1.072 | 3.647 | 237.936 | 1.675 | 0.949 | 1.238 | 4.114 | 264.486 |
| | TaTS (ours) | 12 | 1.273 | 0.742 | 1.027 | 2.702 | 159.975 | 0.966 | 0.671 | 0.925 | 3.309 | 294.367 | 1.211 | 0.794 | 1.035 | 3.570 | 236.037 |
| | | 24 | 1.421 | 0.788 | 1.090 | 3.221 | 254.155 | 1.264 | 0.823 | 1.080 | 4.068 | 289.731 | 1.414 | 0.827 | 1.107 | 3.160 | 130.456 |
| | | 36 | 1.449 | 0.787 | 1.133 | 2.812 | 148.843 | 1.320 | 0.801 | 1.110 | 3.885 | 280.248 | 1.497 | 0.851 | 1.165 | 3.274 | 133.041 |
| | | 48 | 1.505 | 0.832 | 1.183 | 2.990 | 186.495 | 1.426 | 0.868 | 1.164 | 4.140 | 264.871 | 1.562 | 0.878 | 1.209 | 3.225 | 121.440 |
| | | Avg | 1.412 | 0.787 | 1.108 | 2.931 | 187.367 | 1.244 | 0.791 | 1.070 | 3.851 | 282.304 | 1.421 | 0.838 | 1.129 | 3.307 | 155.243 |

Table 24: Full forecasting results for the Security, Social Good, and Traffic datasets using DLinear, FEDformer, and FiLM as time series models. Compared to numerical-only unimodal modeling and MM-TSFLib, our TaTS framework seamlessly enhances existing time series models to effectively handle time series with concurrent texts. Avg: the average results across all prediction lengths.

| Models | | | DLinear (2023b) | | | | | FEDformer (2022b) | | | | | FiLM (2022a) | | | | |
|---|---|---|---|---|---|---|---|---|---|---|---|---|---|---|---|---|---|---|
| Method | | | MSE | MAE | RMSE | MAPE | MSPE | MSE | MAE | RMSE | MAPE | MSPE | MSE | MAE | RMSE | MAPE | MSPE |
| Security | Uni-modal | 6 | 107.046 | 4.531 | 10.346 | 3.489 | 324.650 | 112.106 | 4.830 | 10.588 | 4.456 | 571.690 | 115.130 | 5.587 | 10.730 | 5.029 | 637.394 |
| | | 8 | 107.700 | 4.649 | 10.378 | 2.992 | 295.719 | 113.897 | 5.200 | 10.672 | 3.223 | 355.929 | 112.097 | 5.034 | 10.588 | 3.228 | 336.503 |
| | | 10 | 109.360 | 4.759 | 10.458 | 2.449 | 216.866 | 117.484 | 5.467 | 10.839 | 2.563 | 216.459 | 112.267 | 4.901 | 10.596 | 2.310 | 188.201 |
| | | 12 | 112.343 | 4.904 | 10.599 | 1.980 | 149.507 | 114.452 | 5.135 | 10.698 | 2.697 | 329.415 | 122.711 | 6.425 | 11.077 | 3.554 | 437.883 |
| | | Avg | 109.112 | 4.711 | 10.445 | 2.728 | 246.685 | 114.485 | 5.158 | 10.699 | 3.235 | 368.373 | 115.551 | 5.487 | 10.748 | 3.530 | 399.995 |
| | MM-TSFLib | 6 | 106.121 | 4.545 | 10.301 | 3.788 | 386.715 | 109.854 | 4.690 | 10.481 | 4.252 | 541.638 | 105.809 | 4.755 | 10.286 | 4.286 | 525.309 |
| | | 8 | 107.445 | 4.670 | 10.366 | 3.101 | 319.555 | 114.833 | 5.233 | 10.716 | 3.119 | 330.374 | 108.045 | 4.638 | 10.394 | 2.727 | 249.934 |
| | | 10 | 108.713 | 4.758 | 10.427 | 2.613 | 251.113 | 115.294 | 5.294 | 10.736 | 2.563 | 233.233 | 111.391 | 4.843 | 10.554 | 1.953 | 137.888 |
| | | 12 | 109.848 | 4.876 | 10.481 | 2.373 | 227.133 | 115.001 | 5.211 | 10.724 | 2.558 | 291.461 | 111.541 | 5.354 | 10.561 | 3.023 | 363.599 |
| | | Avg | 108.032 | 4.712 | 10.394 | 2.969 | 296.129 | 113.737 | 5.107 | 10.664 | 3.123 | 349.177 | 109.197 | 4.897 | 10.449 | 2.997 | 319.183 |
| | TaTS (ours) | 6 | 106.015 | 4.516 | 10.296 | 3.850 | 404.799 | 106.015 | 4.532 | 10.296 | 4.542 | 597.429 | 105.482 | 4.511 | 10.270 | 4.024 | 451.366 |
| | | 8 | 107.477 | 4.623 | 10.367 | 3.050 | 312.447 | 107.678 | 4.618 | 10.377 | 3.303 | 371.986 | 107.657 | 4.792 | 10.376 | 3.150 | 310.920 |
| | | 10 | 108.505 | 4.728 | 10.417 | 2.632 | 260.096 | 107.301 | 4.885 | 10.359 | 3.138 | 375.009 | 109.886 | 4.771 | 10.483 | 2.338 | 193.865 |
| | | 12 | 109.717 | 4.836 | 10.475 | 2.336 | 224.134 | 108.506 | 4.837 | 10.417 | 2.738 | 331.710 | 108.375 | 4.870 | 10.410 | 2.549 | 267.339 |
| | | Avg | 107.928 | 4.676 | 10.389 | 2.967 | 300.369 | 107.375 | 4.718 | 10.362 | 3.430 | 419.034 | 107.850 | 4.736 | 10.385 | 3.015 | 305.873 |
| Social Good | Uni-modal | 6 | 1.018 | 0.627 | 0.859 | 1.503 | 19.120 | 0.821 | 0.386 | 0.649 | 1.086 | 28.066 | 1.123 | 0.604 | 0.878 | 2.432 | 149.647 |
| | | 8 | 1.137 | 0.702 | 0.929 | 1.607 | 19.248 | 0.929 | 0.451 | 0.726 | 1.061 | 17.701 | 1.116 | 0.542 | 0.841 | 2.275 | 128.188 |
| | | 10 | 1.210 | 0.755 | 0.972 | 1.732 | 21.287 | 1.059 | 0.478 | 0.784 | 1.469 | 52.985 | 1.154 | 0.601 | 0.882 | 1.823 | 42.492 |
| | | 12 | 1.238 | 0.763 | 0.982 | 1.705 | 20.522 | 1.105 | 0.588 | 0.846 | 1.876 | 76.675 | 1.653 | 0.869 | 1.125 | 2.998 | 140.138 |
| | | Avg | 1.151 | 0.712 | 0.935 | 1.637 | 20.044 | 0.979 | 0.476 | 0.751 | 1.373 | 43.857 | 1.261 | 0.654 | 0.931 | 2.382 | 115.116 |
| | MM-TSFLib | 6 | 0.946 | 0.583 | 0.810 | 1.481 | 21.957 | 0.816 | 0.395 | 0.667 | 1.182 | 33.446 | 1.070 | 0.529 | 0.824 | 2.269 | 159.103 |
| | | 8 | 1.095 | 0.678 | 0.904 | 1.596 | 20.076 | 0.910 | 0.438 | 0.717 | 1.010 | 15.845 | 1.120 | 0.584 | 0.869 | 2.138 | 83.150 |
| | | 10 | 1.128 | 0.709 | 0.924 | 1.713 | 22.953 | 1.032 | 0.456 | 0.757 | 1.395 | 53.857 | 1.128 | 0.539 | 0.839 | 2.235 | 120.886 |
| | | 12 | 1.164 | 0.724 | 0.940 | 1.655 | 20.260 | 1.091 | 0.560 | 0.838 | 1.737 | 71.467 | 1.624 | 0.852 | 1.111 | 3.005 | 149.026 |
| | | Avg | 1.083 | 0.673 | 0.894 | 1.611 | 21.312 | 0.962 | 0.462 | 0.745 | 1.331 | 43.654 | 1.236 | 0.626 | 0.911 | 2.412 | 128.041 |
| | TaTS (ours) | 6 | 0.859 | 0.516 | 0.740 | 1.453 | 29.534 | 0.746 | 0.365 | 0.620 | 0.983 | 23.097 | 0.992 | 0.588 | 0.834 | 1.623 | 30.115 |
| | | 8 | 0.963 | 0.592 | 0.812 | 1.582 | 29.073 | 0.880 | 0.448 | 0.701 | 1.221 | 34.945 | 1.001 | 0.550 | 0.806 | 1.609 | 28.399 |
| | | 10 | 1.118 | 0.703 | 0.917 | 1.711 | 23.312 | 0.944 | 0.453 | 0.720 | 1.138 | 28.381 | 1.065 | 0.583 | 0.837 | 1.447 | 17.661 |
| | | 12 | 1.085 | 0.676 | 0.888 | 1.657 | 24.350 | 0.983 | 0.453 | 0.718 | 1.315 | 51.258 | 1.358 | 0.782 | 1.025 | 1.920 | 27.141 |
| | | Avg | 1.006 | 0.622 | 0.839 | 1.601 | 26.567 | 0.888 | 0.430 | 0.690 | 1.164 | 34.420 | 1.104 | 0.626 | 0.876 | 1.650 | 25.829 |
| Traffic | Uni-modal | 6 | 0.221 | 0.358 | 0.461 | 0.357 | 0.549 | 0.202 | 0.294 | 0.416 | 0.285 | 0.377 | 0.202 | 0.302 | 0.426 | 0.346 | 0.682 |
| | | 8 | 0.220 | 0.357 | 0.458 | 0.355 | 0.552 | 0.184 | 0.234 | 0.364 | 0.259 | 0.461 | 0.203 | 0.332 | 0.436 | 0.341 | 0.564 |
| | | 10 | 0.217 | 0.347 | 0.451 | 0.358 | 0.601 | 0.198 | 0.251 | 0.386 | 0.271 | 0.474 | 0.203 | 0.340 | 0.438 | 0.323 | 0.447 |
| | | 12 | 0.261 | 0.374 | 0.499 | 0.395 | 0.663 | 0.237 | 0.275 | 0.410 | 0.315 | 0.600 | 0.250 | 0.280 | 0.426 | 0.347 | 0.719 |
| | | Avg | 0.230 | 0.359 | 0.467 | 0.366 | 0.591 | 0.205 | 0.264 | 0.394 | 0.283 | 0.478 | 0.215 | 0.314 | 0.431 | 0.339 | 0.603 |
| | MM-TSFLib | 6 | 0.204 | 0.332 | 0.439 | 0.321 | 0.465 | 0.180 | 0.245 | 0.372 | 0.258 | 0.408 | 0.197 | 0.294 | 0.419 | 0.339 | 0.664 |
| | | 8 | 0.202 | 0.335 | 0.435 | 0.320 | 0.448 | 0.178 | 0.225 | 0.352 | 0.253 | 0.458 | 0.195 | 0.322 | 0.427 | 0.332 | 0.540 |
| | | 10 | 0.205 | 0.325 | 0.435 | 0.320 | 0.478 | 0.184 | 0.234 | 0.364 | 0.258 | 0.446 | 0.195 | 0.328 | 0.428 | 0.312 | 0.424 |
| | | 12 | 0.225 | 0.329 | 0.451 | 0.351 | 0.602 | 0.228 | 0.247 | 0.388 | 0.292 | 0.570 | 0.240 | 0.258 | 0.403 | 0.315 | 0.631 |
| | | Avg | 0.209 | 0.330 | 0.440 | 0.328 | 0.498 | 0.193 | 0.238 | 0.369 | 0.265 | 0.471 | 0.207 | 0.300 | 0.419 | 0.325 | 0.565 |
| | TaTS (ours) | 6 | 0.184 | 0.312 | 0.412 | 0.301 | 0.435 | 0.159 | 0.208 | 0.329 | 0.228 | 0.355 | 0.157 | 0.228 | 0.351 | 0.254 | 0.426 |
| | | 8 | 0.185 | 0.302 | 0.409 | 0.308 | 0.500 | 0.159 | 0.209 | 0.329 | 0.230 | 0.368 | 0.162 | 0.260 | 0.370 | 0.266 | 0.401 |
| | | 10 | 0.184 | 0.297 | 0.403 | 0.309 | 0.524 | 0.160 | 0.206 | 0.323 | 0.229 | 0.370 | 0.169 | 0.269 | 0.380 | 0.262 | 0.358 |
| | | 12 | 0.199 | 0.291 | 0.413 | 0.309 | 0.489 | 0.213 | 0.224 | 0.368 | 0.269 | 0.518 | 0.215 | 0.236 | 0.383 | 0.298 | 0.612 |
| | | Avg | 0.188 | 0.300 | 0.409 | 0.307 | 0.487 | 0.173 | 0.212 | 0.337 | 0.239 | 0.403 | 0.176 | 0.248 | 0.371 | 0.270 | 0.449 |

Table 25: Full forecasting results for the Agriculture, Climate, and Economy datasets using Autoformer, Informer, and Transformer as time series models. Compared to numerical-only unimodal modeling and MM-TSFLib, our TaTS framework seamlessly enhances existing time series models to effectively handle time series with concurrent texts. Avg: the average results across all prediction lengths.

| | Models | | Autoformer (2021) | | | | | Informer (2021) | | | | | Transformer (2017) | | | | |
|---|---|---|---|---|---|---|---|---|---|---|---|---|---|---|---|---|---|
| | Method | | MSE | MAE | RMSE | MAPE | MSPE | MSE | MAE | RMSE | MAPE | MSPE | MSE | MAE | RMSE | MAPE | MSPE |
| Agriculture | Uni-modal | 6 | 0.109 | 0.255 | 0.329 | 0.116 | 0.022 | 0.451 | 0.557 | 0.626 | 0.240 | 0.071 | 0.229 | 0.326 | 0.420 | 0.133 | 0.032 |
| | | 8 | 0.135 | 0.278 | 0.366 | 0.126 | 0.026 | 0.569 | 0.633 | 0.702 | 0.270 | 0.087 | 0.328 | 0.432 | 0.502 | 0.177 | 0.045 |
| | | 10 | 0.173 | 0.311 | 0.413 | 0.138 | 0.031 | 0.618 | 0.633 | 0.724 | 0.263 | 0.088 | 0.358 | 0.422 | 0.510 | 0.166 | 0.045 |
| | | 12 | 0.213 | 0.345 | 0.457 | 0.149 | 0.036 | 0.756 | 0.698 | 0.789 | 0.284 | 0.101 | 0.500 | 0.555 | 0.633 | 0.224 | 0.064 |
| | | Avg | 0.158 | 0.297 | 0.391 | 0.132 | 0.029 | 0.599 | 0.630 | 0.710 | 0.264 | 0.087 | 0.354 | 0.434 | 0.516 | 0.175 | 0.046 |
| | MM-TSFLib | 6 | 0.095 | 0.221 | 0.306 | 0.100 | 0.018 | 0.218 | 0.352 | 0.422 | 0.148 | 0.032 | 0.197 | 0.319 | 0.392 | 0.133 | 0.029 |
| | | 8 | 0.143 | 0.278 | 0.372 | 0.122 | 0.025 | 0.306 | 0.429 | 0.496 | 0.179 | 0.044 | 0.205 | 0.303 | 0.384 | 0.121 | 0.027 |
| | | 10 | 0.176 | 0.311 | 0.416 | 0.137 | 0.031 | 0.301 | 0.398 | 0.480 | 0.160 | 0.040 | 0.215 | 0.332 | 0.411 | 0.135 | 0.030 |
| | | 12 | 0.217 | 0.341 | 0.453 | 0.144 | 0.034 | 0.428 | 0.479 | 0.579 | 0.190 | 0.054 | 0.378 | 0.453 | 0.531 | 0.179 | 0.048 |
| | | Avg | 0.158 | 0.288 | 0.387 | 0.126 | 0.027 | 0.313 | 0.414 | 0.494 | 0.169 | 0.042 | 0.249 | 0.352 | 0.429 | 0.142 | 0.034 |
| | TaTS (ours) | 6 | 0.076 | 0.205 | 0.269 | 0.092 | 0.014 | 0.162 | 0.265 | 0.342 | 0.107 | 0.022 | 0.150 | 0.348 | 0.387 | 0.166 | 0.034 |
| | | 8 | 0.101 | 0.234 | 0.315 | 0.105 | 0.019 | 0.213 | 0.311 | 0.392 | 0.124 | 0.028 | 0.192 | 0.298 | 0.374 | 0.120 | 0.025 |
| | | 10 | 0.138 | 0.299 | 0.371 | 0.134 | 0.026 | 0.296 | 0.388 | 0.465 | 0.154 | 0.038 | 0.209 | 0.296 | 0.384 | 0.114 | 0.026 |
| | | 12 | 0.186 | 0.324 | 0.425 | 0.141 | 0.030 | 0.349 | 0.426 | 0.506 | 0.167 | 0.044 | 0.213 | 0.310 | 0.401 | 0.121 | 0.025 |
| | | Avg | 0.125 | 0.266 | 0.345 | 0.118 | 0.022 | 0.255 | 0.348 | 0.426 | 0.138 | 0.033 | 0.191 | 0.313 | 0.387 | 0.130 | 0.028 |
| Climate | Uni-modal | 6 | 1.116 | 0.861 | 1.056 | 4.456 | 519.418 | 1.084 | 0.829 | 1.041 | 1.617 | 15.451 | 1.032 | 0.817 | 1.016 | 1.656 | 13.521 |
| | | 8 | 1.117 | 0.856 | 1.057 | 3.329 | 177.957 | 1.087 | 0.829 | 1.043 | 2.464 | 66.852 | 1.135 | 0.856 | 1.065 | 1.868 | 22.687 |
| | | 10 | 1.170 | 0.886 | 1.081 | 3.059 | 106.074 | 1.117 | 0.849 | 1.056 | 2.071 | 49.757 | 1.103 | 0.844 | 1.049 | 2.355 | 81.661 |
| | | 12 | 1.123 | 0.855 | 1.059 | 3.364 | 125.779 | 1.150 | 0.859 | 1.071 | 2.557 | 94.732 | 1.100 | 0.841 | 1.048 | 1.663 | 24.078 |
| | | Avg | 1.131 | 0.865 | 1.063 | 3.552 | 232.307 | 1.110 | 0.841 | 1.053 | 2.177 | 56.698 | 1.092 | 0.839 | 1.044 | 1.885 | 35.487 |
| | MM-TSFLib | 6 | 1.021 | 0.817 | 1.010 | 2.871 | 82.634 | 1.019 | 0.811 | 1.009 | 2.844 | 134.946 | 0.962 | 0.771 | 0.981 | 2.030 | 23.607 |
| | | 8 | 1.039 | 0.819 | 1.018 | 3.018 | 99.283 | 0.970 | 0.772 | 0.985 | 2.210 | 43.152 | 0.991 | 0.783 | 0.995 | 1.976 | 22.312 |
| | | 10 | 1.074 | 0.838 | 1.035 | 2.783 | 92.522 | 1.006 | 0.794 | 1.002 | 2.311 | 67.013 | 1.026 | 0.790 | 1.010 | 2.755 | 87.765 |
| | | 12 | 1.078 | 0.836 | 1.038 | 2.731 | 78.154 | 1.009 | 0.791 | 0.999 | 2.499 | 86.993 | 1.011 | 0.789 | 1.000 | 1.895 | 27.637 |
| | | Avg | 1.053 | 0.827 | 1.025 | 2.851 | 88.148 | 1.001 | 0.792 | 0.999 | 2.466 | 83.026 | 0.998 | 0.783 | 0.996 | 2.164 | 40.330 |
| | TaTS (ours) | 6 | 0.880 | 0.741 | 0.938 | 2.222 | 37.958 | 0.881 | 0.741 | 0.939 | 2.217 | 57.260 | 0.859 | 0.728 | 0.926 | 1.970 | 35.573 |
| | | 8 | 0.937 | 0.768 | 0.968 | 2.136 | 36.109 | 0.909 | 0.749 | 0.953 | 1.981 | 29.665 | 0.913 | 0.761 | 0.955 | 2.524 | 73.564 |
| | | 10 | 1.027 | 0.817 | 1.011 | 2.260 | 31.311 | 0.973 | 0.771 | 0.983 | 2.020 | 27.051 | 0.976 | 0.767 | 0.985 | 2.179 | 31.989 |
| | | 12 | 1.075 | 0.831 | 1.033 | 2.317 | 35.133 | 0.958 | 0.764 | 0.974 | 1.727 | 15.480 | 0.930 | 0.757 | 0.960 | 1.871 | 23.016 |
| | | Avg | 0.980 | 0.789 | 0.987 | 2.234 | 35.128 | 0.930 | 0.756 | 0.962 | 1.986 | 32.364 | 0.920 | 0.753 | 0.957 | 2.136 | 41.035 |
| Economy | Uni-modal | 6 | 0.083 | 0.222 | 0.288 | 0.077 | 0.010 | 0.877 | 0.896 | 0.930 | 0.308 | 0.102 | 0.276 | 0.444 | 0.517 | 0.150 | 0.031 |
| | | 8 | 0.069 | 0.210 | 0.263 | 0.073 | 0.008 | 1.606 | 1.238 | 1.260 | 0.425 | 0.187 | 0.676 | 0.797 | 0.816 | 0.274 | 0.078 |
| | | 10 | 0.070 | 0.209 | 0.261 | 0.072 | 0.008 | 1.409 | 1.153 | 1.179 | 0.395 | 0.162 | 0.601 | 0.750 | 0.768 | 0.257 | 0.069 |
| | | 12 | 0.062 | 0.186 | 0.245 | 0.064 | 0.007 | 1.407 | 1.153 | 1.183 | 0.396 | 0.163 | 0.781 | 0.854 | 0.879 | 0.293 | 0.091 |
| | | Avg | 0.071 | 0.207 | 0.264 | 0.071 | 0.008 | 1.325 | 1.110 | 1.138 | 0.381 | 0.153 | 0.584 | 0.711 | 0.745 | 0.243 | 0.067 |
| | MM-TSFLib | 6 | 0.049 | 0.173 | 0.221 | 0.060 | 0.006 | 0.293 | 0.490 | 0.531 | 0.167 | 0.033 | 0.110 | 0.285 | 0.322 | 0.096 | 0.012 |
| | | 8 | 0.045 | 0.171 | 0.211 | 0.060 | 0.005 | 0.445 | 0.643 | 0.660 | 0.220 | 0.051 | 0.157 | 0.359 | 0.389 | 0.122 | 0.018 |
| | | 10 | 0.071 | 0.214 | 0.267 | 0.075 | 0.009 | 0.471 | 0.664 | 0.680 | 0.227 | 0.054 | 0.240 | 0.450 | 0.483 | 0.154 | 0.028 |
| | | 12 | 0.068 | 0.209 | 0.260 | 0.073 | 0.008 | 0.518 | 0.674 | 0.715 | 0.231 | 0.060 | 0.347 | 0.569 | 0.584 | 0.195 | 0.040 |
| | | Avg | 0.058 | 0.192 | 0.240 | 0.067 | 0.007 | 0.432 | 0.618 | 0.646 | 0.211 | 0.049 | 0.213 | 0.416 | 0.445 | 0.142 | 0.025 |
| | TaTS (ours) | 6 | 0.021 | 0.115 | 0.144 | 0.040 | 0.002 | 0.247 | 0.472 | 0.488 | 0.161 | 0.028 | 0.029 | 0.137 | 0.169 | 0.047 | 0.003 |
| | | 8 | 0.023 | 0.119 | 0.150 | 0.041 | 0.003 | 0.166 | 0.369 | 0.396 | 0.125 | 0.018 | 0.039 | 0.162 | 0.192 | 0.055 | 0.004 |
| | | 10 | 0.024 | 0.122 | 0.153 | 0.042 | 0.003 | 0.400 | 0.610 | 0.626 | 0.209 | 0.046 | 0.092 | 0.266 | 0.299 | 0.091 | 0.010 |
| | | 12 | 0.026 | 0.127 | 0.161 | 0.044 | 0.003 | 0.385 | 0.596 | 0.617 | 0.203 | 0.044 | 0.156 | 0.364 | 0.389 | 0.123 | 0.017 |
| | | Avg | 0.024 | 0.121 | 0.152 | 0.042 | 0.003 | 0.299 | 0.512 | 0.532 | 0.174 | 0.034 | 0.079 | 0.232 | 0.262 | 0.079 | 0.009 |

Table 26: Full forecasting results for the Energy, Environment, and Health datasets using Autoformer, Informer, and Transformer as time series models. Compared to numerical-only unimodal modeling and MM-TSFLib, our TaTS framework seamlessly enhances existing time series models to effectively handle time series with concurrent texts. Avg: the average results across all prediction lengths.

| Models | | | Autoformer (2021) | | | | | Informer (2021) | | | | | Transformer (2017) | | | | |
|---|---|---|---|---|---|---|---|---|---|---|---|---|---|---|---|---|---|
| | Method | | MSE | MAE | RMSE | MAPE | MSPE | MSE | MAE | RMSE | MAPE | MSPE | MSE | MAE | RMSE | MAPE | MSPE |
| Energy | Uni-modal | 12 | 0.161 | 0.282 | 0.362 | 1.403 | 24.893 | 0.165 | 0.289 | 0.373 | 1.491 | 35.971 | 0.124 | 0.247 | 0.320 | 1.021 | 8.908 |
| | | 24 | 0.279 | 0.416 | 0.509 | 2.632 | 158.931 | 0.248 | 0.380 | 0.475 | 2.916 | 185.671 | 0.291 | 0.403 | 0.494 | 1.951 | 101.566 |
| | | 36 | 0.364 | 0.464 | 0.564 | 2.384 | 93.988 | 0.368 | 0.492 | 0.585 | 4.022 | 356.146 | 0.329 | 0.435 | 0.542 | 3.964 | 428.707 |
| | | 48 | 0.471 | 0.550 | 0.664 | 3.106 | 149.374 | 0.457 | 0.539 | 0.640 | 5.613 | 589.886 | 0.445 | 0.537 | 0.642 | 3.824 | 285.963 |
| | | Avg | 0.319 | 0.428 | 0.525 | 2.381 | 106.797 | 0.309 | 0.425 | 0.518 | 3.511 | 291.918 | 0.297 | 0.405 | 0.500 | 2.690 | 206.286 |
| | MM-TSFLib | 12 | 0.164 | 0.297 | 0.369 | 1.720 | 78.617 | 0.166 | 0.291 | 0.372 | 1.667 | 36.449 | 0.127 | 0.270 | 0.334 | 1.278 | 11.692 |
| | | 24 | 0.275 | 0.407 | 0.504 | 2.177 | 81.316 | 0.242 | 0.362 | 0.456 | 3.040 | 235.920 | 0.276 | 0.402 | 0.491 | 2.297 | 115.390 |
| | | 36 | 0.362 | 0.464 | 0.561 | 2.176 | 88.219 | 0.342 | 0.467 | 0.566 | 3.221 | 271.106 | 0.316 | 0.439 | 0.530 | 3.504 | 272.971 |
| | | 48 | 0.477 | 0.545 | 0.665 | 3.100 | 192.217 | 0.456 | 0.530 | 0.643 | 5.430 | 602.773 | 0.454 | 0.508 | 0.641 | 5.511 | 748.943 |
| | | Avg | 0.320 | 0.428 | 0.525 | 2.293 | 110.092 | 0.301 | 0.413 | 0.509 | 3.340 | 286.562 | 0.293 | 0.405 | 0.499 | 3.147 | 287.249 |
| | TaTS (ours) | 12 | 0.153 | 0.291 | 0.365 | 1.490 | 33.553 | 0.145 | 0.271 | 0.347 | 1.118 | 11.872 | 0.126 | 0.252 | 0.324 | 1.036 | 12.192 |
| | | 24 | 0.276 | 0.404 | 0.488 | 2.040 | 73.651 | 0.242 | 0.361 | 0.463 | 2.310 | 184.196 | 0.265 | 0.381 | 0.469 | 1.706 | 53.962 |
| | | 36 | 0.360 | 0.467 | 0.555 | 2.167 | 80.434 | 0.301 | 0.415 | 0.514 | 3.658 | 379.160 | 0.301 | 0.419 | 0.515 | 2.804 | 209.452 |
| | | 48 | 0.466 | 0.559 | 0.659 | 3.431 | 178.191 | 0.448 | 0.538 | 0.644 | 4.344 | 378.638 | 0.423 | 0.527 | 0.620 | 5.149 | 416.124 |
| | | Avg | 0.314 | 0.430 | 0.517 | 2.282 | 91.457 | 0.284 | 0.396 | 0.492 | 2.857 | 238.466 | 0.279 | 0.395 | 0.482 | 2.674 | 172.933 |
| Environment | Uni-modal | 48 | 0.503 | 0.544 | 0.670 | 1.953 | 141.941 | 0.414 | 0.481 | 0.602 | 2.190 | 216.476 | 0.412 | 0.483 | 0.600 | 2.190 | 214.543 |
| | | 96 | 0.594 | 0.605 | 0.743 | 2.015 | 135.052 | 0.467 | 0.520 | 0.644 | 2.454 | 290.858 | 0.458 | 0.511 | 0.639 | 2.608 | 340.210 |
| | | 192 | 0.607 | 0.601 | 0.762 | 2.163 | 205.175 | 0.469 | 0.519 | 0.644 | 2.535 | 305.217 | 0.492 | 0.535 | 0.677 | 2.711 | 404.734 |
| | | 336 | 0.690 | 0.643 | 0.823 | 2.624 | 307.413 | 0.487 | 0.529 | 0.693 | 2.726 | 355.026 | 0.479 | 0.517 | 0.688 | 2.824 | 389.196 |
| | | Avg | 0.599 | 0.598 | 0.749 | 2.189 | 197.395 | 0.459 | 0.512 | 0.644 | 2.476 | 291.894 | 0.460 | 0.511 | 0.651 | 2.583 | 337.171 |
| | MM-TSFLib | 48 | 0.442 | 0.492 | 0.626 | 2.008 | 164.960 | 0.419 | 0.473 | 0.604 | 1.997 | 163.772 | 0.420 | 0.471 | 0.606 | 2.027 | 163.407 |
| | | 96 | 0.456 | 0.505 | 0.645 | 1.930 | 154.161 | 0.426 | 0.477 | 0.616 | 2.074 | 173.217 | 0.426 | 0.485 | 0.615 | 2.380 | 252.997 |
| | | 192 | 0.461 | 0.505 | 0.663 | 1.995 | 181.553 | 0.425 | 0.481 | 0.624 | 2.410 | 256.710 | 0.418 | 0.479 | 0.630 | 2.356 | 252.249 |
| | | 336 | 0.447 | 0.506 | 0.666 | 1.994 | 172.427 | 0.427 | 0.489 | 0.651 | 2.283 | 225.380 | 0.435 | 0.487 | 0.657 | 2.551 | 282.514 |
| | | Avg | 0.452 | 0.502 | 0.650 | 1.982 | 168.275 | 0.424 | 0.480 | 0.626 | 2.191 | 204.770 | 0.425 | 0.481 | 0.627 | 2.329 | 237.792 |
| | TaTS (ours) | 48 | 0.274 | 0.374 | 0.486 | 1.159 | 21.042 | 0.272 | 0.377 | 0.483 | 1.103 | 18.922 | 0.268 | 0.378 | 0.481 | 1.074 | 17.152 |
| | | 96 | 0.286 | 0.387 | 0.509 | 1.183 | 21.118 | 0.287 | 0.401 | 0.507 | 1.040 | 15.636 | 0.272 | 0.391 | 0.496 | 1.028 | 15.508 |
| | | 192 | 0.291 | 0.382 | 0.528 | 1.272 | 24.699 | 0.297 | 0.426 | 0.531 | 1.003 | 13.899 | 0.277 | 0.398 | 0.512 | 1.030 | 16.253 |
| | | 336 | 0.288 | 0.403 | 0.534 | 1.122 | 20.057 | 0.286 | 0.419 | 0.533 | 0.989 | 13.846 | 0.287 | 0.420 | 0.534 | 1.006 | 15.226 |
| | | Avg | 0.285 | 0.387 | 0.514 | 1.184 | 21.729 | 0.285 | 0.406 | 0.513 | 1.034 | 15.576 | 0.276 | 0.397 | 0.506 | 1.035 | 16.035 |
| Health | Uni-modal | 12 | 1.389 | 0.895 | 1.121 | 4.898 | 593.771 | 1.173 | 0.743 | 0.990 | 4.065 | 593.314 | 1.143 | 0.741 | 0.998 | 3.233 | 133.066 |
| | | 24 | 2.328 | 1.191 | 1.476 | 6.117 | 859.160 | 1.215 | 0.766 | 1.042 | 3.728 | 389.282 | 1.480 | 0.786 | 1.096 | 2.435 | 59.228 |
| | | 36 | 1.953 | 1.005 | 1.348 | 4.614 | 392.542 | 1.315 | 0.773 | 1.101 | 3.541 | 390.034 | 1.450 | 0.772 | 1.132 | 2.766 | 136.630 |
| | | 48 | 2.179 | 1.064 | 1.410 | 4.865 | 379.673 | 1.408 | 0.809 | 1.137 | 3.445 | 337.635 | 1.439 | 0.806 | 1.137 | 2.834 | 125.016 |
| | | Avg | 1.962 | 1.039 | 1.339 | 5.123 | 556.286 | 1.278 | 0.773 | 1.067 | 3.695 | 427.566 | 1.378 | 0.776 | 1.091 | 2.817 | 113.485 |
| | MM-TSFLib | 12 | 1.331 | 0.864 | 1.112 | 4.427 | 504.600 | 0.997 | 0.673 | 0.928 | 3.024 | 221.350 | 0.953 | 0.686 | 0.920 | 3.309 | 198.713 |
| | | 24 | 1.454 | 0.868 | 1.138 | 3.681 | 181.137 | 1.209 | 0.740 | 1.046 | 3.372 | 257.237 | 1.315 | 0.762 | 1.071 | 2.833 | 137.728 |
| | | 36 | 1.574 | 0.906 | 1.216 | 4.051 | 234.643 | 1.282 | 0.760 | 1.092 | 3.111 | 213.531 | 1.276 | 0.763 | 1.070 | 2.906 | 137.944 |
| | | 48 | 1.616 | 0.911 | 1.235 | 3.884 | 201.678 | 1.371 | 0.785 | 1.137 | 3.146 | 234.342 | 1.330 | 0.780 | 1.109 | 2.769 | 125.761 |
| | | Avg | 1.494 | 0.887 | 1.175 | 4.011 | 280.514 | 1.215 | 0.740 | 1.051 | 3.163 | 231.615 | 1.218 | 0.748 | 1.042 | 2.954 | 150.036 |
| | TaTS (ours) | 12 | 1.320 | 0.863 | 1.105 | 3.769 | 309.199 | 0.949 | 0.658 | 0.910 | 3.707 | 391.895 | 0.909 | 0.655 | 0.862 | 3.251 | 292.588 |
| | | 24 | 1.467 | 0.885 | 1.159 | 3.485 | 152.650 | 1.130 | 0.736 | 1.020 | 3.897 | 352.680 | 1.187 | 0.728 | 1.030 | 3.143 | 192.685 |
| | | 36 | 1.368 | 0.825 | 1.120 | 3.962 | 230.059 | 1.301 | 0.779 | 1.110 | 3.540 | 270.359 | 1.175 | 0.728 | 1.048 | 3.096 | 193.526 |
| | | 48 | 1.483 | 0.871 | 1.180 | 4.108 | 279.227 | 1.354 | 0.833 | 1.143 | 3.962 | 372.090 | 1.297 | 0.762 | 1.113 | 3.074 | 185.199 |
| | | Avg | 1.409 | 0.861 | 1.141 | 3.831 | 242.784 | 1.183 | 0.752 | 1.046 | 3.776 | 346.756 | 1.142 | 0.718 | 1.013 | 3.141 | 216.000 |

Table 27: Full forecasting results for the Security, Social Good, and Traffic datasets using Autoformer, Informer, and Transformer as time series models. Compared to numerical-only unimodal modeling and MM-TSFLib, our TaTS framework seamlessly enhances existing time series models to effectively handle time series with concurrent texts. Avg: the average results across all prediction lengths.

| | Models | | Autoformer (2021) | | | | | Informer (2021) | | | | | Transformer (2017) | | | | |
|---|---|---|---|---|---|---|---|---|---|---|---|---|---|---|---|---|---|
| | Method | | MSE | MAE | RMSE | MAPE | MSPE | MSE | MAE | RMSE | MAPE | MSPE | MSE | MAE | RMSE | MAPE | MSPE |
| Security | Uni-modal | 6 | 115.544 | 5.268 | 10.749 | 2.437 | 185.108 | 133.168 | 6.626 | 11.540 | 1.591 | 19.155 | 126.925 | 6.203 | 11.266 | 2.547 | 120.160 |
| | | 8 | 113.355 | 4.967 | 10.647 | 2.888 | 298.107 | 134.383 | 6.804 | 11.592 | 1.236 | 6.780 | 132.905 | 6.641 | 11.528 | 1.209 | 14.174 |
| | | 10 | 114.407 | 4.984 | 10.696 | 2.164 | 153.069 | 131.382 | 6.631 | 11.462 | 1.483 | 26.411 | 128.820 | 6.443 | 11.350 | 1.539 | 32.792 |
| | | 12 | 117.801 | 5.253 | 10.854 | 2.692 | 326.651 | 128.201 | 6.430 | 11.323 | 1.408 | 26.498 | 136.753 | 7.039 | 11.694 | 1.168 | 4.854 |
| | | Avg | 115.277 | 5.118 | 10.736 | 2.545 | 240.734 | 131.784 | 6.623 | 11.479 | 1.429 | 19.711 | 131.351 | 6.582 | 11.460 | 1.616 | 42.995 |
| | MM-TSFLib | 6 | 108.841 | 4.861 | 10.433 | 3.004 | 246.068 | 127.216 | 6.197 | 11.279 | 2.326 | 99.099 | 127.277 | 6.233 | 11.282 | 2.592 | 135.752 |
| | | 8 | 113.693 | 5.031 | 10.663 | 2.799 | 265.988 | 128.914 | 6.402 | 11.354 | 1.660 | 40.618 | 126.589 | 6.227 | 11.251 | 2.010 | 84.511 |
| | | 10 | 112.545 | 5.098 | 10.609 | 1.885 | 120.031 | 128.775 | 6.434 | 11.348 | 1.532 | 38.764 | 127.986 | 6.368 | 11.313 | 1.520 | 32.213 |
| | | 12 | 110.699 | 4.902 | 10.521 | 2.177 | 199.450 | 130.907 | 6.626 | 11.441 | 1.443 | 33.523 | 132.033 | 6.685 | 11.491 | 1.217 | 11.881 |
| | | Avg | 111.445 | 4.973 | 10.556 | 2.466 | 207.884 | 128.953 | 6.415 | 11.356 | 1.740 | 53.001 | 128.471 | 6.378 | 11.334 | 1.835 | 66.089 |
| | TaTS (ours) | 6 | 107.430 | 4.658 | 10.365 | 4.255 | 528.013 | 127.814 | 6.269 | 11.306 | 2.263 | 110.185 | 122.803 | 5.927 | 11.082 | 2.966 | 185.980 |
| | | 8 | 107.108 | 4.750 | 10.349 | 3.531 | 455.878 | 125.990 | 6.217 | 11.225 | 2.390 | 135.504 | 123.882 | 6.057 | 11.130 | 2.505 | 151.979 |
| | | 10 | 110.061 | 4.891 | 10.491 | 3.615 | 562.774 | 125.737 | 6.248 | 11.213 | 2.231 | 134.082 | 125.666 | 6.262 | 11.210 | 2.232 | 128.702 |
| | | 12 | 109.361 | 4.847 | 10.458 | 3.060 | 457.480 | 127.103 | 6.369 | 11.274 | 1.885 | 86.102 | 125.971 | 6.290 | 11.224 | 1.899 | 90.926 |
| | | Avg | 108.490 | 4.787 | 10.416 | 3.615 | 501.036 | 126.661 | 6.276 | 11.255 | 2.192 | 116.468 | 124.581 | 6.134 | 11.162 | 2.401 | 139.397 |
| Social Good | Uni-modal | 6 | 1.234 | 0.684 | 0.973 | 1.910 | 69.347 | 0.773 | 0.425 | 0.668 | 0.872 | 13.761 | 0.848 | 0.427 | 0.684 | 1.208 | 27.858 |
| | | 8 | 1.236 | 0.664 | 0.956 | 1.998 | 80.017 | 0.871 | 0.449 | 0.708 | 0.933 | 11.089 | 0.877 | 0.460 | 0.719 | 1.520 | 56.020 |
| | | 10 | 1.332 | 0.708 | 1.026 | 2.116 | 93.499 | 0.910 | 0.498 | 0.755 | 0.905 | 6.360 | 0.976 | 0.541 | 0.783 | 1.725 | 63.227 |
| | | 12 | 1.312 | 0.749 | 1.046 | 2.231 | 75.925 | 0.926 | 0.644 | 0.818 | 1.028 | 7.619 | 0.938 | 0.508 | 0.750 | 1.389 | 28.551 |
| | | Avg | 1.278 | 0.701 | 1.000 | 2.064 | 79.697 | 0.870 | 0.504 | 0.737 | 0.934 | 9.707 | 0.910 | 0.484 | 0.734 | 1.460 | 43.914 |
| | MM-TSFLib | 6 | 1.200 | 0.664 | 0.954 | 1.890 | 66.499 | 0.772 | 0.414 | 0.666 | 0.726 | 8.039 | 0.784 | 0.422 | 0.673 | 0.938 | 18.507 |
| | | 8 | 1.175 | 0.589 | 0.891 | 1.935 | 84.135 | 0.829 | 0.405 | 0.673 | 0.746 | 6.869 | 0.845 | 0.426 | 0.669 | 1.056 | 19.505 |
| | | 10 | 1.268 | 0.689 | 1.002 | 1.756 | 44.172 | 0.876 | 0.466 | 0.724 | 0.855 | 6.248 | 0.913 | 0.486 | 0.731 | 1.252 | 25.176 |
| | | 12 | 1.272 | 0.739 | 1.032 | 2.048 | 56.789 | 0.879 | 0.541 | 0.765 | 0.888 | 7.772 | 0.882 | 0.510 | 0.744 | 1.021 | 13.844 |
| | | Avg | 1.229 | 0.670 | 0.970 | 1.907 | 62.899 | 0.839 | 0.457 | 0.707 | 0.804 | 7.232 | 0.856 | 0.461 | 0.704 | 1.067 | 19.258 |
| | TaTS (ours) | 6 | 1.057 | 0.564 | 0.877 | 1.451 | 33.099 | 0.724 | 0.448 | 0.660 | 0.765 | 7.156 | 0.740 | 0.375 | 0.616 | 0.700 | 3.824 |
| | | 8 | 1.184 | 0.686 | 0.954 | 1.476 | 20.587 | 0.788 | 0.463 | 0.696 | 0.890 | 8.039 | 0.805 | 0.393 | 0.632 | 0.828 | 11.467 |
| | | 10 | 1.278 | 0.714 | 0.986 | 1.578 | 21.226 | 0.828 | 0.468 | 0.691 | 0.880 | 8.487 | 0.842 | 0.485 | 0.700 | 1.062 | 11.719 |
| | | 12 | 1.261 | 0.699 | 0.979 | 1.544 | 21.227 | 0.899 | 0.455 | 0.685 | 0.785 | 4.917 | 0.842 | 0.424 | 0.650 | 0.777 | 5.973 |
| | | Avg | 1.195 | 0.666 | 0.949 | 1.512 | 24.035 | 0.810 | 0.459 | 0.686 | 0.830 | 7.150 | 0.807 | 0.419 | 0.649 | 0.842 | 8.246 |
| Traffic | Uni-modal | 6 | 0.201 | 0.295 | 0.416 | 0.316 | 0.542 | 0.197 | 0.352 | 0.434 | 0.303 | 0.328 | 0.203 | 0.344 | 0.438 | 0.299 | 0.307 |
| | | 8 | 0.205 | 0.296 | 0.419 | 0.299 | 0.440 | 0.202 | 0.358 | 0.440 | 0.320 | 0.369 | 0.224 | 0.364 | 0.460 | 0.302 | 0.285 |
| | | 10 | 0.200 | 0.297 | 0.413 | 0.310 | 0.479 | 0.196 | 0.351 | 0.430 | 0.296 | 0.314 | 0.209 | 0.353 | 0.445 | 0.298 | 0.295 |
| | | 12 | 0.241 | 0.303 | 0.448 | 0.350 | 0.653 | 0.213 | 0.358 | 0.448 | 0.326 | 0.397 | 0.200 | 0.323 | 0.426 | 0.316 | 0.416 |
| | | Avg | 0.212 | 0.298 | 0.424 | 0.319 | 0.528 | 0.202 | 0.355 | 0.438 | 0.311 | 0.352 | 0.209 | 0.346 | 0.442 | 0.304 | 0.326 |
| | MM-TSFLib | 6 | 0.195 | 0.248 | 0.390 | 0.247 | 0.360 | 0.166 | 0.297 | 0.387 | 0.285 | 0.391 | 0.164 | 0.291 | 0.384 | 0.271 | 0.340 |
| | | 8 | 0.204 | 0.274 | 0.401 | 0.277 | 0.404 | 0.169 | 0.306 | 0.393 | 0.285 | 0.372 | 0.172 | 0.305 | 0.395 | 0.274 | 0.318 |
| | | 10 | 0.207 | 0.274 | 0.409 | 0.268 | 0.407 | 0.169 | 0.305 | 0.391 | 0.282 | 0.358 | 0.169 | 0.297 | 0.391 | 0.283 | 0.393 |
| | | 12 | 0.241 | 0.292 | 0.420 | 0.343 | 0.795 | 0.183 | 0.290 | 0.392 | 0.292 | 0.414 | 0.181 | 0.288 | 0.392 | 0.287 | 0.389 |
| | | Avg | 0.212 | 0.272 | 0.405 | 0.284 | 0.492 | 0.172 | 0.299 | 0.391 | 0.286 | 0.384 | 0.171 | 0.295 | 0.390 | 0.279 | 0.360 |
| | TaTS (ours) | 6 | 0.161 | 0.225 | 0.356 | 0.253 | 0.440 | 0.159 | 0.282 | 0.375 | 0.266 | 0.337 | 0.159 | 0.274 | 0.373 | 0.269 | 0.384 |
| | | 8 | 0.167 | 0.237 | 0.353 | 0.258 | 0.416 | 0.161 | 0.289 | 0.378 | 0.270 | 0.340 | 0.156 | 0.268 | 0.367 | 0.266 | 0.386 |
| | | 10 | 0.163 | 0.214 | 0.339 | 0.250 | 0.477 | 0.157 | 0.276 | 0.368 | 0.274 | 0.399 | 0.157 | 0.270 | 0.367 | 0.267 | 0.383 |
| | | 12 | 0.217 | 0.242 | 0.379 | 0.293 | 0.575 | 0.181 | 0.279 | 0.387 | 0.289 | 0.429 | 0.183 | 0.285 | 0.391 | 0.292 | 0.423 |
| | | Avg | 0.177 | 0.229 | 0.357 | 0.264 | 0.477 | 0.164 | 0.281 | 0.377 | 0.275 | 0.376 | 0.164 | 0.274 | 0.374 | 0.274 | 0.394 |

Table 28: Full forecasting results for various datasets using different time series modeling methods. Our TaTS framework seamlessly enhances existing time series models to effectively handle time series with concurrent texts. Avg: the average results across all prediction lengths.

| Methods | | | TaTS (ours) + iTransformer | | TaTS (ours) + PatchTST | | TaTS (ours) + FiLM | | N-BEATS (2020) | | N-HiTS (2022) | | TCN (2018) | | ChatTime (2025) | | GPT4MTS (2024) | |
|---|---|---|---|---|---|---|---|---|---|---|---|---|---|---|---|---|---|---|
| Datasets | | | MSE | MAE | MSE | MAE | MSE | MAE | MSE | MAE | MSE | MAE | MSE | MAE | MSE | MAE | MSE | MAE |
| Time-MMD (2024a) | Agriculture | 6 | 0.067 | 0.184 | 0.066 | 0.171 | 0.087 | 0.205 | 2.436 | 1.210 | 2.274 | 1.134 | 3.957 | 1.770 | 0.502 | 0.441 | 0.197 | 0.292 |
| | | 8 | 0.094 | 0.210 | 0.096 | 0.217 | 0.110 | 0.223 | 3.671 | 1.446 | 1.575 | 0.866 | 5.078 | 1.919 | 0.510 | 0.449 | 0.352 | 0.426 |
| | | 10 | 0.122 | 0.251 | 0.126 | 0.260 | 0.146 | 0.249 | 3.078 | 1.416 | 1.441 | 0.910 | 4.464 | 1.866 | 0.505 | 0.445 | 0.358 | 0.413 |
| | | 12 | 0.153 | 0.271 | 0.166 | 0.292 | 0.196 | 0.328 | 3.883 | 1.759 | 2.119 | 1.217 | 3.173 | 1.633 | 0.517 | 0.452 | 0.404 | 0.444 |
| | | Avg | 0.109 | 0.229 | 0.114 | 0.235 | 0.135 | 0.251 | 3.267 | 1.458 | 1.852 | 1.032 | 4.168 | 1.797 | 0.508 | 0.447 | 0.327 | 0.393 |
| | Climate | 6 | 1.020 | 0.797 | 0.976 | 0.782 | 0.912 | 0.758 | 1.123 | 0.894 | 1.114 | 0.881 | 1.137 | 0.875 | 1.507 | 1.007 | 1.062 | 0.844 |
| | | 8 | 1.025 | 0.797 | 0.995 | 0.803 | 0.917 | 0.751 | 1.070 | 0.844 | 0.981 | 0.791 | 1.046 | 0.804 | 1.524 | 1.012 | 1.177 | 0.893 |
| | | 10 | 1.033 | 0.808 | 1.022 | 0.796 | 0.947 | 0.759 | 1.093 | 0.850 | 1.152 | 0.877 | 1.043 | 0.857 | 1.578 | 1.027 | 1.100 | 0.863 |
| | | 12 | 1.033 | 0.812 | 1.022 | 0.810 | 1.005 | 0.821 | 1.085 | 0.856 | 1.166 | 0.884 | 1.167 | 0.926 | 1.664 | 1.031 | 1.169 | 0.893 |
| | | Avg | 1.028 | 0.804 | 1.004 | 0.798 | 0.945 | 0.772 | 1.093 | 0.861 | 1.103 | 0.858 | 1.098 | 0.866 | 1.568 | 1.019 | 1.127 | 0.873 |
| | Economy | 6 | 0.008 | 0.077 | 0.009 | 0.080 | 0.009 | 0.080 | 0.727 | 0.782 | 0.224 | 0.432 | 5.390 | 2.315 | 0.042 | 0.156 | 0.013 | 0.091 |
| | | 8 | 0.008 | 0.077 | 0.008 | 0.078 | 0.009 | 0.079 | 0.874 | 0.882 | 0.675 | 0.793 | 5.811 | 2.406 | 0.045 | 0.157 | 0.014 | 0.096 |
| | | 10 | 0.009 | 0.079 | 0.009 | 0.079 | 0.009 | 0.079 | 0.999 | 0.849 | 0.486 | 0.647 | 5.426 | 2.324 | 0.052 | 0.165 | 0.015 | 0.098 |
| | | 12 | 0.008 | 0.076 | 0.009 | 0.080 | 0.009 | 0.081 | 1.438 | 1.166 | 0.389 | 0.465 | 5.556 | 2.352 | 0.059 | 0.188 | 0.016 | 0.101 |
| | | Avg | 0.008 | 0.077 | 0.009 | 0.079 | 0.009 | 0.080 | 1.010 | 0.920 | 0.444 | 0.584 | 5.546 | 2.349 | 0.049 | 0.166 | 0.014 | 0.096 |
| | Energy | 12 | 0.106 | 0.234 | 0.106 | 0.234 | 0.118 | 0.245 | 0.146 | 0.271 | 0.195 | 0.336 | 0.165 | 0.318 | 0.128 | 0.253 | 0.117 | 0.244 |
| | | 24 | 0.226 | 0.355 | 0.206 | 0.336 | 0.222 | 0.347 | 0.280 | 0.413 | 0.612 | 0.582 | 0.407 | 0.507 | 0.258 | 0.363 | 0.212 | 0.347 |
| | | 36 | 0.306 | 0.411 | 0.305 | 0.412 | 0.311 | 0.414 | 0.431 | 0.482 | 0.380 | 0.474 | 0.512 | 0.569 | 0.361 | 0.465 | 0.328 | 0.424 |
| | | 48 | 0.421 | 0.502 | 0.416 | 0.501 | 0.434 | 0.510 | 0.460 | 0.532 | 0.301 | 0.460 | 0.634 | 0.654 | 0.473 | 0.588 | 0.421 | 0.497 |
| | | Avg | 0.265 | 0.376 | 0.258 | 0.371 | 0.271 | 0.379 | 0.329 | 0.424 | 0.372 | 0.463 | 0.430 | 0.512 | 0.305 | 0.417 | 0.269 | 0.378 |
| | Environment | 48 | 0.268 | 0.370 | 0.271 | 0.377 | 0.269 | 0.373 | 0.448 | 0.504 | 0.440 | 0.523 | 0.805 | 0.693 | 0.583 | 0.594 | 0.312 | 0.390 |
| | | 96 | 0.267 | 0.370 | 0.279 | 0.376 | 0.279 | 0.377 | 0.540 | 0.582 | 0.421 | 0.522 | 0.738 | 0.686 | 0.575 | 0.591 | 0.345 | 0.408 |
| | | 192 | 0.272 | 0.366 | 0.272 | 0.366 | 0.271 | 0.367 | 0.522 | 0.592 | 0.635 | 0.660 | 1.466 | 0.982 | 0.577 | 0.594 | 0.358 | 0.440 |
| | | 336 | 0.261 | 0.369 | 0.269 | 0.366 | 0.267 | 0.366 | 0.563 | 0.614 | 0.593 | 0.628 | 0.407 | 0.490 | 0.585 | 0.599 | 0.377 | 0.453 |
| | | Avg | 0.267 | 0.369 | 0.273 | 0.371 | 0.272 | 0.371 | 0.518 | 0.573 | 0.522 | 0.583 | 0.854 | 0.713 | 0.580 | 0.594 | 0.348 | 0.422 |
| | Health | 12 | 0.939 | 0.649 | 0.990 | 0.659 | 1.211 | 0.794 | 1.572 | 0.838 | 1.723 | 0.866 | 2.484 | 1.027 | 1.482 | 0.802 | 1.157 | 0.704 |
| | | 24 | 1.251 | 0.712 | 1.288 | 0.764 | 1.414 | 0.827 | 1.680 | 0.981 | 1.621 | 0.905 | 2.072 | 1.070 | 1.645 | 0.956 | 1.743 | 0.904 |
| | | 36 | 1.489 | 0.781 | 1.397 | 0.780 | 1.497 | 0.851 | 1.834 | 0.980 | 1.687 | 0.908 | 1.557 | 0.859 | 1.732 | 0.937 | 1.950 | 0.938 |
| | | 48 | 1.581 | 0.834 | 1.456 | 0.808 | 1.562 | 0.878 | 1.556 | 0.952 | 1.635 | 0.911 | 1.639 | 0.922 | 1.813 | 0.942 | 2.217 | 0.957 |
| | | Avg | 1.315 | 0.744 | 1.283 | 0.753 | 1.421 | 0.838 | 1.660 | 0.938 | 1.666 | 0.898 | 1.938 | 0.970 | 1.668 | 0.909 | 1.766 | 0.875 |
| | Security | 6 | 107.113 | 4.856 | 106.160 | 4.696 | 105.482 | 4.511 | 131.917 | 6.866 | 135.416 | 6.896 | 129.616 | 6.801 | 130.751 | 6.854 | 118.425 | 5.102 |
| | | 8 | 112.560 | 5.204 | 108.803 | 5.052 | 107.657 | 4.792 | 113.095 | 6.287 | 144.874 | 7.780 | 128.578 | 6.936 | 134.526 | 6.875 | 118.952 | 5.106 |
| | | 10 | 113.789 | 5.227 | 111.110 | 5.090 | 109.886 | 4.771 | 163.555 | 7.344 | 156.898 | 6.797 | 160.129 | 7.077 | 131.632 | 6.895 | 119.521 | 5.253 |
| | | 12 | 114.754 | 5.318 | 112.699 | 5.237 | 108.375 | 4.870 | 111.694 | 5.974 | 115.307 | 6.437 | 128.062 | 6.676 | 135.513 | 6.923 | 120.732 | 5.331 |
| | | Avg | 112.054 | 5.151 | 109.693 | 5.019 | 107.850 | 4.736 | 130.065 | 6.618 | 138.124 | 6.978 | 136.596 | 6.873 | 133.106 | 6.887 | 119.407 | 5.198 |
| | Social Good | 6 | 0.942 | 0.398 | 0.923 | 0.436 | 0.992 | 0.588 | 1.752 | 0.654 | 1.446 | 0.626 | 1.535 | 0.917 | 1.213 | 0.608 | 1.214 | 0.485 |
| | | 8 | 0.967 | 0.433 | 0.900 | 0.461 | 1.001 | 0.550 | 0.952 | 0.551 | 1.033 | 0.531 | 1.129 | 0.935 | 1.252 | 0.621 | 1.422 | 0.560 |
| | | 10 | 0.994 | 0.463 | 0.996 | 0.461 | 1.065 | 0.583 | 1.116 | 0.627 | 1.112 | 0.594 | 1.208 | 0.996 | 1.278 | 0.667 | 1.264 | 0.572 |
| | | 12 | 1.045 | 0.514 | 1.069 | 0.501 | 1.358 | 0.782 | 1.445 | 0.827 | 1.498 | 0.846 | 1.385 | 0.996 | 1.313 | 0.712 | 1.757 | 0.622 |
| | | Avg | 0.987 | 0.452 | 0.972 | 0.465 | 1.104 | 0.626 | 1.316 | 0.665 | 1.272 | 0.649 | 1.314 | 0.961 | 1.264 | 0.652 | 1.414 | 0.559 |
| | Traffic | 6 | 0.174 | 0.218 | 0.155 | 0.204 | 0.157 | 0.228 | 0.352 | 0.409 | 0.295 | 0.372 | 0.779 | 0.775 | 0.369 | 0.435 | 0.185 | 0.227 |
| | | 8 | 0.177 | 0.213 | 0.162 | 0.210 | 0.162 | 0.260 | 0.329 | 0.467 | 0.279 | 0.385 | 0.674 | 0.704 | 0.361 | 0.432 | 0.190 | 0.240 |
| | | 10 | 0.186 | 0.225 | 0.167 | 0.214 | 0.169 | 0.269 | 0.342 | 0.474 | 0.281 | 0.404 | 0.702 | 0.758 | 0.363 | 0.427 | 0.189 | 0.246 |
| | | 12 | 0.213 | 0.212 | 0.361 | 0.241 | 0.215 | 0.236 | 0.364 | 0.505 | 0.217 | 0.368 | 0.675 | 0.739 | 0.359 | 0.422 | 0.219 | 0.272 |
| | | Avg | 0.187 | 0.217 | 0.172 | 0.209 | 0.176 | 0.248 | 0.347 | 0.464 | 0.268 | 0.382 | 0.708 | 0.744 | 0.363 | 0.429 | 0.195 | 0.246 |
| FNSPID (2024) | Delta Airlines (DAL) | 6 | 0.064 | 0.161 | 0.059 | 0.160 | 0.065 | 0.167 | 0.253 | 0.417 | 0.191 | 0.342 | 0.247 | 0.422 | 0.068 | 0.169 | 0.069 | 0.179 |
| | | 12 | 0.110 | 0.233 | 0.116 | 0.224 | 0.126 | 0.235 | 0.320 | 0.470 | 0.260 | 0.416 | 0.341 | 0.509 | 0.128 | 0.235 | 0.118 | 0.223 |
| | | Avg | 0.087 | 0.197 | 0.086 | 0.192 | 0.095 | 0.201 | 0.286 | 0.444 | 0.226 | 0.379 | 0.294 | 0.466 | 0.098 | 0.202 | 0.093 | 0.201 |
| | IBM (IBM) | 6 | 0.324 | 0.401 | 0.341 | 0.401 | 0.642 | 0.534 | 0.820 | 0.667 | 0.913 | 0.675 | 2.001 | 1.150 | 0.396 | 0.453 | 0.392 | 0.424 |
| | | 12 | 0.804 | 0.602 | 0.758 | 0.580 | 1.140 | 0.856 | 1.390 | 0.887 | 1.517 | 0.937 | 1.870 | 1.096 | 0.807 | 0.619 | 0.886 | 0.627 |
| | | Avg | 0.564 | 0.501 | 0.550 | 0.490 | 0.891 | 0.695 | 1.105 | 0.777 | 1.215 | 0.806 | 1.936 | 1.123 | 0.602 | 0.536 | 0.639 | 0.525 |
| | JPMorgan Chase (JPM) | 6 | 1.173 | 0.817 | 1.275 | 0.819 | 1.606 | 0.912 | 1.788 | 1.063 | 2.217 | 1.082 | 3.971 | 1.785 | 1.332 | 0.843 | 1.644 | 0.922 |
| | | 12 | 2.214 | 1.124 | 2.470 | 1.161 | 3.421 | 1.281 | 3.050 | 1.287 | 4.635 | 1.383 | 3.557 | 1.637 | 2.742 | 1.244 | 2.622 | 1.323 |
| | | Avg | 1.693 | 0.970 | 1.872 | 0.990 | 2.513 | 1.096 | 2.419 | 1.175 | 3.426 | 1.232 | 3.764 | 1.711 | 2.037 | 1.043 | 2.133 | 1.122 |
| | NVIDIA (NVDA) | 6 | 0.032 | 0.123 | 0.041 | 0.151 | 0.037 | 0.172 | 0.257 | 0.424 | 0.093 | 0.224 | 0.431 | 0.560 | 0.038 | 0.139 | 0.043 | 0.149 |
| | | 12 | 0.054 | 0.159 | 0.055 | 0.162 | 0.063 | 0.177 | 0.288 | 0.445 | 0.151 | 0.301 | 0.483 | 0.587 | 0.067 | 0.183 | 0.066 | 0.170 |
| | | Avg | 0.043 | 0.141 | 0.048 | 0.156 | 0.050 | 0.174 | 0.272 | 0.434 | 0.122 | 0.262 | 0.457 | 0.574 | 0.053 | 0.161 | 0.054 | 0.159 |
| | Pfizer (PFE) | 6 | 0.240 | 0.355 | 0.255 | 0.363 | 0.329 | 0.419 | 0.767 | 0.623 | 0.662 | 0.608 | 0.601 | 0.557 | 0.354 | 0.451 | 0.283 | 0.389 |
| | | 12 | 0.412 | 0.476 | 0.439 | 0.482 | 0.567 | 0.535 | 0.586 | 0.520 | 0.987 | 0.718 | 0.654 | 0.561 | 0.462 | 0.503 | 0.456 | 0.490 |
| | | Avg | 0.326 | 0.416 | 0.347 | 0.422 | 0.448 | 0.477 | 0.676 | 0.572 | 0.824 | 0.663 | 0.628 | 0.559 | 0.408 | 0.477 | 0.369 | 0.439 |
| | Tesla (TSLA) | 6 | 0.101 | 0.241 | 0.071 | 0.204 | 0.098 | 0.235 | 0.166 | 0.306 | 0.262 | 0.452 | 3.784 | 1.929 | 0.123 | 0.256 | 0.166 | 0.315 |
| | | 12 | 0.183 | 0.321 | 0.245 | 0.390 | 0.122 | 0.253 | 0.210 | 0.358 | 0.247 | 0.381 | 3.168 | 1.773 | 0.192 | 0.343 | 0.196 | 0.292 |
| | | Avg | 0.142 | 0.281 | 0.158 | 0.297 | 0.110 | 0.244 | 0.188 | 0.332 | 0.254 | 0.416 | 3.476 | 1.851 | 0.158 | 0.300 | 0.181 | 0.303 |
| FNF (2024b) | Bitcoin Price | 6 | 1.760 | 0.916 | 1.552 | 0.858 | 1.751 | 0.899 | 123.135 | 9.413 | 105.978 | 9.806 | 43.935 | 5.848 | 2.525 | 1.085 | 1.808 | 0.894 |
| | | 8 | 2.147 | 1.006 | 1.998 | 0.964 | 2.458 | 1.080 | 126.985 | 9.703 | 163.110 | 10.690 | 53.318 | 6.412 | 3.352 | 1.163 | 2.521 | 1.050 |
| | | 10 | 3.007 | 1.212 | 2.621 | 1.128 | 3.399 | 1.287 | 157.388 | 11.113 | 179.247 | 10.759 | 65.162 | 7.253 | 3.853 | 1.486 | 3.014 | 1.208 |
| | | 12 | 3.521 | 1.313 | 3.185 | 1.229 | 3.491 | 1.304 | 159.445 | 11.192 | 187.741 | 10.884 | 68.887 | 7.464 | 4.631 | 1.641 | 3.542 | 1.456 |
| | | Avg | 2.609 | 1.112 | 2.339 | 1.045 | 2.775 | 1.142 | 141.738 | 10.355 | 159.019 | 10.535 | 57.826 | 6.744 | 3.590 | 1.344 | 2.721 | 1.152 |
| | Web Traffic | 6 | 15.106 | 2.504 | 17.348 | 2.755 | 17.378 | 2.755 | 21.596 | 3.003 | 21.993 | 2.912 | 16.625 | 2.157 | 19.265 | 2.869 | 18.106 | 2.832 |
| | | 8 | 18.474 | 2.740 | 17.715 | 2.625 | 16.11 | 2.600 | 19.330 | 2.754 | 21.001 | 2.828 | 20.144 | 2.826 | 21.724 | 2.924 | 18.421 | 2.745 |
| | | 10 | 18.313 | 2.783 | 18.008 | 2.643 | 16.23 | 2.516 | 20.866 | 2.901 | 24.046 | 2.930 | 21.976 | 2.947 | 20.164 | 2.875 | 19.902 | 2.792 |
| | | 12 | 19.084 | 2.823 | 18.783 | 2.590 | 22.721 | 3.217 | 27.141 | 3.409 | 26.011 | 3.374 | 27.759 | 3.439 | 21.841 | 2.969 | 20.611 | 2.942 |
| | | Avg | 17.744 | 2.712 | 17.964 | 2.653 | 18.110 | 2.772 | 22.233 | 3.017 | 23.263 | 3.011 | 21.626 | 2.842 | 20.748 | 2.909 | 19.260 | 2.827 |
| | Electricity Demand | 6 | 0.262 | 0.379 | 0.257 | 0.374 | 0.260 | 0.375 | 0.385 | 0.470 | 0.338 | 0.443 | 0.436 | 0.506 | 0.510 | 0.544 | 0.271 | 0.388 |
| | | 8 | 0.274 | 0.391 | 0.257 | 0.370 | 0.244 | 0.360 | 0.419 | 0.497 | 0.426 | 0.504 | 0.501 | 0.565 | 0.524 | 0.562 | 0.284 | 0.369 |
| | | 10 | 0.288 | 0.405 | 0.173 | 0.266 | 0.252 | 0.366 | 0.407 | 0.494 | 0.335 | 0.454 | 0.558 | 0.582 | 0.536 | 0.571 | 0.292 | 0.398 |
| | | 12 | 0.295 | 0.408 | 0.327 | 0.424 | 0.310 | 0.408 | 0.451 | 0.522 | 0.355 | 0.471 | 0.567 | 0.601 | 0.558 | 0.588 | 0.337 | 0.428 |
| | | Avg | 0.280 | 0.396 | 0.254 | 0.358 | 0.266 | 0.377 | 0.416 | 0.496 | 0.364 | 0.468 | 0.516 | 0.564 | 0.532 | 0.566 | 0.296 | 0.395 |

## E.4 FULL IMPUTATION RESULTS

Table 29: Full imputation results for the Climate, Economy, and Traffic datasets using PatchTST, DLinear and FiLM as time series models. Compared to numerical-only unimodal modeling and MM-TSFLib, our TaTS framework seamlessly enhances existing time series models to effectively handle time series with concurrent texts. Promotion: the improvement of the best baseline.

| Models | | PatchTST (2023) | | | | | DLinear (2023) | | | | | FiLM (2022a) | | | | |
|---|---|---|---|---|---|---|---|---|---|---|---|---|---|---|---|---|
| Metric | | MSE | MAE | RMSE | MAPE | MSPE | MSE | MAE | RMSE | MAPE | MSPE | MSE | MAE | RMSE | MAPE | MSPE |
| Climate | Uni-modal | 1.111 | 0.846 | 1.052 | 3.898 | 442.010 | 0.969 | 0.801 | 0.983 | 1.881 | 59.681 | 1.123 | 0.829 | 1.059 | 2.647 | 128.421 |
| | MM-TSFLib | 1.010 | 0.821 | 1.002 | 2.156 | 81.983 | 0.963 | 0.802 | 0.980 | 1.394 | 18.580 | 1.130 | 0.833 | 1.061 | 1.949 | 51.811 |
| | **TaTS (ours)** | 0.878 | 0.720 | 0.937 | 1.573 | 35.877 | 0.912 | 0.757 | 0.951 | 1.701 | 25.242 | 0.820 | 0.718 | 0.902 | 1.478 | 30.622 |
| | Promotion | 13.1% | 12.3% | 6.5% | 27.0% | 56.2% | 5.3% | 5.5% | 3.0% | -22.0% | -35.9% | 27.0% | 13.4% | 14.8% | 24.2% | 40.9% |
| Economy | Uni-modal | 0.029 | 0.138 | 0.170 | 0.051 | 0.004 | 0.057 | 0.190 | 0.239 | 0.068 | 0.007 | 0.077 | 0.209 | 0.277 | 0.076 | 0.010 |
| | MM-TSFLib | 0.026 | 0.137 | 0.161 | 0.049 | 0.003 | 0.061 | 0.196 | 0.247 | 0.069 | 0.007 | 0.075 | 0.203 | 0.271 | 0.072 | 0.009 |
| | **TaTS (ours)** | 0.017 | 0.107 | 0.128 | 0.038 | 0.002 | 0.045 | 0.171 | 0.210 | 0.061 | 0.006 | 0.054 | 0.168 | 0.232 | 0.061 | 0.007 |
| | Promotion | 34.6% | 21.9% | 20.5% | 22.4% | 33.3% | 21.1% | 10.0% | 12.1% | 10.3% | 14.3% | 28.0% | 17.2% | 14.4% | 15.3% | 22.2% |
| Traffic | Uni-modal | 0.210 | 0.339 | 0.444 | 0.358 | 0.600 | 0.245 | 0.417 | 0.489 | 0.430 | 0.720 | 0.175 | 0.311 | 0.409 | 0.343 | 0.508 |
| | MM-TSFLib | 0.189 | 0.341 | 0.428 | 0.391 | 0.690 | 0.179 | 0.335 | 0.419 | 0.362 | 0.630 | 0.169 | 0.288 | 0.396 | 0.316 | 0.545 |
| | **TaTS (ours)** | 0.131 | 0.248 | 0.331 | 0.312 | 0.647 | 0.134 | 0.297 | 0.352 | 0.266 | 0.270 | 0.137 | 0.242 | 0.354 | 0.281 | 0.388 |
| | Promotion | 30.7% | 26.8% | 22.7% | 12.8% | -7.8% | 25.1% | 11.3% | 16.0% | 26.5% | 57.1% | 18.9% | 16.0% | 10.6% | 11.1% | 23.6% |

## E.5 FULL HYPERPARAMETER STUDY RESULTS OF LEARNING RATE

Figure 11: Parameter study on the learning rate. We evaluate the impact of varying the learning rate in $\{0.00005, 0.0001, 0.00015, 0.0002, 0.00025, 0.0003\}$ by reporting the mean squared error (MSE) of our TaTS framework across datasets. The results demonstrate that TaTS maintains stable performance across different learning rate choices.

## E.6 FULL HYPERPARAMETER STUDY RESULTS OF TEXT EMBEDDING DIMENSION

Figure 12: Parameter study on the projection dimension of paired texts. We vary the text projection dimension in $\{6, 12, 18, 24\}$ and report the mean squared error (MSE) of our TaTS framework across datasets. The results indicate that TaTS maintains robust performance across different choices of text projection dimensions.

## E.7 FULL ABLATION STUDY RESULTS USING DIFFERENT TEXT ENCODERS

We conduct experiments to evaluate the performance of our TaTS with multiple language encoders. Specifically, we evaluation TaTS with BERT-110M[2], GPT2-1.5B[3] and LLaMA2-7B[4] as the language encoders. The results, presented in Figure 13, demonstrate that TaTS remains robust across different text encoders and consistently outperforms the baselines.

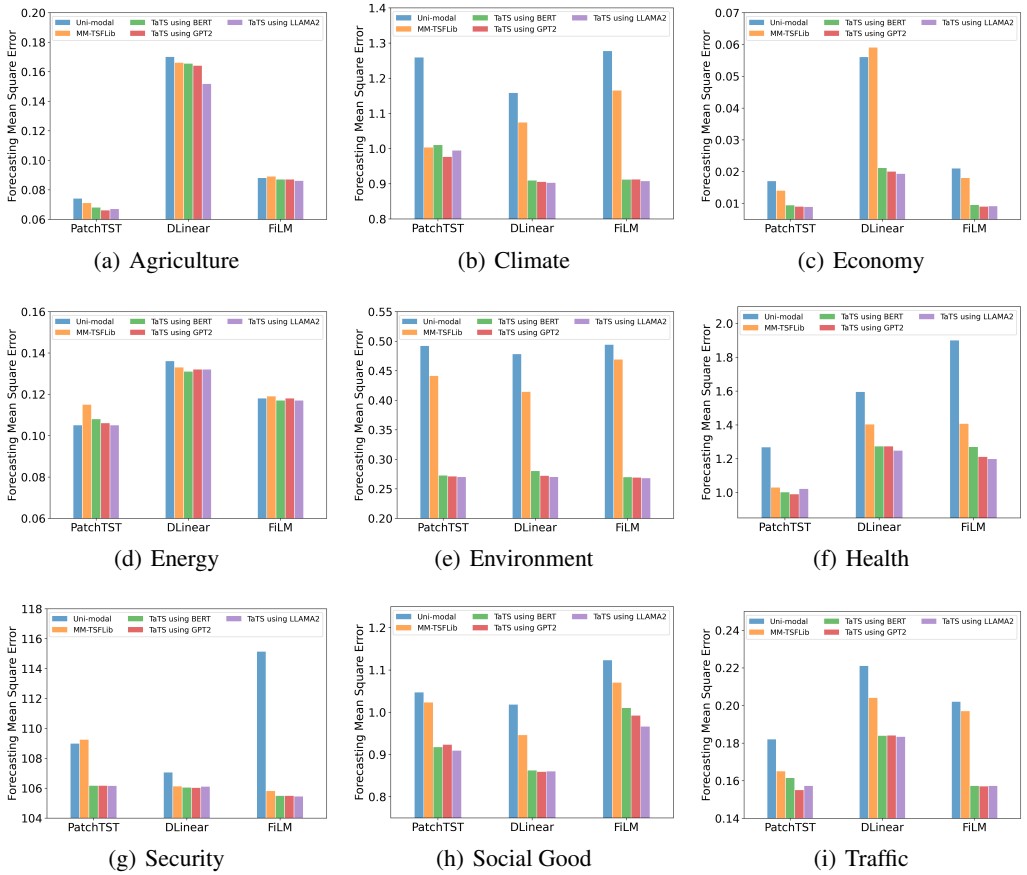

Figure 13: Performance comparison of different text encoders within the TaTS framework. Specifically, we evaluate BERT-110M, GPT2-1.5B, and LLaMA2-7B across multiple datasets using PatchTST (transformer-based model), DLinear (linear-based model), and FiLM (frequency-based model). TaTS maintains relatively stable performance across various models and datasets.

In the main experiments, the text embeddings are obtained by a mean pooling over token embeddings. There are more advanced embedding techniques. To this end, we conducted an additional experiment using LLaMA-3.2-1B to generate sentence-level embeddings, while keeping all other settings the same. The results are shown below in Table 30.

Table 30: Mean Square Error of TaTS forecasting with different text embedding methods.

| Text Embedding Method | Agriculture | Climate | Economy | Security | Social Good | Traffic |
|---|---|---|---|---|---|---|
| GPT-2 average pooling (prediction length 6) | 0.067 | 1.020 | 0.008 | 107.113 | 0.942 | 0.174 |
| LLaMA-3.2-1B sentence embeddings (prediction length 6) | 0.067 | 1.015 | 0.008 | 106.625 | 0.935 | 0.179 |
| GPT-2 average pooling (prediction length 12) | 0.153 | 1.033 | 0.008 | 114.754 | 1.045 | 0.213 |
| LLaMA-3.2-1B sentence embeddings (prediction length 12) | 0.152 | 1.021 | 0.008 | 114.219 | 1.025 | 0.209 |

---

[2] https://huggingface.co/google-bert/bert-base-uncased
[3] https://huggingface.co/openai-community/gpt2
[4] https://huggingface.co/meta-llama/Llama-2-7b

### E.8 FULL EFFICIENCY RESULTS: COMPUTATIONAL OVERHEAD VS. PERFORMANCE GAIN TRADE-OFFS

We conduct experiments to analyze the efficiency of TaTS, with results presented in Figure 14. Each subfigure visualizes the training time per epoch and the forecasting mean squared error (MSE) for different time series models, represented as transparent colored scatter points. The average performance is computed and marked with cross markers: the green cross represents the average performance of TaTS, while the red and blue crosses indicate the average performance of the baseline models. As TaTS introduces a lightweight MLP and augments the original time series with auxiliary variables projected from paired texts, it incurs a slight computational overhead, with an average increase of $\sim 8\%$. Yet this trade-off results in a $\sim 14\%$ average improvement of forecasting MSE.

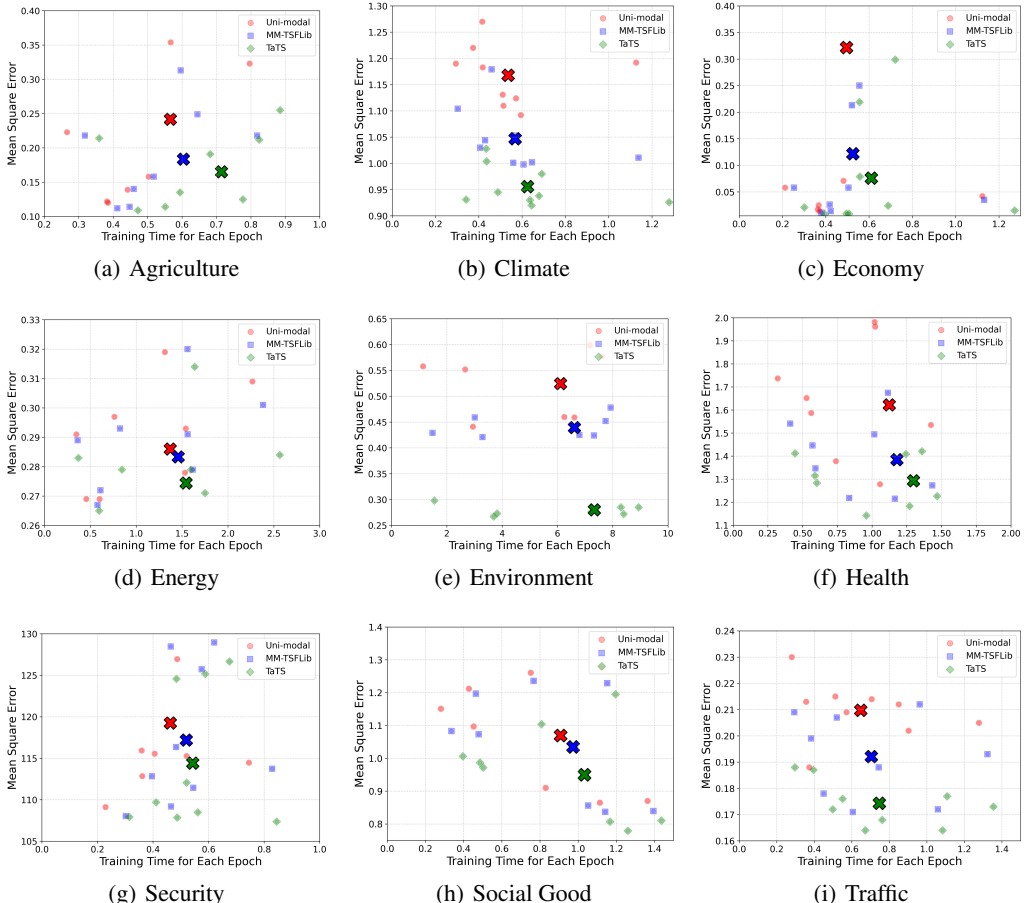

Figure 14: Efficiency and forecasting performance comparison of uni-modal time-series modeling, MM-TSFLib, and our TaTS framework. While TaTS incurs a slight increase in training time due to the augmentation of auxiliary variables projected from paired texts, it significantly enhances forecasting accuracy, achieving a lower MSE.

