# OpenReview forum: "Language in the Flow of Time: Time-Series-Paired Texts Weaved into a Unified Temporal Narrative"
_ICLR.cc/2026/Conference — ICLR 2026 Poster_

### Official Review · Reviewer_1Lkf · 2025-10-27

**Soundness:** 3
**Presentation:** 3
**Contribution:** 3
**Rating:** 6
**Confidence:** 4

**Summary:**

This paper introduces the TaTS framework, a novel approach for integrating contextual textual information with numerical time series data to significantly enhance forecasting accuracy. The authors propose a method that projects auxiliary variables from paired texts to augment the original time series representation, theoretically underpinned by the Platonic Representation Hypothesis, which posits a convergence of multimodal representations into shared spaces. The central contribution lies in the demonstrated effectiveness of this multimodal fusion: empirical results across nine diverse real-world domains consistently show that TaTS achieves an average of approximately 14% improvement in forecasting Mean Squared Error (MSE), justifying the minor incurred computational overhead of around 8% in training time.

**Strengths:**

1. Significant Empirical Improvement: The framework demonstrates a substantial and consistent performance gain (around 14% MSE reduction) across a wide array of domains (e.g., Economy, Climate, Health), indicating high robustness and generalizability.

2. Sound Theoretical Grounding: The methodology is well-motivated by the Platonic Representation Hypothesis, providing a strong conceptual basis for unifying textual and numerical time series representations.

3. Practical Efficiency Analysis: The paper includes a necessary trade-off analysis, clearly quantifying that the high performance benefits come with only a marginal and acceptable increase in computational overhead (around 8%).

**Weaknesses:**

1. Interpretability of Textual Influence: While the model is effective, the specific mechanism by which the projected auxiliary variables qualitatively impact or modulate the time series dynamics is not deeply explored. More detailed visualization or ablation studies on the how the text is woven into the temporal narrative would strengthen the paper.

2. Scalability to Text Length/Complexity: The paper does not thoroughly discuss the framework's scalability or performance behavior when dealing with significantly longer or structurally complex paired texts (e.g., full documents vs. short captions).

3. Sensitivity to Text Quality: A crucial missing component is an analysis of how the model performs when the paired text is noisy, weakly correlated, or intentionally irrelevant, which would better demonstrate the robustness of the feature extraction mechanism.

**Questions:**

Please refer to Weaknesses.

---

> ### Author Response · Authors · 2025-11-22
> **Clarifications on Textual Influence, Text Length Scalability, and Text Quality**
>
> We sincerely thank the reviewer for the constructive and thoughtful feedback. If you have further questions, we would like to provide more details in the discussion phase.
>
>
>
> > Interpretability of Textual Influence (W1)
>
> Interpretability is indeed an important direction for multimodal learning. In the current work, we have already conducted analyses such as alternating or perturbing the textual inputs and examining the resulting performance changes, which provide initial evidence that the projected auxiliary variables meaningfully influence the forecasting process. However, deeper model-level interpretability requires analyzing the underlying time-series backbones that TaTS builds upon, which might be non-trivial and beyond what can be completed within this rebuttal or the scope of the current work. We view this as a valuable direction and plan to extend TaTS with more explicit interpretability tools (e.g., attention visualizations, concept-level probes, and counterfactual text perturbations) in future studies. We would also welcome any suggestions on potential extensions or directions that could further strengthen future work.
>
>
> > Scalability to Text Length (W2)
>
> We appreciate the reviewer’s insightful comment regarding the scalability of TaTS with respect to longer or structurally complex paired texts. **In the revised manuscript, we have added a dedicated analysis in Appendix C.2 that explicitly discusses the computational complexity regarding text length.**
>
> Specifically, if the average text length is $l$ and the language encoder processes each token in $O(t_token)$ time, the total embedding cost scales linearly with the text length as $O(t_{token​}⋅l)$ for each timestamp. Additionally, we conducted experiments and reported the time cost for text embedding on our hardware.
>
> We also note that TaTS is modular with respect to the language encoder: practitioners may substitute lightweight encoders (e.g., small Transformers or MLP-based tokenizers) when dealing with longer documents. This flexibility allows TaTS to adapt to different text lengths and complexity levels while maintaining computational efficiency.
>
>
>
> > Sensitivity to Text Quality (W3)
>
> We appreciate the reviewer’s insights regarding the quality and reliability of textual data in real-world settings. We fully agree, which is why our original submission already includes ablation studies on text-corrupted and text-random-drop datasets. **In the revised manuscript, we provide more discussions on how TaTS may adapt to real-world text scenarios in Appendix B.2 Page 25.**
>
> According to our ablations of text-corrupted datasets, noisy text could cause negative transfer. In this scenario, the improvement from TaTS drops significantly to matching or even being worse than the uni-modal baseline. A practical mitigation is to stochastically drop some texts to control the influence of the text modality. The drop rate can be tuned on a validation set to identify the most effective way to use (or selectively ignore) textual descriptions in practice.
>
> To further validate this mitigation, **in Section 5.2, we added a new ablation in which we intentionally corrupt the texts to induce negative transfer and then apply random text dropping**. The results show that dropping noisy texts allows TaTS to recover performance close to the unimodal baseline, demonstrating the effectiveness of this strategy in handling unreliable text.
>
> - Average TaTS performance with various prediction lengths and with iTransformer
> | Method | Climate MSE | Climate MAE | Security MSE | Security MAE | Traffic MSE | Traffic MAE |
> |-|-|-|-|-|-|-|
> | TaTS + corrupted data | 1.242 | 0.895 | 117.82 | 5.767 | 0.223 | 0.265 |
> | TaTS + 60% corrupted data | 1.218 | 0.879 | 117.43 | 5.730 | 0.215 | 0.260 |
> | TaTS + 20% corrupted data | 1.201 | 0.878 | 116.68 | 5.713 | 0.216 | 0.255 |
> | uni-modal + original data | 1.183 | 0.871 | 115.94 | 5.660 | 0.213 | 0.238 |

---

### Official Review · Reviewer_aWnj · 2025-10-28

**Soundness:** 3
**Presentation:** 3
**Contribution:** 3
**Rating:** 6
**Confidence:** 5

**Summary:**

The paper introduces TaTS, a simple framework that integrates timestamped text with numerical time series by converting text embeddings into auxiliary features for standard models. It is motivated by Chronological Textual Resonance (CTR), which captures periodic alignment between textual and numeric signals, and defines TT-Wasserstein to quantify this alignment. TaTS works with existing architectures without modification, improving forecasting and imputation performance across multiple datasets while adding minimal computational cost.

**Strengths:**

- Evaluated across multiple datasets and diverse time-series backbones, supporting robustness and generalizability.
- Clear, well-structured framework: CTR (motivation), TaTS (mechanism), and TT-Wasserstein (diagnostic) are distinctly defined and complement each other.
- Plug-and-play simplicity: TaTS adds a small projector to turn per-timestamp texts into features and works with existing TS backbones without architecture changes.

**Weaknesses:**

- CTR/TT-Wasserstein focus on magnitude-only spectral alignment, favoring stable periodicities and largely ignoring phase—so they can undervalue datasets where texts are leading or lagging indicators or where alignment is time-varying/non-stationary, even if the texts are predictive.
- Limited discussion of TS→text (inverse direction): brief related-work mention and a single zero-shot ChatTime baseline; no deeper analysis or dedicated experiments.
- Limited native multimodal baselines: ChatTime is the only truly native multimodal method evaluated, while others are adapted uni-modal TS models; MM-TSFLib is further extended for imputation since it doesn’t support it natively.

**Questions:**

Please address the identified weaknesses and limitations noted above.

---

> ### Author Response · Authors · 2025-11-22
> **Clarifications, TS->Text Discussion, and Expanded Multimodal Baselines**
>
> We sincerely thank the reviewer for the constructive and thoughtful feedback. If you have further questions, we would like to provide more details in the discussion phase.
>
> > TT-Wasserstein undervalues leading, lagging, or non-stationary texts (W1)
>
> We appreciate the reviewer’s insights regarding the availability, alignment, and quality of textual descriptions in real-world settings. These are indeed important considerations for deploying multimodal forecasting models as we discussed in our Limitations section (Appendix B.2.).
>
> Specifically, in cases where texts are not aligned or the alignment is time-varying, we want to clarify that small temporal shifts (leading/lagging) primarily change the phase but not the frequency of the textual patterns, so TaTS can still benefit from the preserved periodic structure. However, we agree that when the text alignment is non-stationary, TT-Wasserstein may no longer fully capture the mismatch. In such cases, practitioners may (1) apply existing alignment or synchronization techniques during preprocessing, and (2) use TT-Wasserstein as a diagnostic to decide how strongly the text modality should be incorporated. When the text is extremely noisy even after preprocessing, we recommend reducing reliance on the text modality. Our ablations show that in such cases, the improvement from TaTS drops significantly to matching or even being worse than the uni-modal baseline.
>
>
> > Discussion on TS->Text (W2)
>
> We thank the reviewer for pointing out our missing discussion on TS->text. In the revision, we expanded the related-work discussion to reflect this emerging line of research. Specifically, **on Page 10, we now highlight recent advances in temporal reasoning and time-series question answering** [1]. We also reference new benchmarks such as MTBench [2] and TSAIA [3], as well as TimeXL [4] on generate interpretable explanations.
>
>
> > Comparison with native multimodal baselines. (W3)
>
> We fully agree with the reviewer on the importance of comparing with native multimodal baselines, which is a key motivation for us to compare with ChatTime and MM-TSFLib.
>
>
> In this rebuttal, to strengthen the empirical study, we incorporated several additional multimodal competitors. Specifically, we added comparisons with GPT4MTS [2] in Table 4 and in Table 28.
>
> - Comparison of our TaTS and GPT4MTS on Forecast with News datasets. We report average MSE and MAE metrics across various prediction lengths.
> | Method | Bitcoin Price | Web Traffic | Electricity Demand |
> |-|-|-|-|
> | TaTS + iTransformer (ours) MSE | 2.609 | 17.744 | 0.280 |
> | GPT4MTS MSE | 2.721 | 19.260 | 0.296 |
> | TaTS + iTransformer (ours) MAE | 1.112 | 2.712 | 0.396 |
> | GPT4MTS MAE | 1.152 | 2.827 | 0.395 |
>
> We also noticed some concurrent work on multimodal time series this year. Referring to their released preprints, we provide further comparisons with several very recent multimodal algorithms [3, 4, 5]. The results are summarized in Table 8 in Appendix B.1.
>
> - Comparison of our TaTS and several concurrent multimodal baselines on Time-MMD datasets. We report average MSE metrics across various prediction lengths.
> | Method | Agriculture | Climate | Health |
> |-|-|-|-|
> | TaTS + iTransformer (ours) | 0.201 | 1.227 | 1.372 |
> | TFHTS | 0.571 | 1.782 | 1.514 |
> | MCD-TSF | 0.222 | 1.583 | 1.496 |
> | TeR-TSF | 0.338 | 1.348 | 1.384 |
>
> We compute average rankings across datasets (with lower MSE corresponding to a better rank). TaTS achieves the best overall ranking of 1.83, slightly outperforming the second-best recent method, TeR-TSF, which obtains 2.08.
>
> Overall, the revised manuscript now includes comprehensive comparisons with **a broad suite of native multimodal baselines: MM-TSFLib, ChatTime, GPT4MTS, TFHTS, MCD-TSF, and TeR-TSF**. We hope these expanded comparisons could help address your concern regarding multimodal baselines.
>
>
>
>
>
> [1] A survey of reasoning and agentic systems in time series with large language models. Preprint.
>
> [2] MTBench: A Multimodal Time Series Benchmark for Temporal Reasoning and Question Answering. Preprint.
>
> [3] When LLM Meets Time Series: Can LLMs Perform Multi-Step Time Series Reasoning and Inference. Preprint.
>
> [4] Explainable multi-modal time series prediction with llm-in-the-loop. Preprint.
>
> [5] Time-MMD: Multi-Domain Multimodal Dataset for Time Series Analysis. Liu et al. NeurIPS 2024.
>
> [6] GPT4MTS: Prompt-based Large Language Model for Multimodal Time-series Forecasting. AAAI 2024.
>
> [7] Unveiling the potential of text in high-dimensional time series forecasting. Preprint, 2025.
>
> [8] Multimodal conditioned diffusive time series forecasting. Preprint, 2025.
>
> [9] Text reinforcement for multimodal time series forecasting. Preprint, 2025.

---

### Official Review · Reviewer_w9od · 2025-10-28

**Soundness:** 3
**Presentation:** 2
**Contribution:** 3
**Rating:** 6
**Confidence:** 3

**Summary:**

This paper presents TaTS, a plug-in framework that treats timestamp-aligned texts as auxiliary variables for time-series models. By embedding the texts, projecting them through a small MLP, and concatenating them to numeric inputs, the approach allows standard forecasters and imputers to exploit textual signals without changing their architectures. The work is motivated by Chronological Textual Resonance (CTR), the observation that paired texts often share periodic structure with the target series, and it introduces TT-Wasserstein to quantify this alignment. Extensive experiments across varied datasets and backbones indicate consistent gains, particularly when TT-Wasserstein suggests strong alignment. The computational overhead is modest, which enhances the practical appeal of the method.

**Strengths:**

- The paper introduces a simple, model-agnostic plug-in that treats text embeddings as auxiliary variables, simplifying integration into existing pipelines while keeping computational overhead low.
- The work provides an interpretable alignment rationale via CTR and the TT-Wasserstein metric, which helps explain when and why text improves performance and guides practical deployment.
- Extensive experiments with clear ablations (shuffle/drop text, encoder swaps, multiple backbones and datasets) demonstrate stable gains.

**Weaknesses:**

- The central mechanism is projection followed by concatenation, which may be outperformed by comparably simple baselines, such as gated residuals or lightweight cross-attention, unless comparisons are conducted under strict parameter matching.
- The definition and estimation of TT-Wasserstein lack sufficient statistical treatment; sensitivity to windowing, normalization, and frequency resolution is not examined, and confidence intervals are absent.
-  The data-splitting procedure is not described in adequate detail, creating uncertainty about potential information leakage from contemporaneous or retrospective texts.
- The framework offers no automatic protection against negative transfer when CTR is weak, which could undermine robustness in noisy or weakly aligned settings.

**Questions:**

- How are splits constructed per dataset to ensure texts at time t do not reveal contemporaneous or near-future outcomes, and are audits/filters applied for retrospective or outcome-summarising language?
- Which FFT, windowing, and normalization settings are used for both modalities, how sensitive are results to these choices, and can confidence intervals be reported for the correlation?
- How does TaTS compare with (a) a small gated residual that can down-weight text channels and (b) a parameter-matched cross-attention/FiLM block?
- Has weighting or dropping text channels based on TT-Wasserstein been tested to mitigate negative transfer when alignment is weak?
- What are the training/inference wall-clock costs for different text encoders (e.g., GPT-2, BERT, LLaMA), and what rule of thumb should practitioners use to choose an encoder?

---

> ### Author Response · Authors · 2025-11-22
> **On Alternative Fusion Designs, TT-Wasserstein, Negative Transfer Protection, and Experiment Details (Part I)**
>
> We sincerely thank the reviewer for the constructive and thoughtful feedback. If you have further questions, we would like to provide more details in the discussion phase.
>
>
> > Alternative designs of combining modalities (W1 & Q3)
>
> We thank the reviewer for the insightful comments on alternative fusion designs. Since our primary goal is to investigate the benefit of augmenting time series with textual signals, we opted for the straightforward fusion mechanism (projection + concatenation) that preserves the temporal structure while minimizing architectural confounds.
>
> During this rebuttal, we have added **new experiments in Section 5.2 that compare our MLP-based projection with the alternatives the reviewer suggested**. The results (Table 6) show that both alternatives achieve similar performance to the MLP projection. Our findings indicate that the performance gains of TaTS stem primarily from leveraging temporally aligned textual information, rather than from architectural sophistication in the fusion block. We agree that exploring more fine-grained multimodal fusion designs is an interesting direction, and we now explicitly identify this as future work in the revision.
>
> - TaTS Ablation on Alternative Modality Fusion Architectures
>
> | TaTS Setting              | Climate MSE | Climate MAE | Security MSE | Security MAE | Traffic MSE | Traffic MAE|
> |---------------------------|------------:|------------:|-------------:|-------------:|------------:|------------:|
> | MLP projection (original) | 0.992       | 0.791       | 109.865      | 4.968        | 0.178       | 0.224       |
> | gated residual            | 0.998       | 0.794       | 110.254      | 4.965        | 0.177       | 0.227       |
> | cross-attention           | 0.989       | 0.792       | 109.729      | 4.976        | 0.178       | 0.221       |
>
>
>
> > Stable Estimation of TT-Wasserstein (Q2 & W2)
>
> We thank the reviewer for their insights regarding the statistical treatment of TT-Wasserstein. We have substantially expanded our analysis in Appendix A.1 to address this.
>
> **Our default configuration (used throughout the main paper) employs a rectangular window, L1-normalization, and the native FFT frequency grid determined by the full time-series length. To evaluate robustness, we conduct a series of controlled ablations** in which we vary one component at a time while keeping all others fixed. Specifically, we experiment with three commonly used windowing functions (Hann, Hamming, and Blackman), two alternative amplitude normalizations (min–max and log-scaled), and two additional frequency resolutions (zero-padding and down-sampling). The results (in Table 10) show that TT-Wasserstein is stable across different configuration choices. We additionally quantify estimation uncertainty using a block bootstrap over time for both modalities and report the results in Table 11.
>
> - TT-Wasserstein measure of Time-MMD datasets (partial)
> | Configuration | Climate | Economy | Security | Energy | Health | Environment |
> |-|-|-|-|-|-|-|
> | Default | 0.025 | 0.022 | 0.049 | 0.307 | 0.233 | 0.302 |
> | Hann Window | 0.029 | 0.016 | 0.041 | 0.304 | 0.243 | 0.306 |
> | Minmax Normalization | 0.035 | 0.025 | 0.043 | 0.345 | 0.291 | 0.332 |
> | Down-sampled frequency resolution | 0.026 | 0.039 | 0.050 | 0.259 | 0.197 | 0.298 |
>
>
>
>
>
> > Data Splitting (W3 & Q1)
>
> For all datasets, we follow a chronological 80%/10%/10% train–validation–test split. When forecasting the value at timestamp T, the model only receives time-series inputs and text inputs with timestamps \leq T-1; texts at timestamp T or later are masked. This ensures that no contemporaneous information is accessible during prediction.
>
> All datasets we use provide pre-aligned text–time-series pairs by associating each text with the timestamp at which it was originally released. While we do not apply automatic filters, we conducted human audits on random samples from each dataset to check for retrospective or outcome-summarizing leakage. We did not observe such leakage in any of the datasets.
> **We have added this description to the revised manuscript, Appendix D.4.3. for clarity.**

---

> ### Author Response · Authors · 2025-11-22
> **On Alternative Fusion Designs, TT-Wasserstein, Negative Transfer Protection, and Experiment Details (Part II)**
>
> > Automatic protection against negative transfer (W4 & Q4)
>
> Though beyond the scope of current research, we thank the reviewer for their insights on robustness. In our original submission, ablations on corrupted-text and random-drop settings show that severely degraded texts can indeed cause negative transfer and reduce gains to the level of (or slightly below) the unimodal baseline.
>
> Following the reviewer’s suggestion, we further evaluated whether weighting or selectively dropping text channels can mitigate this issue. **In the revision (Section 5.2), we added a new ablation** in which we (i) intentionally corrupt the textual inputs to induce negative transfer, and (ii) apply stochastic text dropping. The results show that random text dropping effectively suppresses the influence of unreliable textual inputs and allows TaTS to recover performance close to the unimodal baseline.
>
> - Average TaTS performance with various prediction lengths and with iTransformer
> | Method | Climate MSE | Climate MAE | Security MSE | Security MAE | Traffic MSE | Traffic MAE |
> |-|-|-|-|-|-|-|
> | TaTS + corrupted data | 1.242 | 0.895 | 117.82 | 5.767 | 0.223 | 0.265 |
> | TaTS + 60% corrupted data | 1.218 | 0.879 | 117.43 | 5.730 | 0.215 | 0.260 |
> | TaTS + 20% corrupted data | 1.201 | 0.878 | 116.68 | 5.713 | 0.216 | 0.255 |
> | uni-modal + original data | 1.183 | 0.871 | 115.94 | 5.660 | 0.213 | 0.238 |
>
>
> These findings **support the reviewer’s intuition that reducing the contribution of unreliable texts provides a practical safeguard against negative transfer**. Practitioners can tune the drop rate on a validation set to identify the most effective way to use (or selectively ignore) textual descriptions. We also view more principled weighting based on our TT-Wasserstein as a promising direction for future work.
>
>
>
>
>
> > Wall clock costs
>
> TaTS does not train any text encoder; the encoder is used only once to obtain fixed text embeddings. In practice, we batch-embed all paired texts on GPU and cache the resulting embeddings to disk, so the embedding step incurs a one-time preprocessing cost rather than a per-iteration training overhead. **The actual wall-clock times measured on our hardware are reported in Table 11 (Appendix C.2)**.
>
> **In Appendix C.2, we also provide the time-complexity analysis.** We find that, if the average text length is l and the language encoder processes each token in O(t_token) time, the total embedding cost scales linearly with the text length as O(t_token​⋅l) for each timestamp.
>
> Therefore, we recommend using smaller encoders when computational resources are limited or when the dataset contains many timestamps or long paired texts. Practitioners may also refer to our reported wall-clock costs to estimate the preprocessing time for their own applications.

---

### Official Review · Reviewer_vhxM · 2025-10-30

**Soundness:** 2
**Presentation:** 3
**Contribution:** 2
**Rating:** 4
**Confidence:** 3

**Summary:**

This paper introduces TaTS, a new multimodal time series forecasting benchmark compatible with any time series forecasting model. TaTS embeds the textual descriptions of time series, projects them using an MLP, and then combines them with the original time series to create a new input for forecasting models. The motivation behind this architecture is that time series paired with text often exhibit periodic patterns that reflect the dynamics of the numerical time series. Additionally, the authors introduce TT-Wasserstein, a metric designed to evaluate the alignment between the time series and the associated text.

**Strengths:**

1. The proposed architecture is model agnostic and delivers good improvement of the base model.

2. Incorporate textual information into time series if timely and interesting.

**Weaknesses:**

1. The paper lacks comparison with other multimodal algorithms. For example, the authors don't compare their approach with TimeMMD, which is also model-agnostic and offers similar improvements over unimodal models. There are also several other multimodal algorithms available that could serve as valuable comparisons.

2. The method requires access to a textual description for each timestamp, which may be difficult to obtain in real-world applications.

**Questions:**

1. How did you choose the competitors and why not incorporating multimodal algorithms?

2. How the method can adapt if not enough textual descriptions are not available for each time stamp if there are not alignment between time series and text or at least it is not known ?

3. it seems the information from the text is not complementary but redundant information with the time series as captured by the low TT-Wasserstein. In this redundant information case one would expect the text not to improve the result and expect more an improvement when the information is complementary and critical for the forecast?

---

> ### Author Response · Authors · 2025-11-22
> **Expanded Multimodal Baselines, Complex Real-World Text Scenarios, and Clarifications (Part I)**
>
> We sincerely thank the reviewer for the constructive and thoughtful feedback. If you have further questions, we would like to provide more details in the discussion phase.
>
>
> > Comparison with TimeMMD and other multimodal algorithms (W1 & Q1)
>
> We thank the reviewer for highlighting the importance of comparing with multimodal baselines. We fully agree, and this was a key motivation behind the experiments in both our original and revised submissions.
>
> (1) Comparison with TimeMMD / MM-TSFLib (already in our submission)
>
> MM-TSFLib, included as a major baseline in our Table 2, corresponds exactly to the method referred to as TimeMMD. **Both were introduced in the same paper [1], where TimeMMD denotes the dataset and MM-TSFLib is the multimodal forecasting approach proposed in that paper**. We will further clarify this distinction in the revised manuscript if needed.
>
>
> (2) Choice of Baselines
>
> In the original submission, we compared TaTS against a range of strong baselines, including standard unimodal forecasting models, several covariate-based methods, and two multimodal baselines: MM-TSFLib (TimeMMD) and ChatTime, that were publicly released and open-sourced at the time of submission.
>
> (3) Newly added comparisons during rebuttal
>
> To strengthen the empirical study, we incorporated several additional multimodal competitors. Specifically, we added comparisons with GPT4MTS [2] in Table 4 and in Table 28.
>
> - Comparison of our TaTS and GPT4MTS on Forecast with News datasets. We report average MSE and MAE metrics across various prediction lengths.
> | Method | Bitcoin Price | Web Traffic | Electricity Demand |
> |-|-|-|-|
> | TaTS + iTransformer (ours) MSE | 2.609 | 17.744 | 0.280 |
> | GPT4MTS MSE | 2.721 | 19.260 | 0.296 |
> | TaTS + iTransformer (ours) MAE | 1.112 | 2.712 | 0.396 |
> | GPT4MTS MAE | 1.152 | 2.827 | 0.395 |
>
> we also noticed some concurrent works on multimodal time series this year. Referring to their released preprints, we provide further comparisons with several very recent multimodal algorithms [3, 4, 5]. The results are summarized in Table 8 in Appendix B.1.
>
> - Comparison of our TaTS and several concurrent multimodal baselines on Time-MMD datasets. We report average MSE metrics across various prediction lengths.
> | Method | Agriculture | Climate | Health |
> |-|-|-|-|
> | TaTS + iTransformer (ours) | 0.201 | 1.227 | 1.372 |
> | TFHTS | 0.571 | 1.782 | 1.514 |
> | MCD-TSF | 0.222 | 1.583 | 1.496 |
> | TeR-TSF | 0.338 | 1.348 | 1.384 |
>
> We compute average rankings across datasets (with lower MSE corresponding to a better rank). TaTS achieves the best overall ranking of 1.83, slightly outperforming the second-best recent method, TeR-TSF, which obtains 2.08.
>
> **Overall, the revised manuscript now includes comprehensive comparisons with a broad suite of multimodal baselines: MM-TSFLib (TimeMMD), ChatTime, GPT4MTS, TFHTS, MCD-TSF, and TeR-TSF**, demonstrating the strong and often leading performance of TaTS. We hope these expanded comparisons could help address your concern regarding multimodal baselines.
>
>
> [1] Time-MMD: Multi-Domain Multimodal Dataset for Time Series Analysis. Liu et al. NeurIPS 2024.
>
> [2] GPT4MTS: Prompt-based Large Language Model for Multimodal Time-series Forecasting. AAAI 2024.
>
> [3] Unveiling the potential of text in high-dimensional time series forecasting. Preprint, 2025.
>
> [4] Multimodal conditioned diffusive time series forecasting. Preprint, 2025.
>
> [5] Text reinforcement for multimodal time series forecasting. Preprint, 2025.

---

> ### Author Response · Authors · 2025-11-22
> **Expanded Multimodal Baselines, Complex Real-World Text Scenarios, and Clarifications (Part II)**
>
> > How our method adapts to more complex real-world scenarios (W2 & Q2)
>
> We appreciate the reviewer’s insights regarding the availability, alignment, and quality of textual descriptions in real-world settings. These are indeed important considerations for deploying multimodal forecasting models as we discussed in our Limitations section (Appendix B.2.). Before addressing specific scenarios, we highlight that our TT-Wasserstein metric is designed precisely to measure the alignment between text and time series. This metric can serve as a practical diagnostic tool to decide when and how textual information should be incorporated.
>
> (a) **Texts are not available in some timestamps.** TaTS can naturally accommodate this by inserting a placeholder token such as “no information available,” as demonstrated in our ablation study on text-random-drop datasets. This allows the model to exploit available textual cues without requiring a complete sequence of text.
>
> (b) **Texts are not aligned or not sure if texts are aligned.** Small temporal shifts (leading/lagging) primarily change the phase but not the frequency of the textual patterns, so TaTS can still benefit from the preserved periodic structure. For larger or uncertain misalignments, practitioners may first improve alignment using standard preprocessing techniques and then rely on TT-Wasserstein as a diagnostic of alignment.
>
> (c) **Texts could be noisy or even misleading.** As noted by Reviewer w9od, noisy text could cause negative transfer. A practical mitigation is to stochastically drop some texts to control the influence of the text modality. The drop rate can be tuned on a validation set to identify the most effective way to use (or selectively ignore) textual descriptions in practice.
>
>
>
>
> > On text being “redundant” due to low TT-Wasserstein (Q3)
>
> We appreciate the reviewer’s careful observation regarding the low TT-Wasserstein values and the possibility that text may be redundant with the numerical time series. We clarify two key points below.
>
> (1) Low TT-Wasserstein does not imply full redundancy.
>
> A low TT-Wasserstein score indicates strong periodic alignment between the text and the time series, but it is not zero, meaning the text and the numerical values are not perfectly redundant. The text still contains auxiliary semantic cues that are not explicitly present in the raw numerical sequence. Our ablations confirm that even when alignment is strong, these additional semantic signals provide measurable performance gains.
>
>
> (2) TT-Wasserstein captures only periodic alignment, not all complementary information.
>
> TT-Wasserstein is designed to measure alignment in temporal structure (e.g., periodicity), but it does not exhaustively capture all types of complementary information. There could be other complementary information that is not related to periodicity.

---

### Author Response · Authors · 2025-11-22
**Thank You for the Review and General Response**

We sincerely thank the reviewers for their constructive and thoughtful feedback, which greatly helped us improve the manuscript. We have uploaded a revised version that we believe addresses the majority of the concerns. All modifications are highlighted in orange for ease of reference.

> In the revision, we have added the following content.

(1) **Expanded multimodal baselines.**
We added additional comparison and discussion on multimodal baselines on Page 7 Table 4, Page 43 Table 28 (GPT4MTS results), Page 24, Page 25 Table 12. The revised manuscript now includes comprehensive comparisons with MM-TSFLib (TimeMMD), ChatTime, GPT4MTS, and three concurrent works: TFHTS, MCD-TSF, and TeR-TSF. *(Addresses Reviewers vhxM and aWnj.)*

(2) **New ablation on modality-fusion strategies.**
On Page 8, as suggested by Reviewer w9od, we added a new ablation study on alternative design choices for combining modalities. *(Addresses Reviewer w9od.)*

(3) **Further analysis on mitigating negative transfer from noisy texts.**
On Page 9, we added a new analysis examining negative transfer under extremely noisy text. Following Reviewer w9od’s suggestion, we evaluated random text dropping and found it effectively reduces the influence of unreliable textual inputs. *(Addresses Reviewers w9od and 1Lkf.)*

(4) **Additional discussion on real-world low-quality or partially missing texts.**
Building on the above (3), we expanded Appendix B.2 Page 25 to discuss how TaTS can adapt to practical scenarios with missing or low-quality texts. *(Addresses Reviewers vhxM, aWnj, and 1Lkf.)*


(5) **Stable Estimation of TT-Wasserstein.**
In Appendix A.1 Page 22 23, we expanded our analysis of TT-Wasserstein to more thoroughly examine its statistical stability., we evaluate robustness across different windowing choices, normalization schemes, and frequency resolutions, and we quantify estimation uncertainty using a block-bootstrap procedure. *(Addresses Reviewer w9od.)*

(6) **Expand related works on TimeSeries->Text.**
We expanded our related-work discussion on Page 10 to include research on generating texts according to time series. The revision now highlights recent progress in temporal reasoning and time-series question answering. *(Address Reviewer aWnj.)*

(7) **Wall-clock time and complexity analysis on text embedding.**
In Appendix C.2 Page 26 27, we provided more details on the computational cost of text embedding. Specifically, we added both empirical wall-clock measurements and a formal complexity analysis. *(Address Reviewer w9od and 1Lkf.)*

(8) **Experiment details on data splitting and leakage prevention.**

In Appendix D.4.3. Page 31, we clarified the details of our leakage-preventing data-splitting protocol in the revision. *(Addresses Reviewer w9od.)*


> We provide **a quick reference of the revised pages for each reviewer**.

- Reviewer vhxM (initial score 4, confidence 3): our revision (1) (4) addresses your comments. Please refer to our orange highlights on Page 7 Table 4, Page 43 Table 28, Page 24, Page 25 Table 12;  Appendix B.2 Page 25.

- Reviewer w9od (initial score 6, confidence 3): our revision (2) (5) (8) (3) (7) addresses your comments. Please refer to our orange highlights on Page 8; Appendix A.1 Page 22 23; Appendix D.4.3. Page 31; Appendix C.2 Page 26 27.

- Reviewer aWnj (initial score 6, confidence 5): our revision (4) (6) (1) addresses your comments. Please refer to our orange highlights on Appendix B.2 Page 25; Page 10; Page 7 Table 4, Page 43 Table 28, Page 24, Page 25 Table 12.

- Reviewer 1Lkf (initial score 6, confidence 4): our revision (7) (3) (4) addresses your comments. Please refer to our orange highlights on Appendix C.2 Page 26 27; Page 9; Appendix B.2 Page 25


We once again thank all reviewers for their time and insightful feedback. We are happy to address any additional questions or clarifications during the discussion phase.

---

### Author Response · Authors · 2025-12-03
**Summary for Area Chair**

Dear Area Chair,

We sincerely appreciate your time and effort in overseeing the review process for our submission. Given the exceptional circumstances this year and the substantial additional workload on ACs, we have prepared a brief summary of our rebuttal to support and streamline your decision-making.

### **Strength of our work appreciated by reviewers**

Three reviewers provided positive ratings prior to the rebuttal. Our scores for soundness/presentation/contribution were 2/3/3/3, 3/2/3/3, 2/3/3/3, respectively. The main concerns raised by Reviewer vhxM, who assigned a score of 4 before rebuttal, focused on (1) adding multimodal baselines and (2) discussing practical strategies for adapting our method to more complex real-world scenarios. We note that our original submission already included comparisons with the requested MM-TSFLib baseline, and that “TimeMMD” refers to the dataset introduced in the same work as MM-TSFLib. During the rebuttal, we have thoroughly addressed both concerns through additional comparisons and expanded discussions.

- Our paper introduces a simple, model-agnostic, plug-and-play framework for integrating textual information into time series forecasting, addressing a timely and practically important multimodal modeling problem *(by Reviewers vhxM, w9od, aWnj)*.

- The paper provides a clear and interpretable alignment rationale through the CTR notion and the TT-Wasserstein diagnostic, offering insight into when and why text improves forecasting and how practitioners can deploy multimodal models effectively *(by Reviewer w9od, Reviewer aWnj)*.

- The framework is well-motivated and presents a coherent structure connecting CTR (motivation), TaTS (mechanism), and TT-Wasserstein (diagnostic) *(by Reviewers aWnj, 1Lkf)*.

- Extensive experiments across diverse datasets and backbones with clear ablations demonstrate strong, and robust improvements. *(by Reviewers vhxM, w9od, aWnj, 1Lkf)*.

- The approach is computationally efficient, adding only modest overhead despite substantial performance gains *(by Reviewers w9od, 1Lkf)*.


### **Raised concerns and how we address them**

**Details on this can be found in our General Response**. We appreciate all the reviewers’ efforts in reviewing this manuscript and provided point-by-point rebuttals to each reviewer. We added 5 pages of additional analysis, experiments, and discussion to incorporate the suggestions from reviewers and to further enhance the comprehensiveness of the paper.

The main shared concerns across reviewers were:

(1) the need for additional multimodal baselines. In this rebuttal, we have expanded our comparisons to include more multimodal baselines. *(the "(1)" in General Response*)

(2) strategies for handling real-world low-quality or partially missing texts. Following Reviewer w9od’s suggestion, we evaluated text dropping and found that it effectively mitigates the impact of unreliable textual inputs. In addition, we have expanded Appendix B.2 to discuss how TaTS can be adapted to practical deployment scenarios. *(the "(3)(4)" in General Response*)

### **Summary of discussion phase**

After we posted our rebuttal responses, none of the reviewers provided further comments or follow-up discussion before the identity leakage occurred on OpenReview.


### **Summary of manuscript change**

Following the reviewers’ suggestions, we have revised the manuscript accordingly, adding 5 additional pages in total. **All modifications are documented in our General Response**, where we explicitly specify the corresponding updated pages.


### **Declaration regarding the identity leakage incident**

We were unaware of the identity leakage until after the OpenReview bug had been fixed, and we did not make use of the bug at any point. We confirm that we did not know the identity of any reviewer throughout the entire review process.



Once again, we want to express our gratitude to you and all reviewers for evaluating our manuscript and helping us improve it.


Best regards,
Authors of submission 8212

---

### Meta-Review · Area_Chair_P3XP · 2026-01-06

**Summary:**

This paper presents TaTS, a "plug-and-play" framework designed to improve time series forecasting by incorporating related text data (like news articles or captions). The core idea is that text often follows the same seasonal or periodic patterns as the numerical data it describes. The reviewers generally liked that the method is simple to use and doesn't require changing the architecture of existing time series models. It was tested across nine different real-world domains and consistently showed better results than models using only numbers.

**Reviewer Concerns:**

During the discussion, reviewers initially worried about how TaTS would compare to other multimodal methods and how it would handle "messy" real-world situations, such as when text is missing or contains errors. The authors provided a very strong response by adding several new benchmarks (like GPT4MTS and TFHTS) and demonstrating that TaTS remains a top performer. They also showed that using "placeholder" tokens or "text-dropping" strategies can effectively protect the model from being confused by low-quality or irrelevant text. Other technical questions regarding the statistical stability of their alignment metric (TT-Wasserstein) were also clearly resolved with new experimental data.

**Reviewer Scores:**

Reviewer vhxM (score 4) primarily questioned the lack of multimodal baselines and real-world robustness; because the authors added multiple new competitors (like GPT4MTS and TFHTS) and a noise-mitigation strategy, it is highly likely this reviewer would have raised their score to at least a 6. The other three reviewers (w9od, aWnj, and 1Lkf) were already leaning toward acceptance (score 6) and had their requests for ablation studies and complexity analyses fully satisfied in the updated manuscript.

---

### Decision · Program_Chairs · 2026-01-26

Accept (Poster)

---

> ### Public Comment · ~Yu-Cheng_Wu1 · 2026-05-09
>
> Methods like PatchTST and DLinear treat each variable independently without cross-variable interaction, so how can they benefit from the extra text variables, as in Table 2, when their outputs are also not included in the loss calculation?